# Emergent constraints on the hydrological impacts of land use and land cover change

Zefeng Chen [1] ✉, Alessandro Cescatti [2], Ruofei Xing [3,4] & Giovanni Forzieri [1]

Land use and land cover changes have substantial effects on the terrestrial water cycle, but their sign and magnitude remain elusive at large scales. State-of-the-art Earth system models disagree on how these changes affect terrestrial evapotranspiration. Here we use the observation-based transpiration-specific Bowen ratio to correct modelled evapotranspiration changes induced by land use and land cover changes globally and regionally within a hierarchical emergent constraint framework. We show that the constraint reverses the sign of the original model estimates at the global scale and over Central and South America, and narrows the inter-model spread. The misrepresentation of transpiration-specific Bowen ratio and its variations across plant functional types in models is the main source of this bias. Applying an analogous constraint framework to a future afforestation scenario, the constrained simulations project stronger evapotranspiration enhancements and weaker decreases in terrestrial water availability compared to the original simulations, particularly in tropics and subtropics.

About three-quarters of the Earth's land surface has been altered by human activities like afforestation, deforestation for agricultural expansion, grazing and urbanization[1,2]. The effects of such anthropogenic land use and land cover change (LULCC) on the terrestrial water cycle are considered of comparable magnitude to those originating from climate change[3–7]. LULCC directly influences the scale and timing of terrestrial evapotranspiration (ET) due to the inherent differences between plant functional types (PFTs) in growing-season length, root development, leaf traits, and water-use efficiency[4,8,9]. In addition, it contributes to indirectly controlling precipitation (P) patterns locally and remotely in downwind regions through moisture recycling[5,10]. Such LULCC-driven processes may lead to shifts in soil moisture and total terrestrial water availability (WA, defined as P minus ET)[5,9] and trigger cascading detrimental or beneficial impacts on socio-ecological systems and services crucial for the human well-being, such as biodiversity conservation[11], carbon uptake and storage[12,13], food security and timber production[14]. Furthermore, afforestation, one of the most obvious manifestations of LULCC, has a strong capacity to

absorb atmospheric $CO_2$ and is considered a key solution to meet climate targets[15,16]. But the scalability of this land use management option has recently been questioned due to its potential threat to water supply for societies[17,18]. Therefore, a robust and accurate assessment of the long-term effects of LULCC scenarios on the terrestrial water cycle is of paramount importance for developing sustainable water management and more integrated and realistic land-based climate mitigation strategies.

Numerous studies based on in-situ hydrological measurements have explored the ongoing changes in the terrestrial water cycle in response to LULCC but their results are elusive and conclusions are still largely controversial[19]. Previous regional and catchment-scale studies have led to contradictory findings, i.e., that forest gain has either a negative[20], negligible[21], or positive effect[22,23] on water yield. These results suggest that the hydrological response to forest change may vary greatly from one catchment to another[24] due to different background climate, geomorphological characteristics and soil properties[25]. Furthermore, it is unclear to what extent the results of

[1]Department of Civil and Environmental Engineering, University of Florence, Florence, Italy. [2]European Commission, Joint Research Centre, Ispra, Italy. [3]National Key Laboratory of Water Disaster Prevention, Hohai University, Nanjing, China. [4]Business School, Hohai University, Nanjing, China. ✉e-mail: zefeng.chen@unifi.it

such analyses based on ground-based hydrological observations can be extrapolated to larger areas[26,27]. Repeated and consistent satellite observations can provide a more systematic characterization in space and time of the terrestrial water cycle at the regional to global scales[28]. However, such observational products can only partially capture the non-local biophysical effects driven by changes in large-scale circulation and the associated impacts on cloud cover and P typically associated to large-scale LULCC[29,30]. Earth system models (ESMs) as well as land surface models (LSMs, i.e., the land component of ESMs) can provide a comprehensive assessment of the hydrological response to LULCC at the global scale by integrating local and non-local effects[31,32]. Nevertheless, their simulations lack consensus on the direction and magnitude of the response of key hydrological variables (e.g., ET, P, and runoff) to LULCC, because of the highly heterogeneous representation of key biological and physical processes across ESMs/LSMs[33,34]. Results from a recent modelling study focusing on the hydrological response to a global deforestation experiment emphasize the substantial uncertainty in hydrological projections under LULCC[27]. This is particularly critical considering the strong coupling between terrestrial water and carbon cycles[12,35,36]. Indeed, the resulting uncertainties in projections of hydrological response to LULCC largely hamper a proper quantification of the associated impacts on the terrestrial carbon budget and may ultimately limit the identification and development of effective land-based climate mitigation policies.

Here we use multi-source observations to enhance the representation of the modelled LULCC effect on terrestrial evapotranspiration ($\delta ET^{LULCC}$) within a hierarchical emergent constraint framework[37]. Factorial simulations of an ensemble of ESMs participating in the Coupled Model Intercomparison Project Phase 6 (CMIP6)[38] are used to disentangle the $\delta ET^{LULCC}$ signal during the period 1982-2014 at the global scale (Supplementary Tables 1–3). We apply such an emergent constraint framework to future scenarios to derive more accurate projections of afforestation impacts on terrestrial ET ($\delta ET^{AFF}$) and WA ($\delta WA^{AFF}$) over 2015-2099. We extend the constraint analyses over the reference regions[39] used in the Intergovernmental Panel on Climate Change Sixth Assessment Report (IPCC AR6), to provide more spatially detailed information about the LULCC-driven hydrological changes and better assist water management strategies and land-based mitigation plans. Finally, we investigate the dependence of the modelled LULCC impact on the surface energy partitioning across PFTs to elucidate the underlying mechanisms responsible for the inter-model spread.

## Results

### Large disagreement among ESMs on the LULCC hydrological effects

The CMIP6 model ensemble – under prescribed land use and land cover forcing from the LUH2 dataset[40,41] – shows that global forest area has decreased significantly ($p < 0.01$, Mann-Kendall test) by $0.018 \pm 0.008$ million km$^2$ yr$^{-1}$ (or $0.016 \pm 0.007$ % yr$^{-1}$ in area fraction) during the period 1982-2014 (Fig. 1a). Over the same period, global cropland area has increased significantly ($p < 0.01$) by $0.017 \pm 0.002$ million km$^2$ yr$^{-1}$ (or $0.014 \pm 0.002$ % yr$^{-1}$), whereas global grassland area first has experienced an increase ($0.012 \pm 0.010$ million km$^2$ yr$^{-1}$ or $0.010 \pm 0.009$ % yr$^{-1}$, $p < 0.01$) and then a decrease ($-0.019 \pm 0.010$ million km$^2$ yr$^{-1}$ or $-0.016 \pm 0.010$ % yr$^{-1}$, $p < 0.01$) with a change point around year 2000 (Fig. 1a). Overall, global land has experienced a massive conversion from forests to croplands during the past three decades. The inter-model spread in the trends of forest, cropland and grassland area (here expressed in terms of standard deviation) can largely be attributed to the difference in the assumptions embedded within the land-use translator of the ESMs (the program that translates the original LUH2 dataset into cover fractions of trees, crops, and grasses)[40]. In addition, ESMs such as CESM2 and UKESM1-0-LL explicitly consider the competition between natural vegetation types and

these effects may further amplify their difference in terms of forest and grassland areas with respect to other ESMs[42,43].

According to an ensemble of historical simulations from twelve CMIP6 models, the LULCC – generally characterized by the conversion from forests to croplands over the period 1982-2014 – has increased terrestrial ET globally ($\delta ET^{LULCC}$) by an average rate of $0.057 \pm 0.155$ mm yr$^{-2}$ (Fig. 1b). Such positive effects on ET are counter-intuitive with regard to the widespread reduction in forest cover during this period (Fig. 1a), raising further concerns about the capacity of ESMs in properly capturing ET dynamics in response to LULCC[27]. Moreover, the standard deviation of the ensemble is 2.5 times the mean value and further underscores the substantial uncertainty and the large discrepancy amongst single ESM runs. Only seven out of twelve individual ESMs agree on the overall positive sign of $\delta ET^{LULCC}$, and the spread in magnitude is considerable (global average signal ranging from $0.005$ mm yr$^{-2}$ to $0.380$ mm yr$^{-2}$, Fig. 1b). Similar patterns emerge in terms of the relative contribution of LULCC on historical change in terrestrial ET ($13.3 \pm 33.4$%, Supplementary Fig. 1). These results highlight the lack of consensus in direction and magnitude of $\delta ET^{LULCC}$ across state-of-the-art ESM simulations when the signal is aggregated at the global scale.

The spatially aggregated estimates described above largely mask the variability in the LULCC effect across different regions of the Planet. The ensemble mean $\delta ET^{LULCC}$ is positive in Eastern and Southern Africa, and India, whereas it is negative in Australia, Southern Siberia, and South America (Fig. 1c, d). We point out that the ensemble standard deviation of $\delta ET^{LULCC}$ is more than 5 times the ensemble mean value over 44.7% of the global vegetated land. In contrast, it appears lower than the ensemble mean only in about 0.6% of the domain (Fig. 1e). Such remarkable inter-model spread is more pronounced in high-latitude regions of the Northern Hemisphere (50°N-65°N) and over the tropics of the Southern Hemisphere (5°S-10°S) (Fig. 1e, f). Estimates of mean and standard deviation of $\delta ET^{LULCC}$ obtained from different model ensembles and different formulations (full ESM ensemble vs. LUMIP ESM ensemble, details in Methods) exhibit consistent spatial patterns (Fig. 1d, f). Altogether, these results unequivocally show substantial uncertainties in ESM-based assessment of the hydrological response to land use and land management practices and call for additional analysis to improve predictions.

### Emergent constraints on modelled global LULCC effects over the historical period

To minimize the uncertainty and further improve the robustness of ESM-based assessment on the hydrological effect of LULCC, we propose the use of a hierarchical emergent constraint[37]. To this aim, we combine the CMIP6 modelled global $\delta ET^{LULCC}$ during the historical period, for which no direct observations exist, with the corresponding global-averaged mean annual ratio between sensible heat and latent heat through transpiration (Tr) (represented by a Tr-specific Bowen ratio ($B_{ts}$), details in Methods), for which observational estimates are available. Observational estimates of $B_{ts}$ are obtained based on the combination of four observation-based Tr/ET products and two reanalysis climate datasets (details in Methods). Specifically, the observation-based Tr/ET estimates used here are derived from the LAI-based upscaling approach[44], and remote sensing models[45-47], in combination with 20CR v3[48] and ERA5-Land climate datasets[49]. This emergent constraint approach grounds on the tight linear positive relationship across ESMs ($r = 0.931$, $p < 0.01$) between the modelled global $\delta ET^{LULCC}$ and the natural logarithm of modelled global $B_{ts}$ (i.e., $\ln(B_{ts})$) derived for the observational period (1982-2014) (Fig. 2a). Results derived from one-thousand random sampling further confirm the robustness and significance of the inter-model relationship between $\delta ET^{LULCC}$ and $\ln(B_{ts})$ (Supplementary Fig. 2a, b; details in Supplementary Text 1). Similar to the model-based estimates of $\delta ET^{LULCC}$ (Fig. 1), the global mean $\ln(B_{ts})$ also exhibits substantial

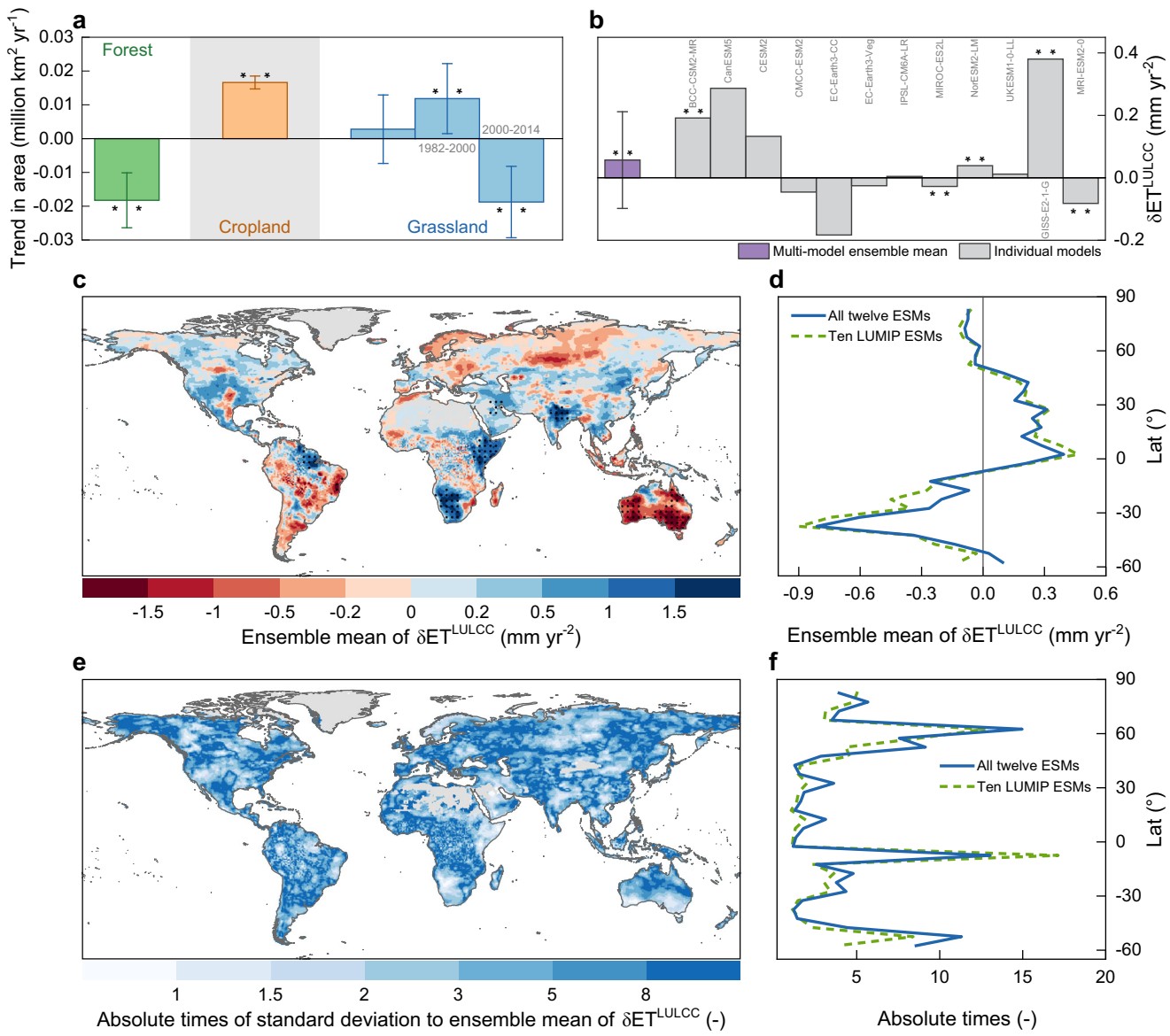

**Fig. 1 | Modelled effect of land use and land cover change on terrestrial evapotranspiration and its inter-model difference. a** Global mean trends in forest area (green), cropland area (yellow), and grassland area (blue) during the period 1982–2014, as derived from the CMIP6 model ensemble driven by the land use forcing dataset LUH2. For grassland area, its trends during the subperiods 1982–2000 and 2000–2014 are also provided. Error bars represent the standard deviation of trends derived from ensemble members. Two asterisks indicate that the trend is statistically significant with $p < 0.01$. Significance of the trend is assessed by Mann-Kendall test. **b** Global mean effect of land use and land cover change on terrestrial evapotranspiration ($\delta ET^{LULCC}$) during the period 1982–2014, as derived from 12 ESMs and multi-model ensemble mean by Eqs. (1) and (2). Error bars represent the standard deviation of effects derived from ensemble members. Two asterisks indicate that the effect is statistically significant ($p < 0.05$). Significance of the effect is assessed by $t$-test. **c** Spatial pattern of $\delta ET^{LULCC}$ estimate by

multi-model ensemble mean. Regions labelled by black dots indicate the effect that is statistically significant ($p < 0.05$, $t$-test). Dots are spaced 3° in both latitude and longitude, and statistics were computed over 9° × 9° spatial moving windows. Non-vegetated areas, defined as multi-year (1982-2014) average leaf area index (LAI) < 0.15 m$^2$ m$^{-2}$, are excluded in our analysis and are shown in grey. **d** Zonal mean of $\delta ET^{LULCC}$ estimate by multi-model ensemble mean at 5° latitudinal resolution. The blue solid line represents the result based on full 12 ESMs, while the green dashed line represents the result based on the ten ESMs participating in LUMIP project (i.e., GISS-E2-1-G and MRI-ESM2-0 are excluded). **e**, **f** Same as (**c**, **d**), but for absolute times of standard deviation of $\delta ET^{LULCC}$ estimate across ESMs to associated ensemble mean value. ArcGIS Pro was used as technical tool for data processing and visualizing this figure. However, no ESRI basemaps or proprietary ESRI datasets, and no screenshots of the ArcGIS software interface were used.

difference across ESMs, ranging from 0.352 to 1.360 (Fig. 2a, c). Such emergent relationship shown above reveals that ESMs with high $\ln(B_{ts})$ values generally have a large LULCC-induced ET increase, while ESMs with low $\ln(B_{ts})$ values show a weak increase or even decrease in ET under same LULCC conditions (all CMIP6 ESMs adopt the same land-use forcing dataset LUH2[41]) (Fig. 2a). Within the proposed modelling framework, $B_{ts}$ is transformed to its natural logarithm (1) to better reflect the inherent non-linear relationship between $B_{ts}$ and $\delta ET^{LULCC}$ (Supplementary Figs. 3a and 4; details in Supplementary Text 2), and

(2) to meet the linear assumption of emergent constraints[50]. This emergent relationship provides a robust and effective constraint to the wide range of modelled $\delta ET^{LULCC}$ with the global $\ln(B_{ts})$ values from an eight-member ensemble of observation-based datasets (four Tr/ET products × two climate datasets).

Emergent constraint results show that historical LULCC has led to a decrease in ET with a global average rate of -0.031 mm yr$^{-2}$ during the period 1982-2014, which is opposite in sign to the original CMIP6 ensemble mean (0.057 mm yr$^{-2}$) (Fig. 2b). After applying the emergent

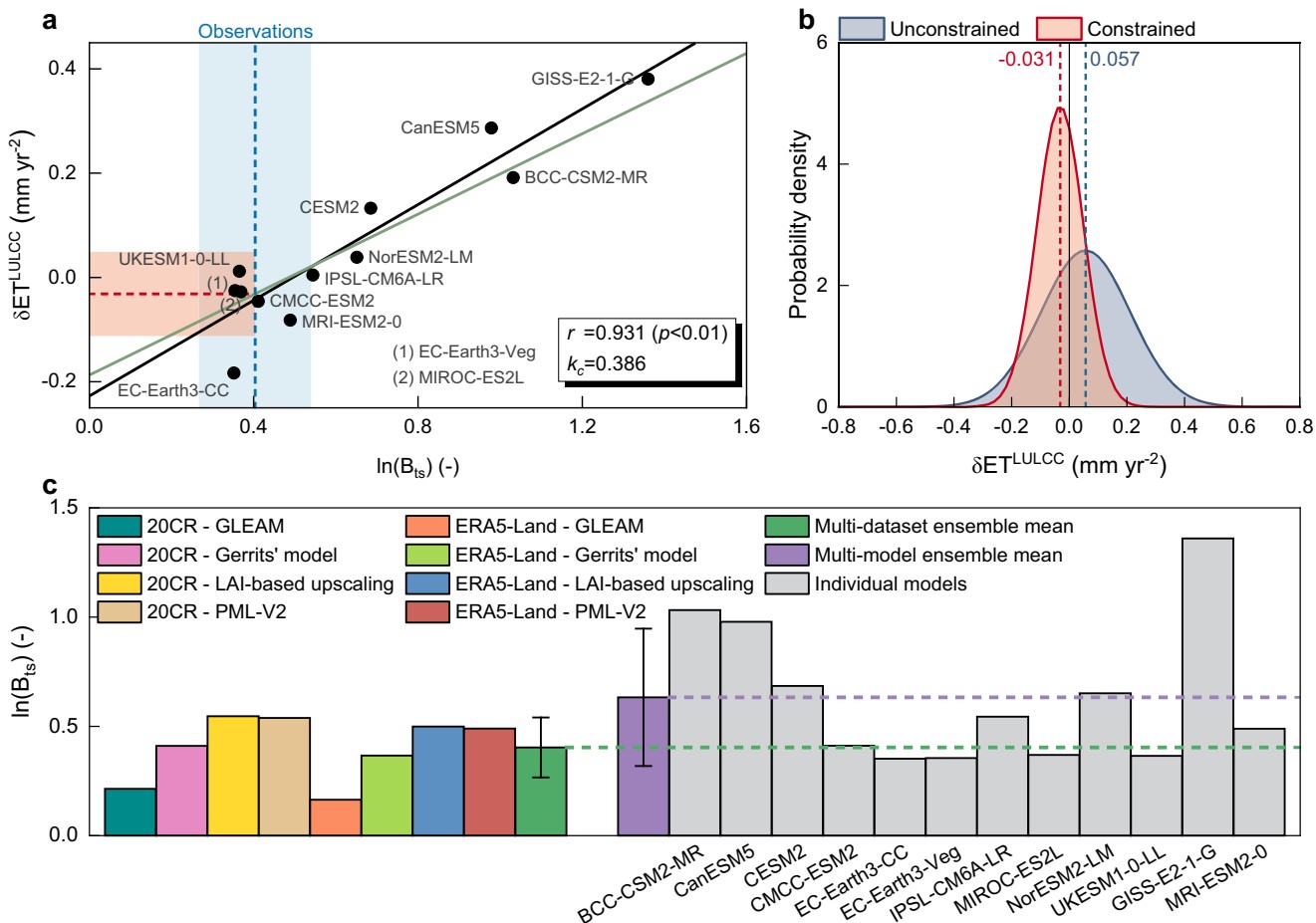

**Fig. 2 | A constraint on the effect of land use and land cover change on terrestrial evapotranspiration. a** Emergent relationship between the modelled effect of land use and land cover change on annual evapotranspiration ($\delta ET^{LULCC}$) and the modelled natural logarithm value of global averaged transpiration-specific Bowen ratio ($\ln(B_{ts})$) during the period 1982–2014. LULCC over this period is generally characterized as the conversion from forests to croplands (Fig. 1a). Each dot denotes a CMIP6 ESM result, which corresponds to $\delta ET^{LULCC}$ estimate shown in Fig. 1b. The black solid line indicates the best-fit regression line across ESMs, with correlation coefficient ($r$) provided in the panel. The green solid line indicates the observational correction based on Eq. (5) with associated slope ($k_c$) provided in label. The vertical blue dashed line and shaded areas represent the observation-based estimate of $\ln(B_{ts})$ and its uncertainty (one standard deviation). Such

observation-based estimate is derived from an eight-member ensemble of observation-based combined datasets (Methods). The horizonal red dashed line and shaded areas show the resulting constrained estimate of $\delta ET^{LULCC}$ and its uncertainty based on the hierarchical emergent constraint approach (Methods). **b** The probability density functions of global averaged $\delta ET^{LULCC}$ for the original results of CMIP6 ESMs (blue) and the observationally constrained results (red). **c** Global averaged $\ln(B_{ts})$ during the period 1982–2014, as derived from multiple observation-based combined datasets and on CMIP6 ESMs and their ensemble mean. Error bars represent the standard deviation of $\ln(B_{ts})$ estimates derived from ensemble members (i.e., eight combined datasets for multi-dataset ensemble mean; twelve individual ESMs for multi-model ensemble mean), respectively.

constraint, the overall transition from positive to negative $\delta ET^{LULCC}$ during the past three decades results from the combination of (1) the positive relationship between the effect of LULCC (characterized as forest-cropland conversion globally) and $\ln(B_{ts})$ and (2) the overall overestimation of $B_{ts}$ in current-version ESMs (modelled $\ln(B_{ts})$: $0.633 \pm 0.314$; observed $\ln(B_{ts})$: $0.404 \pm 0.137$) (Fig. 2a, c). The latter bias is largely attributable to the substantial ESM underestimation of the ratio of Tr to total terrestrial ET (i.e., Tr/ET in Supplementary Fig. 5c). The negative value of $\delta ET^{LULCC}$ obtained after constraint is highly consistent with previous findings of observation-based assessments on global ET change[4,51], and in accordance with the previously reported decrease in latent heat induced by the conversion of forests to croplands[52]. The emerging negative signal of $\delta ET^{LULCC}$ over the observational period can be partly attributed to the massive deforestation in the tropics and the induced decrease in Tr and canopy interception that exceed any potential increase in soil evaporation[8,53]. Furthermore, the constrained distribution shows a standard deviation of $0.081 \text{ mm yr}^{-2}$, which corresponds to a significant ($p < 0.05$, F-test)

reduction in model spread by 48.0% compared to the unconstrained (original) CMIP6 ensemble (Fig. 2b).

A comprehensive set of experiments performed on the selection of the model ensemble (Supplementary Table 4) and on the source of observational data including the subsets from the full ensemble of observation-based datasets (Supplementary Table 5) and the eddy covariance measurements at FLUXNET sites[54,55] (Supplementary Fig. 6 and Data 1 and 2), further confirms the robustness of our results and the validity of our methodology (details in Supplementary Text 1). Moreover, additional analyses based on the classic emergent constraint[56,57] show a consistent sign switch in $\delta ET^{LULCC}$ after constraint and a comparable magnitude of the constrained $\delta ET^{LULCC}$ ($-0.048 \pm 0.089 \text{ mm yr}^{-2}$), aligning with the estimates derived from the hierarchical emergent constraint described above (Fig. 2 and Supplementary Fig. 7). Such high consistency demonstrates the substantial independence of our results on the type of emergent constraint method applied and further reinforces the robustness of our results (details in Supplementary Text 3).

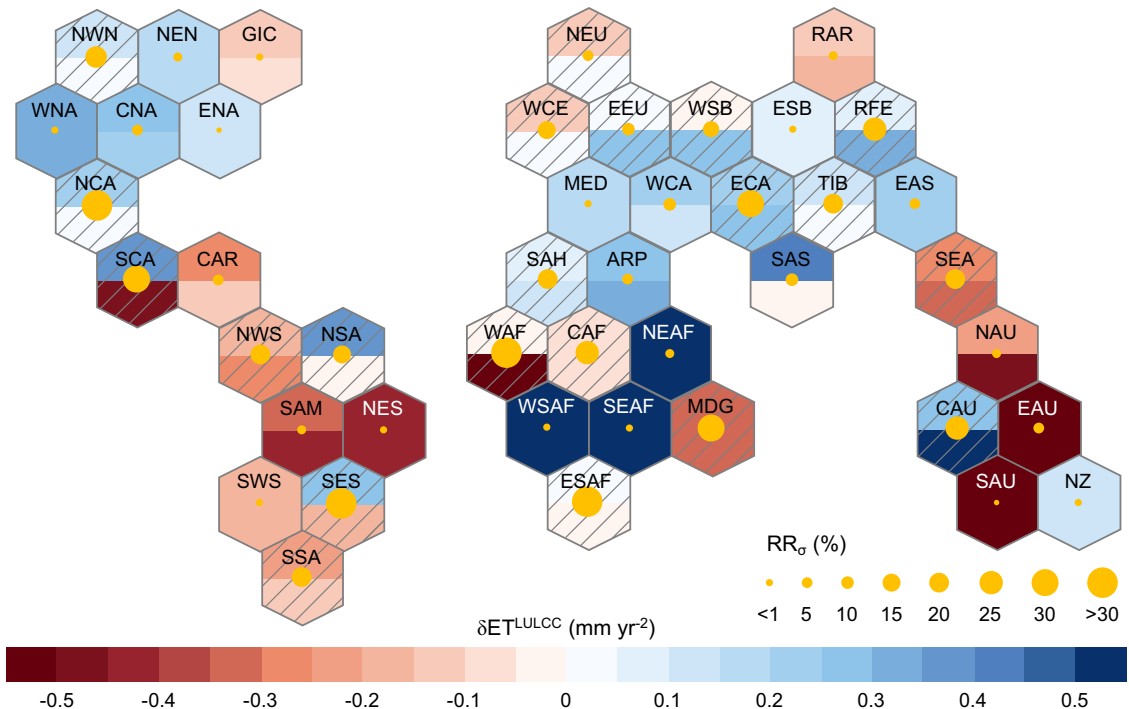

**Fig. 3 | Constrained effect of historical land use and land cover change on terrestrial evapotranspiration over IPCC AR6 reference regions.** Within each hexagon, the filled color in the upper and lower parts represent the unconstrained estimate from the original CMIP6 model ensemble and the observationally constrained estimate, respectively. The size of the yellow dot indicates the relative reduction in standard deviation (RR$_\sigma$) after applying the hierarchical emergent constraint approach. The hexagons with the diagonal lines indicate that the emergent relationship is statistically significant at the 90% confidence level. In fact, for these hexagons, their emergent relationship's significance even passes the 95% confidence level. Specific values for making this plot are listed in Supplementary Data 3. Moreover, detailed information of these reference regions is provided in Supplementary Fig. 8.

## Emergent constraints at the regional scale

Global-scale assessments illustrated in the previous section clearly show the effectiveness of the emergent constraint approach in correcting the LULCC-related signal derived from the original CMIP6 ESM simulations and in reducing its uncertainty (Fig. 2). Nevertheless, when emergent constraints are applied at the global scale, they are potentially subject to spatial compensatory effects that may originate from areas with diverse land use changes or opposite sensitivities in the land response to environmental drivers. To better elucidate such potential spatial heterogeneity in the response mechanism, we further apply the proposed emergent constraint framework at the regional scale (Fig. 3). To this end, we estimate the observationally constrained hydrological response to historical LULCC over forty-four IPCC AR6 reference regions (Supplementary Fig. 8, source from Iturbide et al.[39]) (details in Methods). Unconstrained results based on the original simulations from CMIP6 model ensemble generated for the historical period suggest that the LULCC-related signal is much stronger at the regional scale (magnitude ranging from -1.376 ± 1.480 mm yr$^{-2}$ to 0.886 ± 0.967 mm yr$^{-2}$ across reference regions, Fig. 3 and Supplementary Data 3) compared to the global mean level (0.057 ± 0.155 mm yr$^{-2}$, Fig. 1b). According to such simulations, LULCC contributes more than 25% of the historical change in local ET over 49.5% of global land. In hotspot regions experiencing dramatic forest loss like Southeastern South America (SES), Madagascar (MDG), and Southeast Asia (SEA), LULCC plays a dominant role in local ET decrease (relative contribution higher than 50%) (Supplementary Data 3). These results highlight the key relevance of land use and land cover change in long-term variations of the regional water cycle.

We find that applying the emergent constraint always reduces the uncertainty in estimates of ET response (i.e., δET$^{LULCC}$), although the degree of reduction varies across regions (Fig. 3 and Supplementary Data 3). The inter-model correlation between δET$^{LULCC}$ and ln(B$_{ts}$) is statistically significant ($p < 0.05$) for twenty-one reference regions corresponding to 50.2% of the global vegetated land (hexagons with the diagonal lines in Fig. 3). These regions exhibit an average reduction in standard deviation after constraint by 21.7% during the historical period, more than 10-fold greater than the analogous estimates obtained for the remaining regions without significant correlations (1.8%) (Supplementary Data 3).

Correction direction for estimates of δET$^{LULCC}$ varies spatially, primarily because of the distinct types of land cover conversion occurring across regions during the period 1982-2014 and of their interplay with the region-specific background climate. Over 50.2% of global vegetated land where the emergent relationship holds, areas characterized by an increase in forest area (17.0% of global vegetated land), generally shows a higher δET$^{LULCC}$ after applying the emergent constraint compared to the unconstrained estimate, which is opposite to the global mean results (Fig. 3). By contrast, regions that have experienced a decrease in forest area generally show a constrained estimate of δET$^{LULCC}$ lower than that one obtained from the unconstrained outputs, consistently with the global mean results (Figs. 2 and 3). In this respect, δET$^{LULCC}$ over 13.2% of global land turns from positive to negative after applying the emergent constraint (Fig. 3 and Supplementary Data 3). Unlike the counter-intuitive positive effect in these regions from the original ESM simulations, the constrained results of the negative δET$^{LULCC}$ occurring in regions such as Southern Central America (SCA), Northern South America (NSA) and SES are highly consistent with findings reported in previous literature[8,58]. Overall, these results demonstrate that original ESM outputs tend to misrepresent the direction and magnitude of historical LULCC effect on regional ET (Fig. 3), and moreover, confirms the effectiveness of emergent constraint approach in correcting the regional signal of LULCC effect derived from the original ESM simulations.

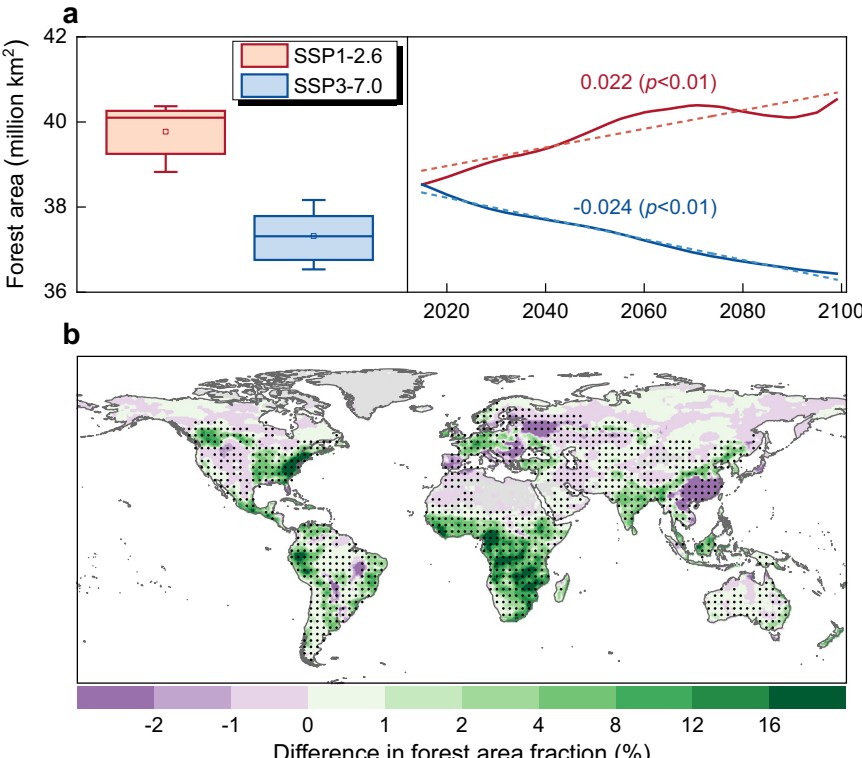

**Fig. 4 | Projected forest area change during 2015–2099 under the SSP1-2.6 and SSP3-7.0 scenarios. a** Boxplot of global annual average forest area during 2015–2099, as derived from CMIP6 model ensemble mean under the SSP1-2.6 (red) and SSP3-7.0 (blue) scenarios, respectively. Boxplot elements: box = values of 25th and 75th percentiles; horizontal line = median; rectangle = mean; whiskers = values of 10th and 90th percentiles. Interannual changes in global forest area over 2015–2099 under the two scenarios. Dashed lines indicate linear regressions of annual forest area series, with labels reporting the overall trends. Statistical significance of trends is assessed by Mann-Kendall test. **b** Spatial pattern of difference in mean annual forest area during 2015–2099 between the two scenarios (SSP1-2.6 versus SSP3-7.0). Regions labelled by black dots indicate differences that are statistically significant ($p < 0.05$, $t$-test). Dots are spaced 3° in both latitude and longitude, and statistics are computed over 9° × 9° spatial moving windows. ArcGIS Pro was used as technical tool for data processing and visualizing this figure. However, no ESRI basemaps or proprietary ESRI datasets, and no screenshots of the ArcGIS software interface were used.

## Improved hydrological impact assessment under future afforestation scenarios

Afforestation, one of the most important climate-related LULCC activities, is considered a nature-based and cost-effective strategy for enhancing terrestrial carbon sequestration and therefore is a key element in the portfolio of mitigation actions of many nations[15,16]. Despite the great potential for carbon neutrality and climate mitigation, afforestation may also pose additional pressures on WA for environment and human society[17,18]. It is therefore of critical importance to provide robust estimates of the potential hydrological impact that may originate from future afforestation scenarios. Global forest area is projected to increase significantly ($p < 0.01$, Mann-Kendall test) during 2015-2099 under the SSP1-2.6 scenario and to decline under the SSP3-7.0 scenario (Fig. 4a). Accordingly, forest area in SSP1-2.6 is higher than that in SSP3-7.0 over most of global land (Fig. 4b), reflecting the former's afforestation-oriented sustainability policies versus the latter's limited climate governance[41,59] (details in Methods). To disentangle the potential hydrological impact of afforestation scenarios, we quantify global $\delta ET^{AFF}$ from original model projections as the difference in ET trend generated under two future scenarios driven respectively by SSP1-2.6 and SSP3-7.0 land use and land cover forcing (Eq. (3), details in Methods) and analyze its inter-model relationship with $\ln(B_{ts})$. We find that $\delta ET^{AFF}$ during 2015-2099 exhibits a significantly negative inter-model relationship with modelled historical (1982-2014) $\ln(B_{ts})$ at the global scale ($r = -0.878$, $p < 0.01$), demonstrating the applicability of the emergent constraint approach also for a future afforestation scenario (Supplementary Fig. 9a). ESMs with low $\ln(B_{ts})$ values generally

have a large ET increase induced by future afforestation, while ESMs with high $\ln(B_{ts})$ values show a weak increase or even decrease in ET under the same afforestation scenario. The opposite relationship of $\ln(B_{ts})$ with future $\delta ET^{AFF}$ and historical $\delta ET^{LULCC}$ is because LULCC during the period 1982-2014 is characterized by a progressive forest loss, therefore with an obvious opposite effect on ET with respect to that one originating from the afforestation scenario considered here (Figs. 1a, 2a and 4a and Supplementary Fig. 9a).

The original CMIP6 model ensemble shows an overall positive effect of afforestation on terrestrial ET during 2015-2099, whose global average rate increases from $0.024 \pm 0.048$ mm yr$^{-2}$ to $0.051 \pm 0.031$ mm yr$^{-2}$ after applying the emergent constraint (Supplementary Fig. 9b) (details in Supplementary Text 4). Similar to results derived for the historical period, more pronounced signals emerge at regional scale under the future afforestation scenario. For example, compared to the global mean level ($0.024 \pm 0.048$ mm yr$^{-2}$, Supplementary Fig. 9b), unconstrained projections suggest a much higher $\delta ET^{AFF}$ in Central Africa (CAF) whose magnitude reaches $0.420 \pm 0.515$ mm yr$^{-2}$, indicating that afforestation could explain 74.1% of future change in local ET (CAF in Fig. 5a and Supplementary Data 4). Under the simulated future afforestation scenario, conversions from croplands to forests are expected to occur in most regions (Supplementary Data 4, exception cases are explained in Methods), generally leading to a higher constrained estimate of $\delta ET^{AFF}$ than the analogous metric derived from the original model ensemble (Fig. 5a). The constrained value of $\delta ET^{AFF}$ differs greatly across afforested regions, ranging from $-0.158 \pm 0.357$ mm yr$^{-2}$ to $0.447 \pm 0.171$ mm yr$^{-2}$ (Fig. 5a).

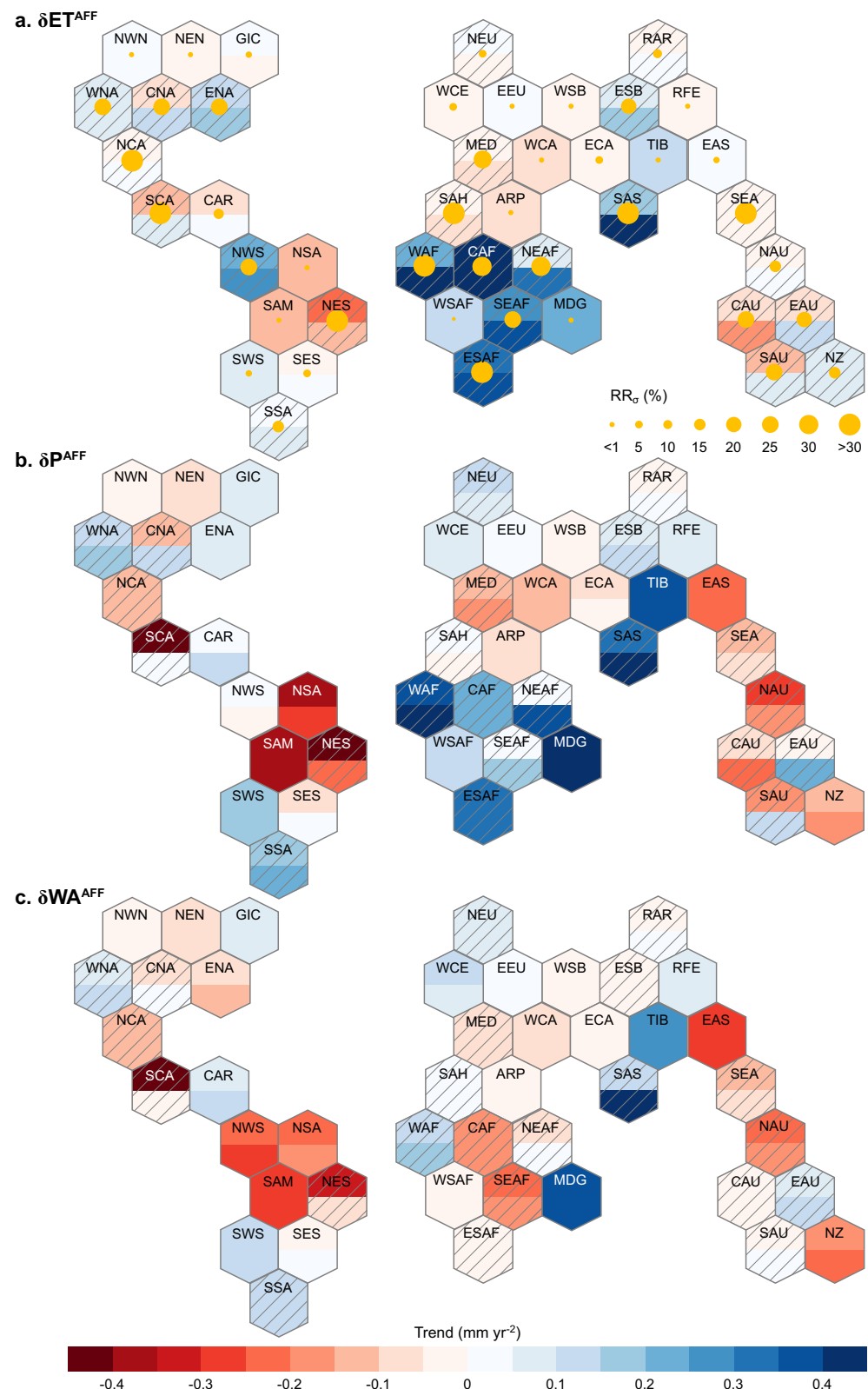

Occurrence of such spatial difference in $\delta ET^{AFF}$ may arise from (1) the varying magnitude of increase in forest cover (Supplementary Data 4), and (2) the divergent hydrological sensitivity to forest gain across regions which is largely controlled by the local climate and landscape conditions[25]. Previous evidence suggested that ET is generally more sensitive to forest cover change in drier regions, highlighting the importance of accessing deeper soil moisture for ecosystems in water-

limited environments[60]. Consistent with the global mean level (0.051 ± 0.031 mm yr⁻², Supplementary Fig. 9b), constrained $\delta ET^{AFF}$ is positive over most (58.8%) part of the afforested land, suggesting the widespread positive effect of afforestation on ET as reported in previous studies[8,61]. Regions with the most notable afforestation-driven increase in ET manifest in Sub-Saharan Africa and South Asia (Fig. 5a and Supplementary Data 4). On the other hand, the negative value of

**Fig. 5 | Constrained projection of future afforestation impact on terrestrial water cycle over IPCC AR6 reference regions. a** Constrained trend in annual evapotranspiration during the period 2015–2099 ($\delta ET^{AFF}$), as derived under the future afforestation scenario (Methods). Within each hexagon, the filled color in the upper and lower parts represent the unconstrained estimate of $\delta ET^{AFF}$ from the original CMIP6 model ensemble and the observationally constrained estimate, respectively. The size of the yellow dot indicates the relative reduction in standard deviation ($RR_\sigma$) after applying the hierarchical emergent constraint approach. The hexagons with the diagonal lines indicate that the emergent relationship is statistically significant at the 90% confidence level. In fact, for these hexagons, their emergent relationship's significance even passes the 95% confidence level.
**b, c** Same as (**a**), but for the trends in annual precipitation ($\delta P^{AFF}$) and terrestrial water availability ($\delta WA^{AFF}$). The hexagons with the diagonal lines indicate that both the emergent relationship and the inter-model relationship between $\delta ET^{AFF}$ and $\delta P^{AFF}$ are statistically significant at the 90% confidence level. Specific values for making these plots are listed in Supplementary Data 4 and 5. Moreover, detailed information of these reference regions is provided in Supplementary Fig. 8.

$\delta ET^{AFF}$ still exists after constraint in some regions such as Northeastern South America (NES) and SEA. This may occur because the shading by increased foliage cover decreases the amount of radiation reaching the ground ultimately leading to a reduction in soil evaporation which offsets the increase in Tr due to afforestation[62].

For twenty-two regions (covering 51.1% of global vegetated land) where emergent relationship and correlation between $\delta ET^{AFF}$ and future afforestation impact on precipitation ($\delta P^{AFF}$) across ESMs are both statistically significant, local $\delta P^{AFF}$ is of comparable magnitude to $\delta ET^{AFF}$ after constraint, particularly in tropics and subtropics (Fig. 5a, b and Supplementary Data 4 and 5). For example, $\delta P^{AFF}$ in North Eastern Africa (NEAF) increases from $0.008 \pm 0.659\,\text{mm yr}^{-2}$ to $0.352 \pm 0.470\,\text{mm yr}^{-2}$, along with the increase in $\delta ET^{AFF}$ from $0.091 \pm 0.354\,\text{mm yr}^{-2}$ to $0.312 \pm 0.253\,\text{mm yr}^{-2}$ after applying the emergent constraint (Supplementary Data 4). The ultimate effect of future afforestation on WA (i.e., $\delta WA^{AFF}$) in the constrained projection is derived as the difference between constrained $\delta P^{AFF}$ and $\delta ET^{AFF}$ (Eq. (11), details in Methods). We estimate that sixteen out of twenty-two reference regions which cover 32.9% of global vegetated land, exhibit an increase in $\delta WA^{AFF}$ after constraint, corresponding to a weakened negative effect of afforestation on WA and even a transition to positive effect in the constrained projection (Fig. 5c and Supplementary Data 4). Such weakening of afforestation-induced negative effect on WA after constraint is particularly evident in Central and South America, Sub-Saharan Africa and South Asia, where constrained $\delta P^{AFF}$ is substantially higher than unconstrained one (Fig. 5b, c). It can be seen that although agreeing on the negative impact of afforestation on WA, the constrained results show a less pronounced afforestation-driven reduction in WA compared to the original projections both at the global scale (unconstrained $\delta WA^{AFF}$: -0.035 mm yr$^{-2}$; constrained $\delta WA^{AFF}$: -0.029 mm yr$^{-2}$) and over broad regions (Fig. 5c and Supplementary Fig. 9). This suggests the general overestimation of original ESM simulations on the negative impact of afforestation on WA. Furthermore, the existence of regions with higher magnitude of positive $\delta P^{AFF}$ than that of $\delta ET^{AFF}$ (i.e., constrained $\delta WA^{AFF} > 0$) suggests that the increase in ET associated with afforestation could be outweighed by more pronounced moisture recycling in these regions. Such patterns emerge in regions such as NEAF, Western Africa (WAF), South Asia (SAS), and Eastern Australia (EAU) (Fig. 5), and appear broadly consistent with observational evidence documenting the buffering effect of forest-climate feedback on local water depletion[18,63].

Altogether these results demonstrate that original ESM simulations overestimate the water security induced by afforestation scenario, particularly in tropics and subtropics (Fig. 5c), and support the value of bias-corrected simulations for a more robust assessment of land-based climate mitigation plans both at the global and regional levels.

## Discussion

Our study provides solid evidence that the model bias in representing the ratio between sensible heat and latent heat through Tr (represented by $\ln(B_{ts})$) results in an inaccurate assessment of global land cover change effect on terrestrial ET and WA in CMIP6 simulations (Figs. 2, 3 and 5). ESMs show considerable differences not only in terms of the magnitude of the LULCC effect on ET but also in the direction of this effect, reflecting a large inter-model spread in $\ln(B_{ts})$ (Figs. 1b, e, f and 2a). LULCC that occurred over the past three decades is prominently characterized by a widespread conversion from forest to cropland patterns (Fig. 1a). Given the generally denser canopies, deeper roots and longer growing season for forests compared to croplands[8], it is counter-intuitive for the original CMIP6 ensemble mean to indicate an increase in global terrestrial ET in response to past widespread conversion from forests to croplands (Fig. 1b). Such evident misrepresentation of the signal of LULCC on the hydrological cycle in state-of-the-art ESMs is particularly critical since these modelling tools are used to frame climate strategies and define future scenarios that are widely used in climate-related policy making[31,64,65]. Our results reveal that the unrealistic positive effect of LULCC on ET obtained from historical simulations arises from the positive bias of $B_{ts}$ in CMIP6 ESMs, which is fundamentally attributable to the strong negative Tr/ET bias (Fig. 2 and Supplementary Fig. 5). Such model limitations have been broadly reported in previous studies[66,67]. We estimate a CMIP6 ensemble mean of global Tr/ET of $0.409 \pm 0.103$, a value which is considerably lower than all the observation-based estimates and is 36.0% lower than their ensemble mean ($0.639 \pm 0.083$) (Supplementary Fig. 5c). Lian et al.[67] found similar results by using field measurements to constrain ESM-based global estimates of Tr/ET and argued that such bias can be attributed to an inaccurate representation of some vegetation-related processes (e.g., interception loss and root water uptake). Here, by adjusting the modeled $\ln(B_{ts})$ to the corresponding observed values within a hierarchical emergent constraint framework, the originally unrealistic positive $\delta ET^{LULCC}$ derived from multi-model ensemble mean has been corrected to a negative effect and its uncertainty reduced by 48.0% (Fig. 2). The negative value of $\delta ET^{LULCC}$ obtained after constraint (-0.031 ± 0.081 mm yr$^{-2}$) agrees with findings reported in previous observation-based studies[4,51,52]. Such global negative signal of $\delta ET^{LULCC}$ during the period 1982-2014 results from the widespread decrease in ET across Central and South America, Western Africa, South East Asia, and Australia, in response to the pronounced expansion of croplands at the cost of forest loss (Fig. 3 and Supplementary Data 3). It is worth noting that, despite the negative effect of LULCC on ET at the global scale (Fig. 2b), an overall increasing trend in ET can still be observed during the past decades, proving that the positive effects of rising atmospheric CO$_2$ concentration and climate change and their associated vegetation greening, have effectively counteracted this negative LULCC effect[8,68,69].

We find that the global mean magnitude of historical LULCC effect on ET (i.e., $\delta ET^{LULCC}$) explains -13% of the overall trend in ET during the period 1982-2014, as estimated based on the CMIP6 model ensemble (Fig. 1b and Supplementary Fig. 1). In comparison, the direct effect of changes in climate factors (e.g., P, temperature and solar radiation) has been estimated of comparable contribution (22.5%) to global ET rise over a similar period, which could be further amplified by the indirect effect from climate-induced vegetation greening[69]. We point out, however, that compensatory effects occurring across the globe may potentially lead to underestimating the role of LULCC in influencing terrestrial water fluxes at the regional scale (Fig. 3). To better inspect the LULCC effect at regional scale, we extend our hierarchical

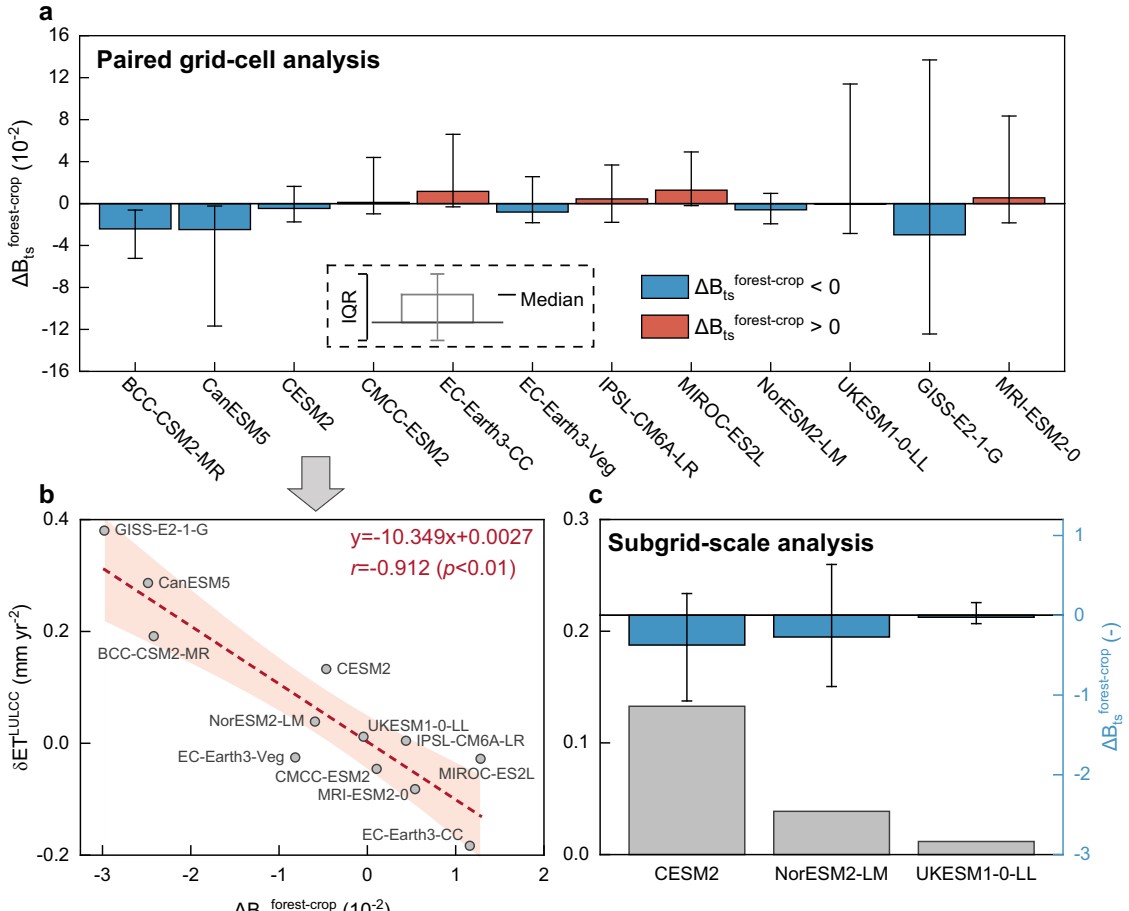

**Fig. 6 | Potential mechanism responsible for inter-model divergence in the effect of land use and land cover change on terrestrial evapotranspiration.** **a** Global median difference in transpiration-specific Bowen ratio between croplands and forests ($\Delta B_{ts}^{forest\text{-}crop}=B_{ts}^{crop}-B_{ts}^{forest}$) during the period 1982-2014, as derived from twelve ESMs by the paired grid-cell method (Methods). Positive and negative values of $\Delta B_{ts}^{forest\text{-}crop}$ are shown in different colors (red and blue, respectively). Error bars represent the interquartile ranges (IQR) of $\Delta B_{ts}^{forest\text{-}crop}$ estimates. **b** Inter-model relationship between the modelled effect of land use and land cover change (generally characterized as a conversion from forests to croplands) on annual evapotranspiration ($\delta ET^{LULCC}$) and modelled median $\Delta B_{ts}^{forest\text{-}crop}$ during the period 1982-2014. Each dot denotes a CMIP6 ESM result, and the red dotted line indicates the best-fit regression line across ESMs, with correlation coefficient ($r$) provided on subplot. Shaded areas denote the 95% confidence range of the linear regression. **c** Global median $\Delta B_{ts}^{forest\text{-}crop}$ during the period 1982-2014, on the basis of subgrid-scale outputs from three ESMs (CESM2, NorESM2-LM, and UKESM1-0-LL) (Methods). Error bars represent the interquartile ranges (IQR) of $\Delta B_{ts}^{forest\text{-}crop}$ estimates. Grey histograms show the $\delta ET^{LULCC}$ corresponding to these three ESMs.

emergent constraint framework to IPCC AR6 reference regions. We find that LULCC dominates historical ET change (relative contribution higher than 50%) in 26.0% of global vegetated land, according to the original CMIP6 model simulations (Supplementary Data 3). Moreover, the absolute magnitude of constrained $\delta ET^{LULCC}$ is more than 2 times (5 times) the global mean level (-0.031 ± 0.081 mm yr$^{-2}$) in 78.4% (58.1%) of the global vegetated land (Figs. 2b and 3 and Supplementary Data 3). These results reinforce the importance of accounting for LULCC to properly understand the driving mechanism behind the change in terrestrial water cycle, particularly at the regional scale.

To disentangle the possible mechanism responsible for the large spread in estimation of $\delta ET^{LULCC}$ across ESMs and its tight relationship with $B_{ts}$, we explore the difference in surface energy partitioning across plant functional types (PFTs). We focus primarily on the difference in $B_{ts}$ between croplands and forests ($\Delta B_{ts}^{forest\text{-}crop}=B_{ts}^{crop}-B_{ts}^{forest}$), as prominent LULCC transition during the historical period (1982-2014, Fig. 1a). $\Delta B_{ts}^{forest\text{-}crop}$ is disentangled from other confounding effects (e.g., background climate and elevation) through a paired grid-cell method[70] based on the comparison of $\Delta B_{ts}$ values between grid-cells experiencing significant forest-cropland conversion and adjacent ones with stable land cover (details in Methods). Results show that most ESMs assume a similar or even lower ratio between sensible heat and latent heat partitioned into Tr process for croplands than the one computed for forests. This is clearly reflected by the widespread existence of negligible or negative $\Delta B_{ts}^{forest\text{-}crop}$ across ESMs (Fig. 6a). These simulation patterns are in clear contrast with observations, which instead suggest that deforestation tends to increase B and reduce Tr/ET, ultimately leading to an intensification of $B_{ts}$[8,71,72]. Such divergence indicates that ESMs may not properly represent the differences in biophysical processes across PFTs and cause an overall misestimation of ET response to PFT conversions (e.g., forest-cropland conversion during the historical period) (Fig. 1b). The strong inter-model correlation between $\Delta B_{ts}^{forest\text{-}crop}$ and $\delta ET^{LULCC}$ ($r = -0.912$, $p < 0.01$) reveals that the occurrence of an unrealistic increase in ET during the conversion from forests to croplands in model simulations may originate from the substantial underestimation of $B_{ts}$ for croplands and/or the overestimation of $B_{ts}$ for forests in ESMs (Fig. 6b). The model underestimation of $B_{ts}$ (i.e., excessively low ratio between sensible heat and latent heat through Tr) for croplands can potentially be associated with the uncertain parameterization and partial representation of some key processes in agricultural lands including stomatal conductance, productivity and leaf area index (LAI) as

documented in previous data-model comparisons[33,34]. Furthermore, current ESMs may not properly represent the extent and magnitude of cropland irrigation and further underestimate the P limitation on rain-fed croplands, further contributing to an underestimation of $B_{ts}$ for this vegetation type[73]. The possible overestimation of $B_{ts}$ for forests may instead arise from the large positive bias in forest sensible heat existing in process-based models[74].

To further investigate this issue, we perform additional experiments by exploiting outputs from three ESMs (i.e., CESM2, NorESM2-LM, and UKESM1-0-LL) generated at the sub-grid level (details in Methods). Such sub-grid simulations provide a direct comparison of the surface energy partitioning between croplands and forests at grid-cell level by ruling out potential contamination effects of background environmental conditions. Results based on these additional experiments further confirm a generally lower $B_{ts}$ for croplands than that one estimated for forests (i.e., $\Delta B_{ts}^{forest-crop} < 0$) occurring in combination with a positive $\delta ET^{LULCC}$ during the historical period (Fig. 6c and Supplementary Fig. 10). Moreover, the evident negative relationship between $\delta ET^{LULCC}$ and $\Delta B_{ts}^{forest-crop}$ across these three ESMs (the more negative $\Delta B_{ts}^{forest-crop}$, the more positive $\delta ET^{LULCC}$), further demonstrates the considerable bias originating from the misrepresentation of $B_{ts}$ in croplands and forests (Fig. 6c).

Overall, these results suggest that the large model spread in $\delta ET^{LULCC}$ (Fig. 1b) originates from the ESM misrepresentation of the difference in surface energy partitioning (particularly latent heat partitioned into Tr) between forests and croplands. Furthermore, results described above help explaining the underlying mechanism of correcting modelled $\delta ET^{LULCC}$ by introducing observed energy partitioning, as shown in Figs. 2 and 3 and Supplementary Figs. 6 and 7a, b.

The emergent constraint framework initially developed to track the changes in ET induced by LULCC over the historical period is then applied to future afforestation scenarios by exploiting the tight relationship between $\delta ET^{AFF}$ and historical $\ln(B_{ts})$ (Supplementary Fig. 9a and Table 6). Well-constrained future $\delta ET^{AFF}$ shows a stronger increase than the original projection of CMIP6 model ensemble (Supplementary Fig. 9b and Table 6). Nevertheless, such stronger positive $\delta ET^{AFF}$ could be partly offset by its associated positive feedback onto P, ultimately leading to a lower decline in terrestrial WA at the global extent compared to that one estimated in previous assessments based on original model simulations[75] (Supplementary Fig. 9). Our results derived at the regional scale further suggest that the compensation from afforestation-induced P increase is widespread and is particularly evident in Sub-Saharan Africa, Tibetan Plateau, and South Asia, as reflected by the fact that $\delta P^{AFF}$ in these regions is 5.9 - 32.5 times higher than the global mean level ($0.022 \pm 0.038$ mm yr$^{-2}$) (Fig. 5b and Supplementary Fig. 9d). These results are corroborated by previous works which have highlighted the limited impact of increased forest area on soil moisture due to the concurrent resulting P increase in wet regions[32], and in regions with a marked orography, such as Loess Plateau, Andes and Tibetan Plateau[23,76]. This positive feedback onto P could be more pronounced in moist forests as reported in related literature, since these forests are "closed" atmospheric systems where 80% of the P originates from upwind ET[10,76]. In addition, for regions like Tibetan Plateau, moisture tends to be sustained because the prevailing winds blow towards the mountains, and the orographic lifting of moisture leads to repeated rainfall[76]. Compared with the original simulations obtained from the CMIP6 model ensemble (i.e., weak increase in ET, decrease in P, and severe decrease in WA), the constrained results show different patterns of the terrestrial water cycle in response to future afforestation, and indicate that unconstrained ESM outputs may overestimate the potential extent of afforestation-induced water scarcity globally over the 21st century (Fig. 5 and Supplementary Fig. 9). In this respect, although some efforts have been devoted recently to relevant topics based on original simulations of ESMs[9,77], our bias-corrected assessments could better capture the

afforestation-related signal and identify key hotspot areas where afforestation is expected to play a critical control on local WA, thereby gaining a more robust projection of the potential impact of afforestation on water supply for society.

All together our results provide a coherent picture of how the misrepresentation of $B_{ts}$ and its variations across PFTs are the main sources of the substantial bias in ESMs in representing water limitation, water use and energy partitioning in terrestrial ecosystems (Figs. 2c and 6 and Supplementary Figs. 5 and 10). In view of the strong coupling between terrestrial water and carbon cycles[35], such mentioned biases not only lead to an incorrect assessment of the impacts of historical LULCC and future afforestation on water cycle (Figs. 2b, 3 and 5 and Supplementary Fig. 9b, d), but may also affect the sensitivity and overall response of primary productivity to water scarcity[78,79], ultimately leading to a misrepresentation of the future growth potential of terrestrial vegetation and terrestrial carbon sink. Such model bias represents a non-negligible issue both for the broad scientific communities and social stakeholders, given that these ESMs (as well as their components like LSMs) are widely used to build scenarios that are influential in the climate policy processes via assessments such as the IPCC AR[80] and the Global Carbon Project (GCP)[81]. Therefore, the model limitations in projecting hydrological consequences shown above (Figs. 2b, 3 and 5 and Supplementary Fig. 9b, d) suggest that using original ESM outputs would result in substantial uncertainties in the effectiveness and risk of land-based policies. Our approach based on emergent constraints shows how we can effectively correct this bias with robust, observation-driven metrics, and derive more accurate assessments without changing the Earth system modelling framework. This opens an alternative efficient data-driven path to improve the prediction capacity of ESMs, without altering the model structure and parameterization. We believe that the proposed approach – adequately modified – could be used analogously for global and regional constraints on model projections in other aspects of terrestrial carbon-water coupling, including terrestrial carbon sequestration, vegetation water uptake, and ecosystem water-use efficiency.

In conclusion, the close inter-model relationship between $\ln(B_{ts})$ and $\delta ET^{LULCC}$ (or $\delta ET^{AFF}$) revealed here (Fig. 2a and Supplementary Fig. 9a), highlights that addressing the uncertainty of Earth system models driven by the surface energy balance should be a key priority for reducing biases in CMIP6 simulations. Future improvements of ESM based on these experimental constraints will have great implications not only for a better understanding of complex land-atmosphere coupling[8], but also for achieving a more accurate assessment on the impact of anthropogenically driven land use and land cover change and particularly extensive afforestation on WA. In addition, a more accurate representation of the ratio between sensible heat and latent heat through Tr in ESMs will also improve their capacity to represent the biophysical climate impact of LULCC as driven by non-radiative processes[33]. Differently from traditional approaches based on time-consuming and labor-intensive model parameterization refinements and functional/structural advances[27], our study provides a simple but effective approach to capture valuable signals of land-atmospheric interactions from existing model outputs within the emergent constraint framework, thereby exploiting an observation-based constraint on both historical and future model simulations. It is well known that large-scale afforestation plays a key role in reaching carbon neutrality and meeting climate mitigation targets[82,83], and meanwhile, water shortages have rising impacts on the vegetation carbon uptake and on the terrestrial carbon budget[12,13]. In this respect, our constrained results may help the reduction of uncertainties in the prospect of implementing land-based climate mitigation policies and further support the development of more practical ones, by providing more accurate projections of the hydrological consequences of future afforestation.

## Methods

### CMIP6 simulations

To explore the effect of historical land use and land cover change (LULCC) on terrestrial water cycle, we use outputs from an ensemble of eleven Earth system models (ESMs) participating in the Land Use Model Intercomparison Project (LUMIP)[40] within the framework of the Coupled Model Intercomparison Project Phase 6 (CMIP6) (https://aims2.llnl.gov/search/cmip6/): BCC-CSM2-MR, CanESM5, CESM2, CMCC-ESM2, EC-Earth3-CC, EC-Earth3-Veg, GFDL-ESM4, IPSL-CM6A-LR, MIROC-ES2L, NorESM2-LM, and UKESM1-0-LL (Supplementary Table 1). These models are selected because they provide all the diagnostic variables required for this study under different factorial experiments. The first available realization (r1) for each model is used here to ensure equal weight to each model in the multi-model ensemble mean. For each model, one experiment with all forcings time-varying and one experiment with land use and land cover kept constant ("historical" and "hist-noLu" in the CMIP6 terminology) during the historical period (1982-2014) are analyzed within a factorial simulation framework (Supplementary Table 2). Except land use and land cover, all other forcings such as greenhouse gases, aerosols and nitrogen deposition are identical for these two sets of experiments, which are run in fully coupled mode.

To project the hydrological impact of future afforestation (AFF), one SSP3-7.0 scenario experiment with corresponding land cover and land use and one SSP3-7.0 scenario experiment with land cover and land use from SSP1-2.6 ("ssp370" and "ssp370-ssp126Lu" in the CMIP6 terminology) for the period 2015-2099 are analyzed in this study. According to the LUMIP protocol[40], SSP1-2.6 represents the afforestation scenario while SSP3-7.0 represents the deforestation scenario, as clearly reflected in Fig. 4. SSP1-2.6 represents the scenario where the combination of high economic growth and technological improvements and climate policies results in substantial reduction in total agricultural land and increase in forest area. Important climate policies from the land use perspective involved in SSP1-2.6 include the increased bioenergy use, and programs of avoided deforestation, reforestation and afforestation[59]. SSP3-7.0 represents the scenario where land use change is hardly regulated and strong expansion of cropland and pasture to cope with food shortage results in large-scale deforestation[41]. Therefore, the difference of outputs between "ssp370-ssp126Lu" experiment and "ssp370" experiment can be used to disentangle the impact of future afforestation scenario (details in section "Assessing LULCC effects from model outputs"). We stress that despite the overall afforestation at the global extent, forest cover in SSP1-2.6 is not always higher than that in SSP3-7.0 for all regions, as shown in Fig. 4b and Supplementary Data 4. Previous studies show possible scattered deforested regions (e.g., Eastern Europe and Southern China) under this future afforestation scenario[9]. Such cases could be related to the anticipated socioeconomic trends and region-specific climate policy considered in future scenarios[41,59,84]. Our study period ends in 2099 instead of 2100, mainly because of the limitation from time coverage of NorESM2-LM simulations in "ssp370-ssp126Lu" experiment. It is noteworthy that GFDL-ESM4 is excluded from historical analyses, since this model shows an unrealistic decreasing trend in ET over 1982-2014 in the "historical" experiment in contrast with simulations from all other ESMs (Supplementary Fig. 11a) as well as previous reports[8,85]. Nevertheless, given the good agreement in future trend in ET between GFDL-ESM4 and the other ESMs and to maximize the sample size when establishing the emergent relationship related to hydrological effect of future afforestation, GFDL-ESM4 is included in subsequent analyses (Supplementary Fig. 11b). Additional results derived by leaving GFDL-ESM4 out of our analyses further demonstrate that including/excluding GFDL-ESM4 would have limited impact on our findings (Supplementary Figs. 9a, b and 12). EC-Earth3-CC and EC-Earth3-Veg are employed solely for historical analyses since their future projections under SSP3-7.0 are not available. Details of

experiments have been provided for each model in Supplementary Table 3. In addition to the aforementioned experiments run in fully-coupled mode, we use outputs from two sets of experiments run in ocean-uncoupled mode conducted within the Aerosol Chemistry Model Intercomparison Project (AerChemMIP)[86] (Supplementary Table 2). For these simulations, sea surface temperature (SST) and sea ice concentration (SIC) are prescribed. Differently from ESMs in LUMIP that are constructed and operated with the full coupling between modules, ocean module and its connection with other modules have been switched off within ESMs in AerChemMIP. In view of this, a pair of experiments in AerChemMIP, i.e., "histSST" and "histSST-noLu", is considered equivalent to the "historical" and "hist-noLu" in LUMIP[86]. Two ESMs, namely GISS-E2-1-G and MRI-ESM2-0, participating in these two sets of experiments provide all variables required for this study, and therefore are employed for our analyses (Supplementary Tables 1 and 3).

Land use and land cover forcing in both historical and future scenario periods for CMIP6 (i.e., in all experiments mentioned above) is prescribed by Land-Use Harmonization 2 (LUH2) dataset[41]. LUH2 is generated based on the History database of the Global Environment (HYDE 3.2) and the Food and Agriculture Organization of the United Nations (FAO) wood harvest data, and constrained by Landsat remote sensing observations. As one of the key inputs of LUH2 dataset, the HYDE 3.2 involves the dynamics of cropland and grazing land, the distinction between rain-fed and irrigated cropland, the distinction between pasture and rangeland according to the intensity of use, and the delineation of protected areas[87]. Furthermore, the inclusion of FAO wood harvest data implies that LUH2 can also reflect the spatio-temporal patterns of wood harvesting.

A set of variables simulated by the ESMs and representative of the mixture of multiple plant functional types (PFTs) within each grid cell are selected for our analyses. These variables include: monthly-scale evapotranspiration (ET), transpiration (Tr), latent heat (LE), sensible heat (H), leaf area index (LAI), precipitation (P), and mean air temperature (T). Full time series of cover fractions of trees, crops and grasses during both historical and future scenario periods are available for five ESMs (CanESM5, CESM2, CMCC-ESM2, IPSL-CM6A-LR, and UKESM1-0-LL), which are also used here to explore the land cover condition and its dynamics globally and regionally (Figs. 1a and 4 and Supplementary Data 3 and 4). We point out that original LUH2 data cluster the global land into five main categories including cropland, grazing land, primary land, secondary land, and urban land[41]. ESMs are based on different assumptions when translating LUH2 raw information into datasets of tree, crop, and grass covers, thereby leading to the slight model spread in estimates of areas of global forests, croplands, and grasslands (Figs. 1a and 4a and Supplementary Data 3 and 4). In addition, the simplification and aggregation of numerous vegetation types within these ESMs may result in some inconsistency across models and between individual model and observed patterns in terms of cover fraction of PFTs. CMIP6 outputs are first resampled to a common 0.5°×0.5° global grid-cell using the bilinear method of interpolation, and then aggregated at the annual temporal resolution (i.e., accumulated value of monthly ET, Tr, and P in a given year; average value of monthly LE, H, LAI, T, and fractional covers of vegetation types in a given year). Following Forzieri et al.[68], grid-cells with multi-year (1982-2014) average LAI < 0.15 $m^2$ $m^{-2}$ – thus characterized by low vegetation density – are excluded from the analyses. To keep consistent the spatial domain of simulations, the multi-year averaged LAI derived from the multi-model ensemble mean is used as common mask for all ESMs.

Furthermore, output from three additional ESMs (i.e., CESM2, NorESM2-LM, and UKESM1-0-LL) provided at subgrid-scale are exploited for the historical period. These simulations are used to quantify the difference in surface energy partitioning between PFTs (details in section "Identifying the difference in energy partitioning

between PFTs"). Subgrid data provide information of LE, H and LAI distributed across four main land use categories within each grid-cell[40]: (1) primary and secondary lands (i.e., forests, grasslands, and bare ground), (2) croplands, (3) pasturelands, and (4) urban lands. These subgrid-scale categories are generally consistent with the original ones in LUH2 dataset, and primary and secondary lands are not further divided into PFTs like forests and grasslands, as performed at the scale of entire grid-cell.

## Observation-based products

The observational ET and Tr are derived from four global gridded datasets, including the LAI-based upscaling[44], the Global Land Evaporation and Amsterdam Model (GLEAM v3.8a)[45,88] (https://www.gleam.eu/), the Gerrits' model[46] (https://data.4tu.nl/articles/_/12718811/1), and the Penman-Monteith-Leuning Model (PML-V2)[47] (https://doi.org/10.6084/m9.figshare.14185739.v6). Further details of these products are provided in Supplementary Table 7. The above-mentioned datasets are selected since they cover the entire reference period 1982-2014 and have been well validated. In addition, the terms of surface energy partitioning, including H and LE are required in this study. Observational LE is derived by combining latent heat of vaporization (2.45 MJ kg⁻¹) and ET estimates derived from the four different observation-based datasets mentioned above. H is obtained from the closure of the energy balance by subtracting LE from the surface net radiation (Rn), under the assumption of no changes in soil heat at the mean annual level[34,68]. Rn is retrieved for the period 1982-2014 from two different reanalysis datasets, namely 20CR v3[48] (https://psl.noaa.gov/data/20thC_Rean/) and ERA5-Land[49] (https://cds.climate.copernicus.eu/). All observational products have been resampled to the common 0.5°×0.5° spatial resolution and used to derive the long-term annual average of variables mentioned above (i.e., Tr, ET, LE, Rn, and H). Based on such an approach, we derive an eight-member ensemble of H estimates, and an eight-member ensemble of Bowen ratio (H/LE) estimates (Supplementary Fig. 5b), each one derived from a different combination of observation-based ET product and climate reanalysis dataset. Sensible fluxes derived from FLUXCOM Model Tree Ensemble[89] (https://www.bgc-jena.mpg.de/geodb/projects/Home.php) are used to verify the consistency of H estimates derived from the closure of the energy balance (Supplementary Table 8). To cover the whole period, FLUXCOM dataset used here is the ensemble mean that encompasses members of different machine learning methods and energy balance corrections within the "RS + METEO" setup.

Eddy covariance measurements of water and energy fluxes derived from 132 FLUXNET sites are also used in this study to further validate the robustness of our results derived from the observation-based gridded products mentioned above (Supplementary Fig. 6a and Data 1 and 2). Site-level datasets of H and LE (ET) are collected directly from the FLUXNET2015 database[55] (https://fluxnet.org/data/fluxnet2015-dataset/). Tr dataset[54] for these 132 sites is derived from the combination of FLUXNET2015 eddy covariance data with three different Tr estimation methods (https://zenodo.org/records/3978408). These methods include the underlying water use efficiency (uWUE) method[90], the Pérez-Priego method[91], and the Transpiration Estimation Algorithm (TEA) method[92]. Here we use the mean of Tr estimates obtained from the aforementioned methods as benchmark, and the standard deviation of these three sets of estimates as a measure of uncertainty in observed Tr (Supplementary Data 2).

## Spatial domain

In this study, we conduct analyses both at the global level and over reference regions used in the Intergovernmental Panel on Climate Change Sixth Assessment Report (IPCC AR6). Analyses focusing on the global level provide a clear overview of land use change effect on terrestrial water cycle during both historical and future scenario periods (Fig. 2 and Supplementary Fig. 9). This helps to resolve the long-

standing argument across model-based assessments about the direction and magnitude of changes in ET and terrestrial water availability (WA) in response to LULCC and afforestation actions[27] and elucidate the underlying mechanism behind such argument (Fig. 6). Given the fact that global mean results may partially mask emerging patterns due to possible spatial compensatory effects (e.g., opposite LULCC effects amongst different areas of the Planet, Fig. 1c), we perform additional experiments to investigate these patterns at the regional scale and better capture the relevance of LULCC in long-term variations in regional water cycle (Figs. 3 and 5). Original IPCC AR6 reference regions include forty-six land regions and fifteen ocean regions with distinct regional climatic features[39]. Here, all ocean regions and two land regions located in Antarctica are excluded from our analyses, since we focus exclusively on the vegetated land (Supplementary Fig. 8). To this end, all estimation approaches described below have been applied both at the global scale and over forty-four IPCC AR6 reference regions.

## Assessing LULCC effects from model outputs

Following similar approaches reported in literature[93–95], the effect of LULCC on annual terrestrial ET (expressed as δET^LULCC) during the period 1982-2014 is derived from factorial simulations of multiple CMIP6 experiments by calculating the difference between the trend in annual ET generated in the "historical" experiment and that one derived in the "hist-noLu" experiment:

$$\Delta ET^{LULCC} = \Delta ET_{hist} - \Delta ET_{hist-noLu} \quad (1)$$

where $ET_{hist}$ and $ET_{hist-noLu}$ are the trends in annual ET in the experiment with all forcings time-varying (i.e., "historical") and in the experiment with land use and land use kept temporally constant (i.e., "hist-noLu"), respectively. The statistical significance of $ET_{hist}$ and $ET_{hist-noLu}$ is evaluated using the nonparametric Mann-Kendall test (Supplementary Fig. 11), and the significance of the resulting δET^LULCC is assessed through $t$-test (Fig. 1b). The term δET^LULCC excludes the effects of all other forcing agents (e.g., greenhouse gases and aerosols) and their associated changes in climate on ET, as these components have been removed from the factorial simulations (Eq. (1)). For both global and regional scale δET^LULCC estimates, the ET terms reported in Eq. (1) are obtained by spatial average weighting each grid-cell value based on its land area (Figs. 1b and 2a). Land area for each grid-cell is derived by the product of the grid-cell area and the percentage of the grid-cell occupied by land (i.e., variable "sftlf" in CMIP6 archive). The same methodology is applied consistently for all global-scale and regional-scale aggregated metrics described in the following sections.

For GISS-E2-1-G and MRI-ESM2-0, δET^LULCC can be derived from the pair simulations run with ocean-uncoupled mode, by replacing $ET_{hist}$ and $ET_{hist-noLu}$ in Eq. (1) with trends in annual ET in the "histSST" experiment and in the "histSST-noLu" experiment, respectively:

$$\Delta ET^{LULCC} = \Delta ET_{histSST} - \Delta ET_{histSST-noLu} \quad (2)$$

Our results demonstrate that including/excluding these two ESMs and their associated estimates does not affect our findings obtained from the full model ensemble (Fig. 1d, f). However, the inclusion of these two additional ESMs help to expand the sample size and to establish a more robust emergent relationship. For this reason, they are adopted in this study.

Moreover, the impact of future afforestation on terrestrial ET (expressed as δET^AFF) is quantified through factorial simulations run under the future scenario period (2015-2099) as follows:

$$\Delta ET^{AFF} = \Delta ET_{ssp370-ssp126Lu} - \Delta ET_{ssp370} \quad (3)$$

where $ET_{ssp370-ssp126Lu}$ and $ET_{ssp370}$ are the trends in annual ET in the "ssp370-ssp126Lu" experiment and the "ssp370" experiment, respectively. The only difference between these two sets of experiments is the land use and land cover forcing, which ensures that Eq. (3) could effectively factor out the effects of all other forcings (e.g., greenhouse gases like $CO_2$, and aerosols) and the secondary radiative forcing on terrestrial ET. As mentioned in the previous section, SSP1-2.6 is an afforestation scenario while SSP3-7.0 is a deforestation scenario. The large difference in forest cover and its dynamics between these two scenarios (Fig. 4) allows to capture the afforestation-induced signal of ET variation.

To improve the accessibility, here we use "LULCC" and "AFF" to represent the land cover conversion during the historical period (1982-2014) and the future afforestation scenario period (2015-2099), respectively.

## Constraining LULCC effect on evapotranspiration based on observations

The emergent constraint approach has gained prominence in recent years to correct present-day simulations and constrain future projections of unobserved quantities of interest in ESMs or their single components (i.e., land surface models and climate models)[50,96,97]. This approach helps to reduce uncertainty in ESM simulations through the combination of an ensemble of original model simulations with contemporary observations. The core concept relies on the fact that, despite substantial differences across ESMs, relationships between two elements ($x$ and $y$, respectively) are implicit within ESM solutions of complex equations and associated parameterizations[50]. Estimates of variable $x$ and variable $y$ may both exhibit the large spread across models, but there may exist strong and robust relationship between these two variables, i.e., $y = f(x) + \varepsilon$, where $\varepsilon$ is a relatively small departure from $f$. If $x$ is a quantity for which observation-based data are readily available, then the relationship $f$ may place a useful constraint on $y$, provided the observation uncertainty in $x$ is small compared to the range of simulated values. Given the fact that such functional relationship cannot be diagnosed from a single ESM but rather becomes apparent through the analysis of a suitably large ESM ensemble, the approach mentioned above is therefore termed 'emergent'[98]. In this respect, when applying the emergent constraint framework, bias correction and uncertainty reduction are not performed for one specific ESM as evaluated in studies focusing on model development[31,99,100]. By contrast, simulation from each ESM is treated as one sample, and all samples are clustered together to identify an overall functional relationship across ESMs (i.e., $f$).

The hierarchical emergent constraint proposed by Bowman et al.[37] is applied in this study. Compared with the classic emergent constraint, the hierarchical emergent framework accounts for both the correlation between two variables (i.e., $x$ and $y$) and errors in the observational datasets and it is, therefore, believed to yield more accurate and precise estimates[101]. The emergent constraint framework identified in this study links the historical $\delta ET^{LULCC}$ and future $\delta ET^{AFF}$ to the contemporary mean transpiration-specific Bowen ratio ($B_{ts}$). Compared with the standard Bowen ratio ($B = H/LE$, the ratio of sensible heat to the latent heat) that represents the overall partition of available energy at the surface, $B_{ts}$ proposed here reflects explicitly the ratio between H and LE used for Tr. As Tr is more sensitive to LULCC than evaporation[8], the resulting $B_{ts}$ may ultimately produce a better constraint on LULCC effect compared to B (Supplementary Fig. 3). The analytical formulation of $B_{ts}$ is as follows:

$$B_{ts} = \frac{H}{LE * \left(\frac{Tr}{ET}\right)} \quad (4)$$

where H and LE are given in W m$^{-2}$, Tr and ET are in mm yr$^{-1}$. All variables in Eq. (4) correspond to mean annual values during the period 1982-

2014. In line with B mentioned previously, the combination of four observation-based Tr/ET products and two climate reanalysis datasets produces an eight-member ensemble of observed $B_{ts}$ (Supplementary Figs. 5a, b and 13).

Since the natural logarithmic model outperforms in capturing the underlying relationship between $B_{ts}$ and the LULCC effect compared to the alternative linear and quadratic models (Supplementary Fig. 3a), predictor $x$ is set to the natural logarithm of $B_{ts}$ during the period 1982-2014 (i.e., $\ln(B_{ts})$). The target variable $y$ included in such a framework refers to $\delta ET^{LULCC}$ over 1982-2014 or $\delta ET^{AFF}$ over 2015-2099, for both constraint applications at the global level (Fig. 2 and Supplementary Fig. 9a, b) and at the regional scale (Figs. 3 and 5a). This non-linearity is further validated through random sampling over global grid-cells with one-thousand replications (Supplementary Fig. 4, details in Supplementary Text 2). In addition, sensitivity analyses of the above-mentioned modelling choices related to the predictor variables have been performed and are elaborated at the end of this section.

Following related literature[37,101–103], the mean value of constrained predictand ($y_c$) can be estimated by the following equation:

$$\bar{y_c} = \bar{y} + k_c(\bar{x_0} - \bar{x}) \quad (5)$$

where $\bar{x}$ and $\bar{y}$ are the multi-model ensemble mean of $x$ ($\ln(B_{ts})$) and $y$ ($\delta ET^{LULCC}$ or $\delta ET^{AFF}$). $\bar{x_0}$ is the mean of observed predictor ($x_0$), which is derived from the mean of the eight-member ensemble of observation-based $\ln(B_{ts})$ estimates (Fig. 2c). The slope $k_c$ is calculated as follows:

$$k_c = \frac{k}{1 + SNR^{-1}} = \frac{r(x,y)\sigma(y)\sigma(x)}{\sigma^2(x) + \sigma^2(x_0)} \quad (6)$$

where $k$ is the slope of the linear equation relating the target $y$ with the predictor $x$, and it is estimated by least square method; $\sigma(*)$ and $\sigma^2(*)$ denote the standard deviation and variance of $*$ (i.e., $x$ or $y$), respectively. $r(x,y)$ refers to the correlation coefficient between $x$ and $y$. $k_c$ is regarded as the corrected $k$, by explicitly accounting for the signal-to-noise ratio (SNR) in the observation[104]. SNR is derived as the ratio between the variance across models ($\sigma^2(x)$) and the variance across observational datasets ($\sigma^2(x_0)$):

$$SNR = \frac{\sigma^2(x)}{\sigma^2(x_0)} \quad (7)$$

The constrained standard deviation of predictand (i.e., the uncertainty) is estimated as below[37,101]:

$$\sigma(y_c) = \sigma(y)\sqrt{1 - \frac{r^2(x,y)}{1 + SNR^{-1}}} = \sigma(y)\sqrt{1 - \frac{r^2(x,y)}{1 + \left(\frac{\sigma^2(x_0)}{\sigma^2(x)}\right)}} \quad (8)$$

where $r^2(x,y)$ is the square of the correlation coefficient between $x$ and $y$.

The unconstrained and constrained ranges of historical $\delta ET^{LULCC}$ and future $\delta ET^{AFF}$ are estimated by assuming a Gaussian distribution[37] (Fig. 2b and Supplementary Figs. 6c, 9b and 12b). The relative reduction in standard deviation ($RR_\sigma$) is calculated as follows:

$$RR_\sigma = \left(1 - \frac{\sigma(y_c)}{\sigma(y)}\right) \times 100\% \quad (9)$$

Clearly, the difference between $y$ and $y_c$ is referred to the correction of emergent constraint. From the above equations, such correction in magnitude and uncertainty are proportional to the slope $k_c$, which is further proportional to the correlation strength between $x$ and $y$. In view of this, the correlation strength largely determines the correction performance after applying the emergent constraint.

Therefore, for analysis at the regional scale, regions where correlation between $\delta ET^{LULCC}$ ($\delta ET^{AFF}$) and $\ln(B_{ts})$ is non-significant or small, the correction to $\delta ET^{LULCC}$ ($\delta ET^{AFF}$) by the emergent constraint approach generally has little effect (Figs. 3 and 5a).

We perform an additional set of experiments to test the sensitivity of our constraint results on the log-transformation of $B_{ts}$. To this aim, we propose a new emergent constraint framework that incorporates $B_{ts}$ in place of $\ln(B_{ts})$ (details in Supplementary Text 2). Outcomes of these analyses suggest that emergent constraint results based on $B_{ts}$ are generally consistent with original ones based on $\ln(B_{ts})$ both at the global and regional scales, but exhibit a weaker reduction in uncertainty (reflected by $RR_\sigma$) for some regions due to the relatively weaker correlation of ET response with $B_{ts}$ compared to that with $\ln(B_{ts})$ (Figs. 2, 3 and 5 and Supplementary Figs. 2a–d, 9, 14 and 15). The high agreement between the two sets of estimates further demonstrates the robustness of our emergent constraint results. On the other hand, to maximize the performance of emergent constraint in reducing model uncertainty, we adopt the emergent constraint framework built upon $\ln(B_{ts})$ throughout our study.

Moreover, we perform additional analyses replicated by using the classic emergent constraint[56,57], and compare the associated estimates against analogous estimates based on the hierarchical emergent constraint described above (details in Supplementary Text 3). Results show a high consistency between the two sets of estimates, further demonstrating the robustness of our constrained results (Fig. 2 and Supplementary Figs. 7 and 9a, b).

## Deriving constrained responses of precipitation and water availability

The original model projection of afforestation impact on P ($\delta P^{AFF}$) during the period 2015-2099 is derived by applying Eq. (3) to P (in place of ET). We find a strong correlation between afforestation-induced P change and ET change across ESMs at the global scale ($r = 0.707$, $p < 0.05$) (Supplementary Fig. 9c). Additional experiments performed on the selection of the model ensemble (Supplementary Table 6) and through random sampling over global grid-cells (Supplementary Fig. 2e, f) further confirms the robustness of our finding about the significant inter-model relationship between $\delta P^{AFF}$ and $\delta ET^{AFF}$ (details in Supplementary Text 4). Similar results have also been reported in previous literature[94]. The strong inter-model correlation between $\delta P^{AFF}$ and $\delta ET^{AFF}$ also exists widely over reference regions (Fig. 5b and Supplementary Data 5). Such relationship enables to constrain $\delta P^{AFF}$ by using observationally constrained estimate of $\delta ET^{AFF}$ (i.e., $\bar{y}_c$ in Eq. (5)) both at the global and regional scales. Hence, following methodological approaches documented in previous studies[101,105], the constrained $\delta P^{AFF}$ is derived as follows:

$$constrained\ \Delta P^{AFF} = a \cdot \left(constrained\ \Delta ET^{AFF}\right) + b \qquad (10)$$

where $a$ denotes the regression coefficient between raw estimates of $\delta P^{AFF}$ and $\delta ET^{AFF}$, and $b$ the is intercept. As shown in Supplementary Fig. 9c, for analysis at the global mean level focusing on the period 2015-2099, $a = 1.2358$, and $b = -0.0408$. Here, we use a correlation analysis to estimate the constrained value of $\delta P^{AFF}$ instead of replicating the emergent constraint framework as performed for $\delta ET^{AFF}$, mainly because P changes show a closer relationship with long-term ET changes compared with $\ln(B_{ts})$. This is likely because some key P determinants, such as afforestation's regulation to atmospheric circulation and vertical ascent, are better represented by ET change than $B_{ts}$ dynamics[106].

Terrestrial water availability (WA), i.e., the net flux of water at the land surface, is here defined as P minus ET[9,69,107]. Thus, WA change during a given period can be computed as $\Delta WA = \Delta P - \Delta ET$. Correspondingly, the observationally constrained net effect of future (2015-

2099) afforestation on WA can be expressed as:

$$constrained\ \Delta WA^{AFF} = constrained\ \Delta P^{AFF} - constrained\ \Delta ET^{AFF} \qquad (11)$$

## Identifying the difference in energy partitioning across PFTs

To elucidate the potential mechanisms behind the large spread in the estimates of LULCC effect across ESMs, we perform an additional analysis based on subgrid-scale model outputs, and quantify the difference in ratio of H to LE used for Tr between croplands and forests (i.e., $\Delta B_{ts}^{forest-crop} = B_{ts}^{crop} - B_{ts}^{forest}$) during the period 1982-2014. As mentioned previously, forests are not an individual category in LUMIP subgrid data, but are classified as primary and secondary lands along with grasslands and bare ground (details in section "CMIIP6 simulations"). In this study, primary and secondary lands are considered as forests, and therefore, subgrid-scale outputs for category of primary and secondary lands (i.e., $H^{psl}$, $LE^{psl}$, and $LAI^{psl}$, where psl is the abbreviation of primary and secondary lands) under the "historical" experiment are used to identify the energy partitioning for forests. To minimize the potential effect of this rough PFT classification on our results, only grid-cells dominated by forests (cover fraction of trees $\geq 60\%$, Forzieri et al.[34]) are considered in our subgrid-scale analyses (Supplementary Fig. 16). Cover fraction of trees for each grid-cell is directly provided by ESMs, as reported in section "CMIIP6 simulations" and grid-cells with mean annual LAI $< 0.15$ m$^2$ m$^{-2}$ are excluded from our subgrid-scale analyses as also performed in previous experiments (Fig. 6c and Supplementary Fig. 10). Since Tr is not available for different PFTs in subgrid-scale data, Tr/ET for forests within each grid-cell at the multi-year (1982-2014) mean level is approximated by Beer's Law in combination with the corresponding mean annual value of $LAI^{psl}$[108]:

$$\frac{Tr^{forest}}{ET} = 1 - \exp(-k_{Rn} LAI^{psl}) \qquad (12)$$

where $k_{Rn}$ is the radiation extinction coefficient, and is generally assumed as 0.6[108]. Beer's Law can provide reasonable and accurate ET partitioning into Tr at the long-time scales (i.e., monthly to annual time scales)[109,110], and therefore is applied here for the estimation at the mean annual level. To further validate the performance of Beer's Law in representing mean annual ET partitioning, Tr/ET estimate based on Beer's Law is compared against that one directly derived from model outputs at the grid-cell scale (Supplementary Fig. 17). To preserve consistency, we derive the estimate based on Beer's law by exploiting the LAI value for the mixture of PFTs for each grid-cell, and use model outputs of mixture Tr and ET under the "historical" experiment. Results show the general agreement between these two sets of estimates, demonstrating the efficiency of Beer's law in deriving multi-year mean level of Tr/ET (Supplementary Fig. 17).

On the basis of Eq. (12), mean annual $B_{ts}$ for forests within each grid-cell during 1982-2014 can be estimated by the following equation:

$$B_{ts}^{forest} = \frac{H^{psl}}{LE^{psl}\left[1 - \exp(-k_{Rn} LAI^{psl})\right]} \qquad (13)$$

Similarly to the estimations of Tr/ET and $B_{ts}$ for forests shown above, Tr/ET and $B_{ts}$ for croplands within each grid-cell can be estimated by replacing variables in Eqs. (12) and (13) with those obtained based on the category of croplands (i.e., $LAI^{crop}$, $H^{crop}$, and $LE^{crop}$).

Considering the limited numbers (three) of ESMs providing subgrid-scale outputs, the abovementioned analysis applied for the estimation of $\Delta B_{ts}^{forest-crop}$ is complemented by a more general statistical framework extendible to the full ESM ensemble, thereby further

increasing the reliability of our findings about the mechanism responsible for the large model spread in $\delta ET^{LULCC}$ estimate (Fig. 6a, b). Similar to approaches applied in previous studies focusing on biophysical climate impact of LULCC[30,70], this framework assumes that, for a given grid-cell, the mean difference in $B_{ts}$ between two periods ($\Delta B_{ts}$) under the realistic simulation (i.e., the "historical" experiment) is equal to the sum of $B_{ts}$ variation induced by LULCC ($\Delta B_{ts}^{LULCC}$) plus the residual signal ($\Delta B_{ts}^{res}$) due to the rising atmosphere $CO_2$ concentration and climate change. $\Delta B_{ts}^{LULCC}$ reflects the change in energy partitioning associated to LULCC-induced changes in vegetation structure and physiology. For grid-cells experiencing significant conversions of forests to croplands, $\Delta B_{ts}^{LULCC} \approx \Delta B_{ts}^{forest-crop}$ in general, and its magnitude can be estimated as follows:

$$\Delta B_{ts}^{forest-crop} \approx \Delta B_{ts}^{LULCC} = \Delta B_{ts} - \Delta B_{ts}^{res} \qquad (14)$$

where the $\Delta$ term represents the mean difference between periods 2000-2014 and 1982-1996. These two 15-year periods are selected since they are temporally independent and represent energy conditions in early 21st century and late 20th century, respectively. Here, grid-cells experiencing significant conversions of forests to croplands are defined as those with both significant ($p < 0.05$, $t$-test) decrease in cover fraction of trees and significant increase in cover fraction of crops between the two periods (2000-2014 versus 1982-1996). In other words, for each grid-cell included in the land mask, $t$-test is performed twice: once between the two time-series (2000-2014 and 1982-1996) for cover fraction of trees, and once between the two time-series for cover fraction of crops. $\Delta B_{ts}^{res}$ is estimated from the signal of $B_{ts}$ variation captured in the adjacent eight grid-cells that surround the target grid-cell. The identified reference grid-cells need to meet three criterions: (1) similar initial land cover and land use condition, defined as the mean differences in cover fractions of trees and crops between reference grid-cells and target grid-cell during 1982-1996 within the range (-10% ~ 10%); (2) stable land cover and land use, namely nonsignificant ($t$-test, $p > 0.05$) changes in both cover fractions of trees and crops between the two periods; and (3) limited difference in elevation with respect to the target grid-cell (-100 m ~ 100 m). Elevation information is provided by CMIP6 ESMs, which is independent of time (variable "orog" in the CMIP6 terminology). For these reference grid-cells, we could assume $\Delta B_{ts}^{LULCC} \approx 0$ and consequently $\Delta B_{ts} \approx \Delta B_{ts}^{res}$. The aforementioned processing enables us to minimize the potential impact of spatial heterogeneity in environmental conditions, such as background vegetation characteristics, climate[111] and elevation[112] on our results. For cases where two or more adjacent reference grid-cells are identified, we use an inverse distance weighting to estimate $\Delta B_{ts}^{res}$ from the n ($n \geq 2$) reference grid-cells, according to the following equation:

$$\Delta B_{ts}^{res} = \frac{\sum_{i=1}^{n} \frac{\Delta B_{tsi}}{d_i}}{\sum_{i=1}^{n} \frac{1}{d_i}} \qquad (15)$$

where $d_i$ represents the distance from the target grid-cell to the reference grid-cell $i$. In such statistical framework, $\Delta B_{ts}^{forest-crop}$ can be derived as the difference between $B_{ts}$ variation in target grid-cell (i.e., $\Delta B_{ts}$ in Eq. (14)) and that in adjacent grid-cells (i.e., $\Delta B_{ts}^{res}$ in Eqs. (14) and (15)). Given that this statistical framework is primarily based on the pairing of neighboring grid-cells that meet specific criterions, we therefore name this statistical framework here "the paired grid-cell method".

Overall, two different methods are applied here to estimate $\Delta B_t^{forest-crop}$, an indicator to quantify the difference in energy partitioning between croplands and forests assumed by various ESMs. The first method relies on the subgrid-cell outputs of three ESMs (i.e., CESM2, NorESM2-LM, UKESM1-0-LL), while the second is based on the

paired comparison at the entire grid-cell scale and is applicable to all ESMs. Results based on the aforementioned methods suggest the substantial disagreement between ESMs in terms of the magnitude and even direction of $\Delta B_{ts}^{forest-crop}$, and such disagreement largely contributes to the large inter-model spread in estimate of hydrological effect of LULCC (Fig. 6).

## Data availability
All datasets used in this study are publicly available as referenced in "Methods". Source data are provided with this paper.

## Code availability
The custom MATLAB (R2024a) codes written to read and analyze data and generate figures are publicly available at https://doi.org/10.5281/zenodo.17020036.

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

## Acknowledgements

This work has received funding from the European Union's Horizon Europe research and innovation programme under the Marie Skłodowska-Curie Actions (grant agreement No. 101152010, TYPIC). Views and opinions expressed are, however, those of the author(s) only and do not necessarily reflect those of the European Union. Neither the European Union nor the granting authority can be held responsible for them. This is ClimTip contribution #89; the ClimTip project has received funding from the European Union's Horizon Europe research and innovation programme under grant agreement No. 101137601. R.X. was supported by the Fundamental Research Funds for the Central Universities (grant agreement No. B240207078) and China Scholarship Council (CSC) Grant (grant agreement No. 202406710180). Furthermore, we thank Dr. Zhongwang Wei (Sun Yat-Sen University) for providing the observational ET dataset derived by LAI-based upscaling. Certain Esri ® ArcGIS ® Imagery in this work are owned by Esri and/or its data contributors and are used herein with permission. Copyright © 2026 Esri and its data contributors. All rights reserved.

## Author contributions

Z.C. conceived and designed the research; Z.C. and R.X. collected and processed raw data, and implemented the data analysis; A.C. and G.F. contributed analysis ideas; Z.C. interpreted the results and drafted the initial manuscript; A.C. and G.F. provided suggestions and further improved writing. All authors approved the final version of this manuscript.

## Competing interests

The authors declare no competing interests.
