## [Transparent Peer Review file · Nature Communications]

Emergent constraints on the hydrological impacts of land use and land cover change

Corresponding Author: Dr Zefeng Chen

Version 0:

Reviewer comments:

Reviewer #1

(Remarks to the Author)

Review of "Emergent constraints on the impacts of land use and land cover change on the terrestrial water cycle" by Chen et al.

The study examines the effects of land use and land cover changes on evapotranspiration in CMIP6 model simulations and reports a global emergent constraint using a set of observational datasets. The constraint differs from the model mean, and the uncertainty also decreases. The authors analyze existing model output and publicly available observational datasets; the novelty is rooted in the type of analysis based on emergent constraints. The manuscript is very well written. However, I do not see a great added value in this study, because (a) some of the emergent constraint relationships do not appear robust, and (b) the reported global signals are very low and thus insignificant in the global water cycle. In my opinion, the drawn conclusions are overstated. I recommend restructuring the study to focus more on hot-spots of land use and land cover change (LULCC) and ET responses, and understand different ET responses on a regional scale, as this is where these water cycle responses matter. Therefore, the manuscript in its current state, with its focus on global constraints, does not fit the outlet of Nature Communications.

General Comments:

1. What do we really learn from a global change estimate in ET? Evapotranspiration patterns and changes are diverse, often balance out on continental scale, and LULCC are also quite diverse effects; Please argue why such a global estimate really matters. I think regional constraints or biome-specific constraints are more useful.
2. How does this compare to the overall terrestrial evapotranspiration? Global evapotranspiration is around 600 mm yr⁻¹ (<https://www.sciencedirect.com/science/article/abs/pii/S0022169423011666>). Thus, an average change of ~0.06 mm yr⁻² corresponds to a change of 0.01 percent. Even the highest outlier estimate in your study of 0.38 mm yr⁻² translates into a 0.06% change per year; Given all uncertainties in the LULCC reconstructions and the observational datasets you use, these signal are negligible.
3. Robustness of fits: Given the twelve data points in Supp. Fig 2a, it is not convincing to me to fit non-linear functions to such a small point cloud. Why do you expect the relationship to be inherently non-linear across models? Within one model, I could see that, but across the ensemble, I do not see why this would be the case. How does the constraint change if you do not apply the log-transform?
4. Relative change vs. absolute change: You write, e.g "Results show that the constrained δ ETLULCC is -0.031 mm yr⁻² at the global scale, which is opposite in sign and 155.4% lower than that of the original CMIP6 ensemble mean (0.057 mm yr⁻²)"; One can put small numbers in relation to each other so they appear high. This is misleading. Your CMIP6 mean indicates 0.01% yr⁻² increase in global evapotranspiration, and your constraint suggests 0.005% yr⁻¹ less global evapotranspiration (assuming 600 mm yr⁻¹). These numbers are negligible. If you compare 2 drops of water to 1 drop of water, you see a 100% increase, but you still have only 2 drops.

Specific Comments:

L23: Is this not an "enhanced" estimate? You report a negative rate in evapotranspiration.

L30: I could not see the increase in "credibility in projections" aspect emerging from your analyses. Could you specify?

L112: Could this also be related to sensor fusion and the onset of the MODIS time series?

L114: A verb is missing; please rephrase to "can largely be attributed to".

L145: Yes, but perhaps not a global estimate.

L156-158: Maybe add a sentence to provide more details on the products, e.g., upscaling products or remote sensing, to improve the flow of the manuscript for the reader.

L195: Please use the present tense consistently throughout the manuscript; change "estimated" to "estimate".

L221-226: Again, this is misleading, as both numbers are very small.

L299: "Greatly" reads a bit strangely here; better to use "strong".

L382-385: Given the low values, CO₂ fertilization and climate change globally have considerably stronger impacts, and the LULCC signal likely disappears completely.

Fig 3c: I do not see an inter-model spread emerging here; these fits are not convincing. The linear fit in Fig 3e is a stretch, and there is no clear pattern in the cloud of a few points.

(Remarks on code availability)

Reviewer #2

(Remarks to the Author)

This is an interesting and compelling manuscript. I raise a number of questions regarding both the methodology, as well as some of the discussion. I generally think that some re-writing is necessary in order to make the manuscript more compelling and more accessible. In particular, I find that some of the discussion through lines 311-324 or so provides a lot of insight into why some of the findings turn out they way they do herein. I would recommend bringing some of this information forward, so that the Reader is better prepared for where this manuscript is headed. Further, I find that the points raised in this section and others suggest there is a very strong latitudinal feature to this analysis and that presumably drives some of the results, in particular any decline in ET with LULCC... This suggests that perhaps a more latitudinal analysis, or perhaps a more regionally-based analysis that compares ACROSS regions might be more appropriate...? Or at least that latitude should figure far more prominently as an important independent variable in the analysis...?

Further, I am not entirely convinced by the methodology is entirely appropriate. Perhaps it could be a bit better defended...? There is no clear reason why a multi-model mean is likely to be better than some of the independent models integrated into it...? Further, why not simply base the analysis on Bts instead of the modelled representations of T...? Why only use this output to constrain already "strained" estimates...? In general, I do think this methodological orientation could be better defended and explained.

The findings, however, are uniquely interesting. Thus I look forward to reading the revised version of this submission.

Please see the attached submission pdf for additional comments and questions.

(Remarks on code availability)

Version 1:

Reviewer comments:

Reviewer #1

(Remarks to the Author)

Review of revised manuscript "Emergent constraints on the impacts of land use and land cover change on the terrestrial water cycle" by Chen et al.

I thank the authors for the in-depth and comprehensive revisions, and for addressing all my major and minor concerns. Especially, the inclusion of the regional assessment in the form of to-the-point visuals makes this study considerably stronger and clearer.

Overall, I still think that the global constraint the authors place at the centre of the manuscript is a) IMO not correctly interpreted and b) does not add much to our understanding of the global effects on the hydrological cycle induced by LULCC. In their rebuttal, the authors claim that I did not understand the analysis; however, they repeat the same rationale as in the manuscript: To demonstrate that the effect is non-negligible, they present the numbers as a relative change rather than an absolute change. As I wrote in my first review report, one can put small numbers in relation to each other so they appear higher; this is misleading. Your global constraint is $0.031 \pm 0.081 \text{ mm yr}^{-2}$; again, a) this is a very small number in the global hydrological cycle and it does not become more impactful when expressed as a relative contribution to the overall ET trend, and b) the interpretation that your constraint is qualitatively different from the CMIP6 mean is a huge over-interpretation. Yes, it switches the sign in the mean, but your uncertainty estimate is considerably larger than the effect size. Is the sign switch still present if you use a bootstrap approach when deriving the emergent relationship in the model ensemble (randomly sample models, conduct your EC analysis, and test whether the sign switch is robust, e.g., EC in Winkler et al., 2019)? Returning to the small effect size, looking at your PDFs in Figure 2, my interpretation is that neither the unconstrained nor the constrained estimate is robustly non-zero (if you leave out GISS-E2-1-G in your multi-model mean, the unconstrained mean

estimate is likely spot-on zero).

Thus, to back your central claim—"The constraint reverses the sign of the original ESM estimates obtained over the observational period ($0.057 \pm 0.155 \text{ mm yr}^{-2}$, mean \pm s.d.) and narrows the inter-model spread, ultimately leading to a more robust global estimate of the LULCC-driven ET change of $-0.031 \pm 0.081 \text{ mm yr}^{-2}$ "—you must a) test that the sign switch is robust, and b) show that your effect is statistically different from zero.

Bottom line, I stick to my recommendation to rearrange the manuscript to focus on your new analysis on hot-spots of land-use and land-cover change (LULCC) and ET responses, and give the different regional ET responses the spotlight, as this is where these water-cycle responses matter. Therefore, the manuscript should be revised accordingly before being considered for publication in Nature Communications.

(Remarks on code availability)

Reviewer #2

(Remarks to the Author)

I appreciate the effort the authors have put into revising the manuscript. I find it is much improved. I likewise appreciate the effort that the Authors have put into looking at and assessing such broad range of datasets and resources. The attempts at producing more realistic estimates of ET, in particular, are both compelling and fascinating. I do still have many questions. But I think it is appropriate to move forward to eventual publication of this submission.

I have still included some comments here and there in the pdf submission. Mostly minor things. But there are still some questions that continue to nag at me... I do think, for example, that some of the discussion about the relative variation in estimations of the Bowen Ratio (Bts), for example, that are available in the Supplementary information, could be brought forward a bit. And I am still a bit concerned about our ability to estimate ET based on tree type, since there is SO MUCH variation in the real world empirical output that is not any where near adequately captured in the model simplifications...

That said, I do think the Authors do an exceptional job of putting this information together and assessing it in meaningful and important ways... Thus, though I would ask the Authors to consider my remaining comments, I do think this submission is more or less ready for publication.

(Remarks on code availability)

Version 2:

Reviewer comments:

Reviewer #1

(Remarks to the Author)

I thank the authors for the detailed additional analyses and the restructuring/revisions of their manuscript. I appreciate that the authors took my concerns about presenting a global constraint that is not sig. different from zero as the main finding into account and agreed that the work should prominently focus on a regional-scale assessment. Thank you for the constructive discussion here. After three rounds of review and given the rebuttals to Reviewer 2 also, this work is ready for publication in Nature Communications.

(Remarks on code availability)

Reviewer #2

(Remarks to the Author)

I am impressed at the lengths the Authors have gone to, to respond to all the very detailed comments of both Reviewers (myself included). I think this is an impressive piece of work and look forward to seeing it in print. I do find that I am repeatedly inspired, in reading this text, to think about further issues that could be addressed and discussed. But this is perhaps the best reflection of the fact that this is an important piece of work and should further inspire others to take these ideas and points still further in future endeavours. Thus, rather than go into additional detailed comments, I would like to encourage publication at this time and hope that others will find this publication as thought provoking as I have.

I did notice here and there that some very minor editing of the manuscript is still possible... The attached provides some indication of possible improvements. Please note that I have made no attempt to be exhaustive. Thus more editing is likely possible... However, these editing issues are minor and not worthy of further discussion.

(Remarks on code availability)

First of all, we would like to thank the two reviewers for their insightful and constructive comments. In the revision of the manuscript, we tried to address all their comments and suggestions to improve the clarity of the methodology and the robustness of the analysis.

In the following, we respond to each reviewer's comment by referring to line numbers of the revised tracked version, when not differently indicated. References cited along with this response letter are reported at the bottom of the document.

Reviewer #1: Questions and our responses

The study examines the effects of land use and land cover changes on evapotranspiration in CMIP6 model simulations and reports a global emergent constraint using a set of observational datasets. The constraint differs from the model mean, and the uncertainty also decreases. The authors analyze existing model output and publicly available observational datasets; the novelty is rooted in the type of analysis based on emergent constraints. The manuscript is very well written. However, I do not see a great added value in this study, because (a) some of the emergent constraint relationships do not appear robust, and (b) the reported global signals are very low and thus insignificant in the global water cycle. In my opinion, the drawn conclusions are overstated. I recommend restructuring the study to focus more on hot-spots of land use and land cover change (LULCC) and ET responses, and understand different ET responses on a regional scale, as this is where these water cycle responses matter. Therefore, the manuscript in its current state, with its focus on global constraints, does not fit the outlet of Nature Communications.

We thank the reviewer for her/his comments. We have substantially revised the manuscript according to the reviewer's comments by further assessing the validity of our methods and the significance of the emerging signals. In addition, we have conducted new experiments focusing on hotspots of land use and land cover change. Such new analyses are described in response to specific comments below.

We sincerely appreciate the important comments raised by the reviewer, particularly the inclusion of regional scale analysis and the increase in confidence of robustness of emergent relationships. Addressing these comments have indeed substantially improved the quality of our analysis and further confirmed the relevance of our work and made the overall narrative more comprehensive.

We would like to stress here the novelty of our study, which – in our opinion – the reviewer may have not fully understood. Our study is not a mere numerical exercise based on the emergent constraint framework. On the contrary, by applying such a framework our contribution aims to provide a clearer understanding of the impact of historical land use and land cover change (LULCC) on terrestrial evapotranspiration (ET), both at the global extent and over IPCC AR6 reference regions. With the analogous emergent constraint, we further quantify the potential effect of future afforestation on ET and terrestrial water availability, all critical aspects

that remain elusive and not yet comprehensively analyzed in previous studies. In brief, our study provides a robust and comprehensive data-driven assessment of the LULCC impact on terrestrial water cycle during the historical period and under future afforestation scenarios, and identify hotspot regions where water availability is heavily influenced by land use change.

We believe our model-data integration approach and associated findings add significant value in the following ways:

- 1) We investigate the effect of historical LULCC on terrestrial ET both at the global and regional scales, through the integration of multi-source observations and Earth system model (ESM) simulations. In previous studies based on original simulations of ESMs and land surface models (LSMs, i.e., the land component of ESMs), the key LULCC-related signal is largely obscured by the uncertainty and bias of models themselves (Wang and Zeng, 2024). Such model limitations have so far hampered a comprehensive understanding and a robust assessment of the ongoing variations in the water cycle under land surface changes.
- 2) We project the impact of future afforestation on terrestrial water cycle and identify key hotspot areas where afforestation is expected to play a critical control on local water availability. Although some efforts have been recently devoted into related topics (Zan et al., 2024; Tang et al., 2025), our bias-corrected assessment with the help of emergent constraint framework provides a more robust projection of the potential threat of afforestation to water supply for societies, facilitating the development of effective land-based climate mitigation strategies.
- 3) We provide novel evidence that the inaccurate assessment of the hydrological effect of large-scale LULCC in CMIP6 ESM simulations is attributable to the model bias in representing the ratio between sensible heat and latent heat through transpiration and its variations across plant functional types (PFTs). Our study is the first one that has revealed the cause of the substantial discrepancy between state-of-the-art ESMs in representing LULCC effects on terrestrial water cycle. This finding highlights the urgent need to address the uncertainty of ESMs driven by the surface energy balance and provides new and important technical reference to the development of enhanced ESMs.

Action taken: We have clarified the novelty of our contribution in the revised version of the manuscript (see lines 360-363, 490-494). Furthermore, changes made to address other comments (e.g., regional scale analysis and additional validation tests) have been specifically described in the following responses.

General comments:

1. What do we really learn from a global change estimate in ET? Evapotranspiration patterns and changes are diverse, often balance out on continental scale, and LULCC are also quite diverse effects; Please argue why

such a global estimate really matters. I think regional constraints or biome-specific constraints are more useful.

Response: We perform the emergent constraint analysis at the global scale, primarily to provide a clearer overview of land use change effect on terrestrial water cycle during both historical and future scenario periods. The resulting bias-corrected global scale estimate helps to resolve the long-standing argument across model-based assessments about the direction and magnitude of changes in ET in response to LULCC and afforestation actions (Wang and Zeng, 2024) and elucidate the underlying mechanism.

We agree with the reviewer that regional assessments are more useful, especially given that global-scale results are potentially subject to spatial compensatory effects that may originate from areas with diverse LULCC and opposite ET response. In fact, in the originally submitted version, local-scale analyses were already considered but were initially performed in the climate- instead of the spatial-domain and reported at the climate bin level (Fig. 4 in the previous version). Now, considering the reviewer's suggestion, we have further enhanced our focus on hotspot regions and elaborated the emerging LULCC-related signals at the regional scale in more details. To this end, we have performed a series of new experiments over forty-four reference regions used in the Intergovernmental Panel on Climate Change Sixth Assessment Report (IPCC AR6), to provide specific constraints at regional level. Original IPCC AR6 reference regions include forty-six land regions and fifteen ocean regions, each with consistent regional climatic features (Iturbide et al., 2020). Here, all ocean regions and two land regions located in Antarctica are excluded from our analyses, since we focus exclusively on the vegetated land (Supplementary Fig. 7). To calculate the emergent constraints at the regional scale, we first extract the gridded datasets from observation-based products and model outputs for each reference region based on the map provided by Iturbide et al. (2020). We then estimate the regional averaged effect of historical LULCC on annual ET (δET^{LULCC}) and that of future afforestation on annual ET (δET^{AFF}) for each model (Eqs. (1)-(3)), the regional mean levels of observed and modelled transpiration-specific Bowen ratio (B_{ts} , Eq. (4)) and their natural logarithm ($\ln(B_{ts})$). Consistent with our global-scale aggregated metrics, these regional-scale estimates are also obtained by spatial average weighting each grid-cell value within the region based on its land area.

Unconstrained results based on the original simulations from CMIP6 model ensemble generated for the historical period suggest that the LULCC-related signal is much stronger at the regional scale (magnitude ranging from -1.376 ± 1.480 mm yr⁻² to 0.886 ± 0.967 mm yr⁻² across reference regions, Fig. 4 and Supplementary Table 9) compared to the global mean level (0.057 ± 0.155 mm yr⁻², Fig. 1b). This finding is in line well with the reviewer's intuition. Over 26.0% of global vegetated land, LULCC is believed to dominate historical change in local ET (relative contribution higher than 50%), according to the original CMIP6 model simulations (Supplementary Table 9). Similarly, more pronounced signals emerge at regional scale under the future afforestation scenario. For example, compared to the global mean level (0.024 ± 0.048 mm yr⁻², Fig. 3d), unconstrained projections suggest a much higher δET^{AFF} in Central Africa (CAF) whose magnitude reaches

0.420±0.515 mm yr⁻², indicating that afforestation could explain 74.1% of future change in local ET (CAF in Fig. 5a and Supplementary Table 10). These results highlight the key relevance of land use and land cover change in long-term variations of the regional water cycle.

We find that applying the emergent constraint always reduces the uncertainty in estimates of ET response (i.e., δET^{LULCC} and δET^{AFF}), although the degree of reduction varies across regions (Figs. 4 and 5a and Supplementary Tables 9 and 11). The inter-model correlation between δET^{LULCC} and $\ln(B_{ts})$ is statistically ($p < 0.05$) significant for twenty-one reference regions corresponding to 50.2% of the global vegetated land (hexagons with the diagonal lines in Fig. 4). Slightly more widespread patterns are found for the interplay between δET^{AFF} and $\ln(B_{ts})$ with twenty-five reference regions – covering 55.5% of the global vegetated land – showing statistically significant ($p < 0.05$) correlations (hexagons with the diagonal lines in Fig. 5a). These regions exhibit an average reduction in standard deviation after constraint by 21.7% and 24.9% during the historical and future scenario periods, respectively, more than 10-fold greater than the analogous estimates obtained for the remaining regions without significant correlations (1.8% and 1.3%) (Supplementary Tables 9 and 11).

Correction direction for estimates of δET^{LULCC} varies spatially, primarily because of the distinct types of land cover conversion occurring across regions during the period 1982-2014 and of their interplay with the specific background climate of the region. Over 50.2% of global vegetated land where the emergent relationship holds, areas characterized by an increase in forest area (17.0% of global vegetated land), generally shows a higher δET^{LULCC} after applying the emergent constraint compared to the unconstrained estimate, which is opposite to the global mean results (Fig. 4). By contrast, more regions have experienced a decrease in forest area, and generally show a constrained estimate of δET^{LULCC} lower than that one obtained from the unconstrained outputs, consistently with the global mean results (Figs. 2 and 4). In this respect, δET^{LULCC} over 13.2% of global land turns from positive to negative after applying the emergent constraint (Fig. 4 and Supplementary Table 9).

Under the simulated future afforestation scenario, conversions from croplands to forests are expected to occur in most regions (Supplementary Table 10, exception cases are explained in Methods), generally leading to a higher constrained estimate of δET^{AFF} than the analogous metric derived from the original model ensemble (Fig. 5a). The constrained value of δET^{AFF} differs greatly across afforested regions, ranging from -0.158±0.357 mm yr⁻² to 0.447±0.171 mm yr⁻² (Fig. 5a). Occurrence of such spatial difference in δET^{AFF} may arise from (1) the varying magnitude of increase in forest cover (Supplementary Table 10), and (2) the divergent hydrological sensitivity to forest gain across regions which is largely controlled by the local climate and landscape conditions (Zhang et al., 2017). Previous evidence revealed that ET is generally more sensitive to forest cover change in drier regions, highlighting the importance of the ability to access deeper soil moisture in water-limited environments (Ning et al., 2020). Consistent with the global mean level (0.051±0.031 mm yr⁻², Fig. 3d), constrained δET^{AFF} is positive over most (58.8%) part of the afforested land, suggesting the widespread

positive effect of afforestation on ET as reported in previous studies (Teuling et al., 2019; Yang et al., 2023). On the other hand, the negative value of constrained δET^{AFF} still exists in some regions after constraint, possibly because the shading by increased foliage cover could decrease the amount of radiation reaching the ground and further lead to the reduction in soil evaporation, which counteracts the increase in Tr due to afforestation (Ukkola et al., 2016).

For twenty-two regions (covering 51.1% of global vegetated land) where emergent relationship and correlation between δET^{AFF} and δP^{AFF} across ESMs are both statistically significant, local δP^{AFF} is of comparable magnitude to δET^{AFF} after constraint, particularly in tropics and subtropics (Fig. 5a,b). For example, δP^{AFF} in North Eastern Africa (NEAF) increases from $0.008 \pm 0.659 \text{ mm yr}^{-2}$ to $0.352 \pm 0.470 \text{ mm yr}^{-2}$, along with the increase in δET^{AFF} from $0.091 \pm 0.354 \text{ mm yr}^{-2}$ to $0.312 \pm 0.253 \text{ mm yr}^{-2}$ after applying the emergent constraint (Supplementary Table 10). We estimate that sixteen out of twenty-two reference regions which cover 32.9% of global vegetated land, exhibit an increase in δWA^{AFF} after constraint, corresponding to a weakened negative effect of afforestation on WA and even a transition to positive effect in the constrained projection (Fig. 5c and Supplementary Table 10). Such weakening of negative afforestation effect on WA after constraint is particularly evident in Central and South America, Sub-Saharan Africa and South Asia, where constrained δP^{AFF} is substantially higher than unconstrained one (Fig. 5b,c). The existence of regions with higher magnitude of positive δP^{AFF} than that of δET^{AFF} (i.e., constrained $\delta WA^{AFF} > 0$) suggests that the increase in ET associated with afforestation could be outweighed by more pronounced moisture recycling in these regions. Such patterns emerge in regions such as NEAF, Western Africa (WAF), South Asia (SAS), and Eastern Australia (EAU) (Fig. 5), and appear broadly consistent with observational evidence documenting the buffering effect of forest-climate feedback on local water depletion (O'Connor et al., 2021; Hoek van Dijke et al., 2022).

Overall, these results demonstrate that original ESM outputs tend to misrepresent the direction and magnitude of historical LULCC effect on regional ET (Fig. 4), and more importantly, overestimate the water scarcity induced by afforestation scenario, particularly in tropics and subtropics (Fig. 5c). These results ultimately support the value of bias-corrected simulations for a more robust assessment of land-based climate mitigation plans at the regional level.

Action taken: We have added the aforementioned results focusing on IPCC AR6 reference regions in the Results section (“Emergent constraints at the regional scale”) (see lines 261-357), and correspondingly, added the new Figs. 4 and 5 and Supplementary Fig. 7 and Tables 9-11 in the revised version. Meanwhile, we have added related discussion on findings obtained at the regional scale (see lines 389-393, 398-415, 472-484), and described the methodological details related to the experiments performed at the regional scale in the Methods section (“Spatial domain”) (see lines 700-718). To keep concise, Supplementary Tables 9-11 are not presented below.

Fig. 4 | Constrained effect of historical land use and land cover change on terrestrial evapotranspiration over IPCC AR6 reference regions. Within each hexagon, the filled color in the upper and lower parts represent the unconstrained estimate from the original CMIP6 model ensemble and the observationally constrained estimate, respectively. The size of the green dot indicates the relative reduction in standard deviation (RR_{σ}) after applying the hierarchical emergent constraint approach. The hexagons with the diagonal lines indicate that the emergent relationship is statistically significant at the 90% confidence level. In fact, for these hexagons, their emergent relationship's significance even passes the 95% confidence level. Specific values for making this plot are listed in Supplementary Table 9. Moreover, detailed information of these reference regions is provided in Supplementary Fig. 7.

a. δET^{AFF}

b. δP^{AFF}

c. δWA^{AFF}

Fig. 5 | Constrained projection of future afforestation impact on terrestrial water cycle over IPCC AR6 reference regions. (a) Constrained trend in annual evapotranspiration during the period 2015-2099 (δET^{AFF}), as derived under the future afforestation scenario (Methods). Within each hexagon, the filled color in the upper and lower parts represent the unconstrained estimate of δET^{AFF} from the original CMIP6 model ensemble and the observationally constrained estimate, respectively. The size of the green dot indicates the relative reduction in standard deviation (RR_{σ}) after applying the hierarchical emergent constraint approach. The hexagons with the diagonal lines indicate that the emergent relationship is statistically significant at the 90% confidence level. In fact, for these hexagons, their emergent relationship's significance even passes the 95% confidence level. **(b and c)** Same as **(a)**, but for the trends in annual precipitation (δP^{AFF}) and terrestrial water availability (δWA^{AFF}). The hexagons with the diagonal lines indicate that both the emergent relationship and the inter-model relationship between δET^{AFF} and δP^{AFF} are statistically significant at the 90% confidence level. Specific values for making these plots are listed in Supplementary Tables 10 and 11. Moreover, detailed information of these reference regions is provided in Supplementary Fig. 7.

Acronym	Full name	Acronym	Full name	Acronym	Full name
GIC	Greenland/Iceland	SAH	Sahara	CAU	Central Australia
NWN	Northwestern North America	WAF	Western Africa	EAU	Eastern Australia
NEN	Northeastern North America	CAF	Central Africa	SAU	Southern Australia
WNA	Western North America	NEAF	North Eastern Africa	NZ	New Zealand
CNA	Central North America	SEAF	South Eastern Africa	EAN	Eastern Antarctica
ENA	Eastern North America	WSAF	West Southern Africa	WAN	Western Antarctica
NCA	Northern Central America	ESAF	East Southern Africa	ARO	Arctic Ocean
SCA	Southern Central America	MDG	Madagascar	NPO	North Pacific Ocean
CAR	Caribbean	RAR	Russian Arctic	EPO	Equatorial Pacific Ocean
NWS	Northwestern South America	WSB	West Siberia	SPO	South Pacific Ocean
NSA	Northern South America	ESB	East Siberia	NAO	North Atlantic Ocean
NES	Northeastern South America	RFE	Russian Far East	EAO	Equatorial Atlantic Ocean
SAM	South American Monsoon	WCA	West Central Asia	SAO	South Atlantic Ocean
SWS	Southwestern South America	ECA	East Central Asia	ARS	Arabian Sea
SES	Southeastern South America	TIB	Tibetan Plateau	BOB	Bay of Bengal
SSA	Southern South America	EAS	East Asia	EIO	Equatorial Indian Ocean
NEU	Northern Europe	ARP	Arabian Peninsula	SIO	South Indian Ocean
WCE	Western and Central Europe	SAS	South Asia	SOO	Southern Ocean
EEU	Eastern Europe	SEA	South East Asia		
MED	Mediterranean	NAU	Northern Australia		

Supplementary Fig. 7. IPCC AR6 WGI reference regions. Full names corresponding to acronyms shown in figure are illustrated in the following table. Non-vegetated areas, defined as multi-year (1982-2014) average leaf area index (LAI) <math> < 0.15 \text{ m}^2 \text{ m}^{-2}</math>, are excluded in our analysis and are shown in grey. It should be noted that due to the limited number (nearly zero) of valid grid-cells, two polar regions (i.e., EAN and WAN) and twelve ocean regions (i.e., ARO, NPO, EPO, SPO, NAO, EAO, SAO, ARS, BOB, EIO, SIO, and SOO) are not considered in our regional-scale analysis. Original figure and associated files are provided in Iturbide et al.¹⁰.

2. How does this compare to the overall terrestrial evapotranspiration? Global evapotranspiration is around 600 mm yr⁻¹ (<https://www.sciencedirect.com/science/article/abs/pii/S0022169423011666>). Thus, an average

change of $\sim 0.06 \text{ mm yr}^{-2}$ corresponds to a change of 0.01 percent. Even the highest outlier estimate in your study of 0.38 mm yr^{-2} translates into a 0.06% change per year; Given all uncertainties in the LULCC reconstructions and the observational datasets you use, these signals are negligible.

Response: We believe that the reviewer may have misunderstood the analysis we have conducted and the relevance of our results in terms of magnitude. Although the global mean ET is around 600 mm yr^{-1} , as correctly reported by the reviewer, its overall trend during the period 1982-2014 derived from the ensemble of twelve CMIP6 models is $0.532 \pm 0.122 \text{ mm yr}^{-2}$. A similar increase rate is estimated from an ensemble of observation-based ET products used in this study ($0.412 \pm 0.195 \text{ mm yr}^{-2}$), and is further corroborated by previous studies using different datasets (e.g., Yan et al. (2013): 0.46 mm yr^{-2} ; Pan et al. (2020): $0.23 \pm 0.28 \text{ mm yr}^{-2}$; Ma et al. (2021): 0.31 mm yr^{-2} ; Yang et al. (2023): $0.66 \pm 0.38 \text{ mm yr}^{-2}$). The value of $\delta \text{ET}^{\text{LULCC}}$ estimated in our study represents the trend in ET induced solely by land use and land cover change during 1982-2014, and therefore its magnitude should be confronted to the overall trend in ET ($0.532 \pm 0.122 \text{ mm yr}^{-2}$), rather than the amount of annual ET (600 mm yr^{-1}). The above-mentioned approach is typically used in related literature to quantify the marginal effect of individual factor/process on the long-term trend in target variable (e.g., Zhu et al., 2017; Bastos et al., 2019; Zhang et al., 2022; Liu et al., 2025). The global mean value of $\delta \text{ET}^{\text{LULCC}}$ derived from the original CMIP6 model ensemble is $0.057 \pm 0.155 \text{ mm yr}^{-2}$, corresponding to a relative contribution of LULCC on historical trend in ET of $13.3 \pm 33.4\%$ (Fig. 1b and Supplementary Fig. 1). Such overall high relative contribution indicates that LULCC-induced signal should not be neglected, instead, deserves to be analyzed and constrained. We point out that the direct effect of changes in climate factors (e.g., precipitation, temperature and solar radiation) has a comparable contribution (22.5%) to global ET rise over a similar period (Chen et al., 2023). As for elevated atmospheric CO_2 concentration (eCO_2), its physiological effect is believed to have a small negative contribution due that the negative effect from eCO_2 -induced reduction in stomatal conductance could outweigh the positive effect from increase in biomass (i.e., CO_2 fertilization) (Lemordant et al., 2018; Liu et al., 2021; Yang et al., 2023). Overall, the LULCC effect on ET trend (the focus of our work) is of comparable magnitude to the effects on trend originating from climate change and eCO_2 . Such comparison with other key drivers of ET clearly highlights the relevance of LULCC effect in the long-term environmental influence on the terrestrial water cycle.

The new set of experiments designed to estimate $\delta \text{ET}^{\text{LULCC}}$ at the regional scale (see our response to your previous comment) further corroborates the considerable effect of LULCC on water fluxes. Indeed, the emerging signal in $\delta \text{ET}^{\text{LULCC}}$ estimated at the global level is subject to compensatory effects originating from different regions. When $\delta \text{ET}^{\text{LULCC}}$ is estimated at regional scale, we find that the ET response to historical LULCC and future afforestation can be much stronger compared to analogous estimates obtained at the global level. We estimate that in 78.4% (58.1%) of the global vegetated land, the absolute magnitude of constrained $\delta \text{ET}^{\text{LULCC}}$ is more than 2 times (5 times) the global mean level ($-0.031 \pm 0.081 \text{ mm yr}^{-2}$); and meanwhile, in 44.3% (17.6%) of the reference regions, the absolute magnitude of $\delta \text{ET}^{\text{AFF}}$ is more than 2 times (5 times) the

global mean level ($0.051 \pm 0.031 \text{ mm yr}^{-2}$), as derived from the observationally constrained CMIP6 model simulations (Figs. 2-5). Correspondingly, LULCC shows a higher contribution to ET variation at the regional scale (Supplementary Tables 9 and 10). We therefore believe that the additional analyses conducted at the regional scale suggested by the reviewer further clarify the importance of this phenomenon on the ET trend and the necessity of a better understanding of the LULCC impact on terrestrial water cycle.

Action taken: We have added these contents into the Discussion section, to highlight the relevance of LULCC in variations of terrestrial water cycle (see lines 398-415).

3. Robustness of fits: Given the twelve data points in Supp. Fig 2a, it is not convincing to me to fit non-linear functions to such a small point cloud. Why do you expect the relationship to be inherently non-linear across models? Within one model, I could see that, but across the ensemble, I do not see why this would be the case. How does the constraint change if you do not apply the log-transform?

Response: We apply the log-transformation to the global averaged value of B_{ts} mainly because the natural logarithm regression has a better performance in fitting the inter-model relationship between B_{ts} and δET^{LULCC} over 1982-2014 compared with the linear regression, as reflected by a higher determination coefficient (r^2) for the former fitting function (Supplementary Fig. 3a). This regression model shows the p -value lower than 0.01, and therefore its result is statistically significant even if it uses only twelve data points.

To further test whether the relationship between B_{ts} and δET^{LULCC} is inherently non-linear across models, we perform a statistical analysis by applying a random sampling technique with one-thousand replications. We therefore generate one-thousand data sets by randomly extracting samples from 60% of vegetated grid-cells globally without replacement. Each data set includes twelve δET^{LULCC} estimates and twelve associated B_{ts} values, each pair of which corresponds to one CMIP6 ESM. Subsequently, we fit linear, natural logarithmic, and quadratic regressions to the inter-model relationship between δET^{LULCC} and B_{ts} for each data set, and use r^2 and root mean squared error ($RMSE$) to identify the regression model that provides the best fit in each data set. We find that mean r^2 of natural logarithmic regression fitting reaches 0.878 and is higher than that of linear and quadratic ones (0.851 and 0.876) (Supplementary Fig. 4a). In addition, mean $RMSE$ of natural logarithmic regression fitting is 0.061 mm yr^{-2} , which is lower when compared to the fitting by linear and quadratic regressions (0.067 mm yr^{-2} and 0.063 mm yr^{-2}) (Supplementary Fig. 4b). As natural logarithmic regression has the best performance in fitting the inter-model relationship between B_{ts} and δET^{LULCC} , $\ln(B_{ts})$ is used in our emergent constraint framework as reference.

To further increase the confidence in our approach, we also compute and compare the inter-model correlation between δET^{LULCC} and $\ln(B_{ts})$, against that between δET^{LULCC} and B_{ts} , during one-thousand replications of random sampling similar to that described above. The Pearson's correlation coefficient between

δET^{LULCC} and $\ln(B_{ts})$ derived based on these one-thousand data sets reaches 0.936 ± 0.005 , higher than that one between δET^{LULCC} and B_{ts} (0.922 ± 0.006) (Supplementary Fig. 2a). Similarly, δET^{AFF} also exhibits a stronger correlation with $\ln(B_{ts})$ across models compared to that one obtained with B_{ts} (-0.878 ± 0.006 vs. -0.866 ± 0.006) (Supplementary Fig. 2c). Given the proportional relation between the correlation strength and the performance of emergent constraint, these additional analyses further confirm the utility of applying a log-transformation to B_{ts} in emergent relationship.

Moreover, we conduct additional analysis to examine whether applying the log-transformation or not would produce substantially different emergent constraint results. To this end, we replicate the analyses by replacing $\ln(B_{ts})$ with B_{ts} , and compare the constrained estimates derived based on B_{ts} with the original ones derived based on $\ln(B_{ts})$. Results based on B_{ts} show that the constrained δET^{LULCC} reaches -0.030 ± 0.072 mm yr⁻² at the global scale during the period 1982-2014, highly consistent with the estimate derived based on $\ln(B_{ts})$ (-0.031 ± 0.081 mm yr⁻²) (Fig. 2 and Supplementary Fig. 12a,b). As for δET^{AFF} , δP^{AFF} and δWA^{AFF} during the period 2015-2099, their global-mean values after the B_{ts} -based constraint are 0.052 mm yr⁻², 0.024 mm yr⁻² and -0.028 mm yr⁻², respectively, which are also in accordance with those after the $\ln(B_{ts})$ -based constraint (0.051 mm yr⁻², 0.022 mm yr⁻² and -0.029 mm yr⁻²) (Fig. 3 and Supplementary Fig. 12c,d). We further compare the constrained estimates derived based on B_{ts} against those derived based on $\ln(B_{ts})$ over IPCC AR6 reference regions (Supplementary Fig. 13). Results show the general agreement between these two sets of estimates, as reflected by the fact that the absolute differences in their magnitudes are very small in all regions (Figs.4 and 5a and Supplementary Fig. 13). Such high consistency of results derived based on B_{ts} and those based on $\ln(B_{ts})$ indicates that not applying the log-transformation to B_{ts} has a negligible impact on our results and further demonstrates the robustness of our constraints on hydrological impact assessments.

Altogether, these additional experiments demonstrate the robustness of our methodology and clarify the utility of applying the log-transformation to B_{ts} to better capture the inherent relationship between LULCC effect and surface energy partitioning across models.

Action taken: We have added these contents into the new Supplementary Text 2 (“Constrained results derived based on B_{ts} ” section) and briefly recalled them in the main text (see lines 815-819, 852-863). Correspondingly, we have added the new Supplementary Figs. 2, 4, 12, and 13 in the revised version. Therein, Supplementary Fig. 2 has been presented in our response to the Specific Comment #11.

Supplementary Fig. 4. Performance comparison of different regression model types in fitting the inter-model relationship for one-thousand random sampling. (a) Boxplot of one-thousand sets of determination coefficient (r^2) of linear, natural logarithmic, and quadratic regression models in fitting the inter-model relationship between the effect of land use and land cover change on annual evapotranspiration ($\delta\text{ET}^{\text{LULCC}}$) and the global averaged transpiration-specific Bowen ratio (B_{ts}) during the period 1982-2014. Each data set is generated by randomly extracting samples from 60% of vegetated grid-cells globally without replacement. Each set includes twelve $\delta\text{ET}^{\text{LULCC}}$ estimates and twelve associated B_{ts} values, each pair of which corresponds to one CMIP6 ESM. Boxplot elements: box = values of 25th and 75th percentiles; horizontal line = median; rectangle = mean; whiskers = values of 10th and 90th percentiles. (b) Same as (a), but for root mean square error (RMSE).

Supplementary Fig. 12. Constrained estimates derived based on transpiration-specific Bowen ratio (B_{ts}) (i.e., not applying the log-transformation). (a) Emergent relationship between the modelled effect of land use and land cover change on annual evapotranspiration ($\delta\text{ET}^{\text{LULCC}}$) and the modelled global averaged transpiration-specific Bowen ratio (B_{ts}) during the period 1982-2014. LULCC over this period is generally characterized as the conversion from forests to croplands at the

global extent. Each dot denotes a CMIP6 ESM result, which corresponds to $\delta\text{ET}^{\text{LULCC}}$ estimate shown in Fig. 1b. The black solid line indicates the best-fit regression line across ESMs, with correlation coefficient (r) provided in the panel. The green solid line indicates an observational correction based on Eq. (5) with associated slope (k_e) provided in label. The vertical blue dashed line and shaded areas represent the observation-based estimate of B_{ts} and its uncertainty (one standard deviation). Such observation-based estimate is derived from an eight-member ensemble of observation-based combined datasets (Methods). The horizontal red dashed line and shaded areas show the resulting constrained estimate of $\delta\text{ET}^{\text{LULCC}}$ and its uncertainty based on the hierarchical emergent constraint approach (Methods). (b) The probability density functions of global averaged $\delta\text{ET}^{\text{LULCC}}$ for the original results of CMIP6 ESMs (blue) and the observationally constrained results (red). (c) Same as (b), but for the global averaged effect of future (2015-2099) afforestation on annual evapotranspiration ($\delta\text{ET}^{\text{AFF}}$). (d) The probability density functions of global averaged effect of future (2015-2099) afforestation on annual precipitation ($\delta\text{P}^{\text{AFF}}$) for the original results of CMIP6 ESMs (blue) and the observationally constrained results (red) by Eq. (10). Such constraint on $\delta\text{P}^{\text{AFF}}$ is based on the combination of the significant linear regression shown in Fig. 3e with (c) the constrained value of $\delta\text{ET}^{\text{AFF}}$.

Supplementary Fig. 13. Constrained estimates over IPCC AR6 reference regions derived based on transpiration-specific Bowen ratio (B_{ts}) (i.e., not applying the log-transformation). Within each hexagon, the filled color in the upper and lower parts represent the unconstrained estimate from the original CMIP6 model ensemble and the B_{ts} observation-constrained estimate, respectively. The size of the green dot indicates the relative reduction in standard deviation (RR_{σ}) after applying the hierarchical emergent constraint approach. The hexagons with the diagonal lines indicate that the emergent relationship is statistically significant at the 90% confidence level. Detailed information of these reference regions is provided in Supplementary Fig. 7.

4. Relative change vs. absolute change: You write, e.g. “Results show that the constrained δET^{LULCC} is $-0.031 \text{ mm yr}^{-2}$ at the global scale, which is opposite in sign and 155.4% lower than that of the original CMIP6 ensemble mean (0.057 mm yr^{-2})”; One can put small numbers in relation to each other so they appear high. This is misleading. Your CMIP6 mean indicates $0.01\% \text{ yr}^{-2}$ increase in global evapotranspiration, and your constraint suggests $0.005\% \text{ yr}^{-1}$ less global evapotranspiration (assuming 600 mm yr^{-1}). These numbers are negligible. If you compare 2 drops of water to 1 drop of water, you see a 100% increase, but you still have only 2 drops.

Response and action taken: According to the reviewer’s comment, we have changed the expression of relative term to the absolute term when comparing the difference in magnitude between constrained estimates and unconstrained ones. Nevertheless, we still would like to use relative reduction in standard deviation (RR_{σ}) to reflect the reduction in uncertainty after applying the emergent constraint, since RR_{σ} as well as relative reduction in variance (RRV) is a key indicator for the emergent constraint performance evaluation, and has been widely used in related studies (e.g., Shiogama et al., 2022; Dai et al., 2024; Chai et al., 2025).

Furthermore, as illustrated in our response to the previous comment (General Comment #2), the LULCC effect on ET trend is substantial and is of comparable magnitude to the effects on the trend from other environmental factors, such as climate change and elevated CO_2 . On the basis of original simulations from CMIP6 model ensemble, we find that compared to the global mean level ($0.057 \pm 0.155 \text{ mm yr}^{-2}$), such LULCC-related signal is much stronger at the regional scale, whose magnitude ranges from $-1.376 \pm 1.480 \text{ mm yr}^{-2}$ to $0.886 \pm 0.967 \text{ mm yr}^{-2}$ over IPCC AR6 reference regions (Fig. 4 and Supplementary Table 9). Results obtained at the regional scale further demonstrate the strong interplay between changes in land surface and terrestrial water cycle.

Specific comments:

1. L23: Is this not an “enhanced” estimate? You report a negative rate in evapotranspiration.

Response and action taken: “Enhanced” here represents the improvement in estimation robustness after bias correction and reduction in uncertainty, instead of the positive rate in evapotranspiration. To avoid misunderstanding, we have changed “enhanced” to “more robust” (see lines 23-24).

2. L30: I could not see the increase in “credibility in projections” aspect emerging from your analyses. Could you specify?

Response: Our study shows that the original simulations of ESMs usually report highly uncertain and counterintuitive results about the ET response to historical LULCC. By contrast, when integrating model simulations with the proposed emergent constraint framework, the derived constrained results could well

reproduce the historical patterns of LULCC effect on ET, that is a negative δET^{LULCC} at the global scale. How well the model or forecast method reproduces historical data and known patterns is a crucial indicator of credibility (Brekke et al., 2008). In this respect, we stressed in previous version that our model-data integration indeed could enhance the credibility of hydrological projections under LULCC. However, we realize that this may not be a widely accepted statement, so we have avoided using the word “credibility” in the revised version of the manuscript (see line 32).

3. L112: Could this also be related to sensor fusion and the onset of the MODIS time series?

Response: Land use and land cover forcing employed in CMIP6 ESMs is derived from the Land-Use Harmonization product (LUH2) (Hurtt et al., 2020). LUH2 product for the historical period (up to 2014) is generated based on the History database of the Global Environment (HYDE 3.2) and the Food and Agriculture Organization of the United Nations (FAO) agricultural land use data, and constrained by Landsat remote sensing observations. Information from MODIS satellite imagery are not exploited in the LUH2 product. Therefore, the contrasting patterns of change in global grassland area before and after 2000 cannot be attributed to the sensor fusion and the onset of the MODIS time series.

4. L114: A verb is missing; please rephrase to “can largely be attributed to”.

Response and action taken: We have modified the text according to the reviewer’s comment (see line 113).

5. L145: Yes, but perhaps not a global estimate.

Response: Following the reviewer’s suggestion (General Comment #1), we have reshaped the narrative, focusing prominently on the emergent constraint results at the regional scale. We believe that our response and actions (a series of new experiments performed at regional scale) detailed in the answers to the previous comments have addressed this specific issue.

6. L156-158: Maybe add a sentence to provide more details on the products, e.g., upscaling products or remote sensing, to improve the flow of the manuscript for the reader.

Response and action taken: Following the reviewer’s suggestion, we have added a sentence to further clarify the observation-based products used here. Details are as follow: “Specifically, observation-based Tr/ET estimates used here are derived from LAI-based upscaling approach⁴⁴, and remote sensing models⁴⁵⁻⁴⁷, in combination with 20CR v3⁴⁸ and ERA5-Land climate datasets⁴⁹.” (see lines 159-162).

7. L195: Please use the present tense consistently throughout the manuscript; change “estimated” to “estimate”.

Response and action taken: We have carefully checked the manuscript and corrected the text – where appropriate – using the present tense consistently, as suggested by the reviewer.

8. L221-226: Again, this is misleading, as both numbers are very small.

Response: We believe that our response and associated actions provided in response to the reviewer’s General Comment #4 have already addressed this specific issue.

9. L299: “Greatly” reads a bit strangely here; better to use “strong”.

Response and action taken: We have changed “greatly” to “strong”, as proposed by the reviewer (see line 376).

10. L382-385: Given the low values, CO₂ fertilization and climate change globally have considerably stronger impacts, and the LULCC signal likely disappears completely.

Response: We stress that LULCC effect on global ET trend is of comparable magnitude to the effects on trend originating from climate change and elevated atmospheric CO₂, as shown in our response to the reviewer’s General Comment #2. Furthermore, results of our new analyses performed at the regional scale show that LULCC plays a far larger and even dominant role in influencing terrestrial ET and water availability in some regions across the globe. For example, δET^{AFF} derived from the unconstrained model outputs is projected to be 0.420 ± 0.515 mm yr⁻² in Central Africa (CAF), suggesting that afforestation could explain 74.1% of future change of local ET (Fig. 5 and Supplementary Table 10). Even after applying the emergent constraint, the constrained estimate of δET^{AFF} could reach 0.433 ± 0.363 mm yr⁻², and corresponding δWA^{AFF} is expected to be -0.195 mm yr⁻², which is ~6.7 times of the global mean level (-0.029 mm yr⁻²) (Fig. 5 and Supplementary Table 10). These results reinforce the importance of accounting for the LULCC to properly understand the driving mechanism behind the hydrological change, particularly at the regional scale.

In this respect, we believe that the addition of a series of regional-scale analyses mentioned previously in response to the reviewer’s General Comment #1 has addressed this specific issue.

Action taken. We have added the new results derived at the regional scale in the revised version of the manuscript (see lines 261-357, 389-393, 398-415, 472-484), and correspondingly, added the new Figs. 4 and 5 and Supplementary Fig. 7 and Tables 9-11.

11. Fig 3c: I do not see an inter-model spread emerging here; these fits are not convincing. The linear fit in Fig 3e is a stretch, and there is no clear pattern in the cloud of a few points.

Response: The inter-model correlation between δET^{AFF} and $\ln(B_{ts})$ reaches 0.878, and that one between δET^{AFF} and δP^{AFF} is 0.707 (Fig. 3c,e). Both these two fits are statistically significant, with the p -value lower than 0.01 and 0.05, respectively, even if using only nine points. Therefore, we firmly believe that these analyses are robust, and the model fittings are significant. Nine points are considered here because only nine ESMs provide all required diagnostic variables under the future afforestation scenarios within CMIP6 project (details in Methods).

To further enhance the confidence on the robustness of the estimated inter-model relationships between δET^{AFF} and $\ln(B_{ts})$ (and δP^{AFF}), we apply a random sampling technique with one-thousand replications, and generate one-thousand data sets by randomly extracting samples from 60% of vegetated grid-cells globally without replacement. The additional validation analyses are similar to those presented in response to the reviewer's General Comment #3 to test the potential effect of log-transformation to B_{ts} . We compute the inter-model correlation between $\ln(B_{ts})$ and δET^{AFF} , and that between δET^{AFF} and δP^{AFF} for each data set. Results show that Pearson's correlation coefficient between $\ln(B_{ts})$ and δET^{AFF} derived based on these one-thousand data sets reaches -0.878 ± 0.006 , with the p -value of $18.6 \times 10^{-4} \pm 3.0 \times 10^{-4}$ (Supplementary Figure 2c,d). The Pearson's correlation coefficient between δET^{AFF} and δP^{AFF} reaches 0.719 ± 0.013 , with the p -value of 0.029 ± 0.004 (Supplementary Figure 2e,f). Such consistent and significant inter-model correlations (all p -value lower than 0.05) demonstrate the robustness of the estimated relationships between $\ln(B_{ts})$ and δET^{AFF} , and between δET^{AFF} and δP^{AFF} , regardless of the sampling set. More importantly, the performance of the emergent constraint in terms of bias correction and reduction in uncertainty is proportionally dependent on the correlation strength, as shown and discussed in the manuscript (Figs.4 and 5, see lines 288-300, 845-851). The consistent strong correlations revealed here reinforce the efficiency of our emergent constraint framework in improving model-based assessment on hydrological effect of future afforestation.

Moreover, to further verify the robustness of our emergent constraint results derived under future afforestation scenario, we conduct nine sets of additional experiments, each of which leaves one specific ESM out of the analysis of the emergent relationship. Results show high inter-model correlations between $\ln(B_{ts})$ and δET^{AFF} (all p -value lower than 0.01) and between δET^{AFF} and δP^{AFF} (p -value lower than 0.05 for seven out of nine, and lower than 0.10 for the remaining two) across all additional experiments (Supplementary Table 8). Furthermore, the constrained values of δET^{AFF} derived in all experiments distribute in a narrow range, from 0.047 ± 0.032 mm yr⁻² to 0.055 ± 0.033 mm yr⁻², and it is generally close to that obtained from the full ESM ensemble (0.051 ± 0.031 mm yr⁻²) (Fig. 3d). Similar results can also be found in the constrained projections of δP^{AFF} and the ultimate effect on terrestrial water availability ($\delta WA^{AFF} = \delta P^{AFF} - \delta ET^{AFF}$)

(Supplementary Table 8). Such high consistency between results from these additional experiments performed on reduced ensemble members and those derived based on the full ESM ensemble, further demonstrates that the strong inter-model correlations consistently exist and our emergent constraints are robust and not conditioned by any specific ESM.

Action taken: We have described the aforementioned experiments and associated results in the new Supplementary Text 1 (“Test on the robustness of emergent constraint approach” section) and briefly recalled them in the main text (see lines 228-232, 869-872) with references to the new Supplementary Fig. 2c-f and Table 8. We believe that these new analyses could help to (1) improve the robustness of our results about the strong inter-model correlation between $\ln(B_{ts})$ and δET^{AFF} , and that between δET^{AFF} and δP^{AFF} , and (2) support the validity of the use of emergent constraint framework to improve the model projection of the hydrological response to future afforestation. Moreover, to further improve the robustness of our emergent constraint results derived for the historical period, we have also applied the random sampling technique to the inter-model relationship between $\ln(B_{ts})$ and δET^{LULCC} (Supplementary Fig. 2a,b), and briefly recalled the associated results in the main text (see lines 165-168).

Supplementary Fig. 2. Inter-model correlation and associated significance level for one-thousand random sampling.

(a) Frequency distribution of one-thousand sets of inter-model correlation (r) between the effect of historical land use and land cover change (generally characterized as forest-cropland conversion) on annual evapotranspiration (δET^{LULCC}) and the global averaged transpiration-specific Bowen ratio (B_{ts}), as well as r between δET^{LULCC} and natural logarithm value of B_{ts} (i.e., $\ln(B_{ts})$) during the period 1982-2014. Each data set is generated by randomly extracting samples from 60% of vegetated grid-cells globally without replacement. Each set includes twelve δET^{LULCC} estimates and twelve associated B_{ts} values, each pair of which corresponds to one CMIP6 ESM. Distribution averages are shown as dashed horizontal lines. (b) Same as (a), but for the p -value associated to the inter-model correlation coefficient between δET^{LULCC} and B_{ts} , and that between δET^{LULCC} and $\ln(B_{ts})$. (c and d), Same as (a and b), but for r with the effect of future (2015-2099) afforestation on annual

evapotranspiration (δET^{AFF}), and the associated p -value. During one-thousand random sampling procedure, each set includes nine δET^{AFF} estimates and nine associated B_{ts} values, each pair of which corresponds to one CMIP6 ESM. (e and f), Same as (a and b), but for r between δET^{AFF} and the modelled effect of future (2015-2099) afforestation on annual precipitation (δP^{AFF}), and the associated p -value.

Supplementary Table 8. Constrained projection of future afforestation impact on terrestrial water cycle after excluding the specific model. For each sensitivity experiment by leaving specific model out of the emergent relationship, the associated inter-model correlation (r) between the effect of future (2015-2099) on annual evapotranspiration (δET^{AFF}) and the historical (1982-2014) natural logarithm value of global averaged transpiration-specific Bowen ratio ($\ln(B_{ts})$), and that between δET^{AFF} and the effect of future (2015-2099) on annual precipitation (δP^{AFF}), and the resulting constrained estimate of δET^{AFF} , δP^{AFF} , and the ultimate effect on terrestrial water availability ($\delta WA^{AFF} = \delta P^{AFF} - \delta ET^{AFF}$, see Eq. (11)) are provided below. Constrained values of δET^{AFF} and δP^{AFF} are expressed as mean \pm standard deviation.

Model left out	$r(\ln(B_{ts}), \delta ET^{AFF})$	$r(\delta ET^{AFF}, \delta P^{AFF})$	Constrained value (mm yr ⁻²)		
			δET^{AFF}	δP^{AFF}	δWA^{AFF}
BCC-CSM2-MR	-0.854 ($p < 0.01$)	0.712 ($p < 0.05$)	0.052 \pm 0.033	0.022 \pm 0.046	-0.030
CanESM5	-0.845 ($p < 0.01$)	0.622 ($p < 0.10$)	0.052 \pm 0.025	0.024 \pm 0.035	-0.028
CESM2	-0.889 ($p < 0.01$)	0.709 ($p < 0.05$)	0.050 \pm 0.033	0.023 \pm 0.041	-0.027
CMCC-ESM2	-0.890 ($p < 0.01$)	0.708 ($p < 0.05$)	0.055 \pm 0.033	0.023 \pm 0.040	-0.032
GFDL-ESM4	-0.879 ($p < 0.01$)	0.747 ($p < 0.05$)	0.053 \pm 0.033	0.035 \pm 0.044	-0.018
IPSL-CM6A-LR	-0.887 ($p < 0.01$)	0.788 ($p < 0.05$)	0.054 \pm 0.033	0.039 \pm 0.039	-0.016
MIROC-ES2L	-0.861 ($p < 0.01$)	0.707 ($p < 0.05$)	0.047 \pm 0.032	0.024 \pm 0.043	-0.024
NorESM2-LM	-0.929 ($p < 0.01$)	0.683 ($p < 0.10$)	0.048 \pm 0.029	0.013 \pm 0.034	-0.034
UKESM1-0-LL	-0.866 ($p < 0.01$)	0.723 ($p < 0.05$)	0.051 \pm 0.033	0.005 \pm 0.032	-0.045

References:

- Bastos, A. et al. Contrasting effects of CO₂ fertilization, land-use change and warming on seasonal amplitude of Northern Hemisphere CO₂ exchange. *Atmos. Chem. Phys.* **19**, 12361-12375 (2019).
- Brekke, L. et al. Significance of model credibility in estimating climate projection distributions for regional hydroclimatological risk assessments. *Clim. Change*, **89**, 371-394 (2008).
- Chai, Y., Yue, Y., Slater, L. & Miao, C. Emergent constraints indicate slower increases in future global evapotranspiration. *npj Clim. Atmos. Sci.* **8**, 46 (2025).
- Chen, Z., Wang, W., Cescatti, A. & Forzieri, G. Climate-driven vegetation greening further reduces water availability in drylands. *Glob. Change Biol.* **29**, 1628-1647 (2023).
- Dai, P., Nie, J., Yu, Y. & Wu, R. Constraints on regional projections of mean and extreme precipitation under warming. *Proc. Natl. Acad. Sci. USA* **121(11)**, e2312400121 (2024).
- Hoek van Dijke, A. et al. Shifts in regional water availability due to global tree restoration. *Nat. Geosci.* **15**, 363-368 (2022).
- Hurt, G. et al. Harmonization of global land use change and management for the period 850-2100 (LUH2) for CMIP6. *Geosci. Model Dev.* **13**, 5425-5464 (2020).
- Iturbide, M. et al. An update of IPCC climate reference regions for subcontinental analysis of climate model data: definition and aggregated datasets. *Earth Syst. Sci. Data* **12**, 2959-2970 (2020).
- Lemordant, L. et al. Critical impact of vegetation physiology on the continental hydrologic cycle in response to increasing CO₂. *Proc. Natl. Acad. Sci. USA* **115(16)**, 4093-4098 (2018).
- Liu, J. et al. Response of global land evapotranspiration to climate change, elevated CO₂, and land use change. *Agric. For. Meteorol.* **311**, 108663 (2021).
- Liu, M., Lin, K. & Tu, X. Increasing evapotranspiration in China: Quantifying the roles of CO₂ fertilization, climate and vegetation changes. *Water Resour. Res.* **61**, e2024WR038148 (2025).
- Ma, N., Szilagyi, J. & Zhang, Y. Calibration-free complementary relationship estimates terrestrial evapotranspiration globally. *Water Resour. Res.* **57**, e2021WR029691 (2021).

13. Ning, T. et al. Effects of forest cover change on catchment evapotranspiration variation in China. *Hydrol. Process.* **34**, 2219-2228 (2020).
14. O’Conner, J. et al. Forests buffer against variations in precipitation. *Glob. Change Biol.* **27**, 4686-4696 (2021).
15. Pan, S. et al. Evaluation of global terrestrial evapotranspiration using state-of-the-art approaches in remote sensing, machine learning and land surface modeling. *Hydrol. Earth Syst. Sci.* **24**, 1485-1509 (2020).
16. Shiogama, H., Watanabe, M., Kim, H. & Hirota, N. Emergent constraints on future precipitation changes. *Nature* **602**, 612-616 (2022).
17. Tang, T., Ge, J., Cao, J. & Shi, H. Land water availability altered by historical land use and land cover change. *npj Clim. Atmos. Sci.* **8**, 230 (2025).
18. Teuling, A. et al. Climate change, reforestation/afforestation, and urbanization impacts on evapotranspiration and streamflow in Europe. *Hydrol. Earth Syst. Sci.* **23**, 3631-3652 (2019).
19. Ukkola, A. et al. Reduced streamflow in water-stressed climates consistent with CO₂ effects on vegetation. *Nat. Clim. Change* **6**, 75-78 (2016).
20. Yan, H. et al. Diagnostic analysis of interannual variation of global land evapotranspiration over 1982–2011: Assessing the impact of ENSO. *J. Geophys. Res. Atmos.* **118**, 8969-8983 (2013).
21. Yang, Y. et al. Evapotranspiration on a greening Earth. *Nat. Rev. Earth. Environ.* **4**, 626-641 (2023).
22. Zan, B. et al. Spatiotemporal inequality in land water availability amplified by global tree restoration. *Nat. Water* **2**, 863-874 (2024).
23. Zhang, M. et al. A global review on hydrological responses to forest change across multiple spatial scales: Importance of scale, climate, forest type and hydrological regime. *J. Hydrol.* **546**, 44-59 (2017).
24. Zhang, X. et al. CO₂ fertilization is spatially distinct from stomatal conductance reduction in controlling ecosystem water-use efficiency increase. *Environ. Res. Lett.* **17**, 054048 (2022).
25. Zhu, Z. et al. Attribution of seasonal leaf area index trends in the northern latitudes with “optimally” integrated ecosystem models. *Glob. Change Biol.* **23**, 4798-4813 (2017).

Reviewer #2: Questions and our responses

This is an interesting and compelling manuscript. I raise a number of questions regarding both the methodology, as well as some of the discussion. I generally think that some re-writing is necessary in order to make the manuscript more compelling and more accessible. In particular, I find that some of the discussion through lines 311-324 or so provides a lot of insight into why some of the findings turn out the way they do herein. I would recommend bringing some of this information forward, so that the Reader is better prepared for where this manuscript is headed. Further, I find that the points raised in this section and others suggest there is a very strong latitudinal feature to this analysis and that presumably drives some of the results, in particular any decline in ET with LULCC... This suggests that perhaps a more latitudinal analysis, or perhaps a more regionally-based analysis that compares across regions might be more appropriate...? Or at least that latitude should figure far more prominently as an important independent variable in the analysis...?

Further, I am not entirely convinced by the methodology is entirely appropriate. Perhaps it could be a bit better defended...? There is no clear reason why a multi-model mean is likely to be better than some of the independent models integrated into it...? Further, why not simply base the analysis on B_{ts} instead of the modelled representations of T ...? Why only use this output to constrain already "strained" estimates...? In general, I do think this methodological orientation could be better defended and explained.

The findings, however, are uniquely interesting. Thus, I look forward to reading the revised version of this submission.

Please see the attached submission pdf for additional comments and questions.

We thank the reviewer for her/his overall positive comments and the constructive input that helped us to improve the clarity of the methods and the relevance of the findings.

In accordance to the reviewer's suggestions, we have structured our revisions on:

- 1) Improving the clarity of the discussion and key findings
- 2) Exploring regional-scale patterns of emergent constraints
- 3) Clarifying the methodology and providing additional evidence of the validity of the overall modelling framework

Specific comments and associated responses have been summarized below, according to the annotations shown in the pdf document uploaded.

Specific comments:

1. Lines 23-24: Perhaps it is important to explain why this coefficient is negative? And how is this related to the change in LULCC? I was honestly confused by this throughout the text... Are you saying that there is a

slope change, but that afforestation will nonetheless increase ET, but by a smaller amount? It is of course illogical to believe that ET would decline with more forest cover, since trees will not "grow" unless they produce ET. Without some interpretation here, this seems to be the suggestion. See, however, the comment to lines 311-324. Some of this text is potentially meaningful here and might help the reader to better follow some of the later conclusions.

Response: The negative trend in evapotranspiration (ET) induced by historical land use and land cover change (LULCC) (i.e., -0.031 ± 0.081 mm yr⁻²) is primarily because global land experiences an overall conversion from forests to croplands during the period 1982-2014 (Figs. 1a and 2). The decrease in ET under the historical conversion from forests to croplands can be attributed to the associated reductions in canopy density, root depth, and growing season length, which tend to ultimately result in lower evaporative capacity in croplands compared to forests (Yang et al., 2023). By contrast, under the future afforestation scenario (i.e., conversion from other vegetation types to forests), global ET is expected to show a positive trend with the rate of 0.051 ± 0.031 mm yr⁻² over 2015-2099 (Fig. 3). These two sets of results respectively derived for the historical period and future scenario, both meet the reviewer's expectation, that is, more forest cover gets more ET and vice versa.

Action taken: We realize that the overall characteristics of historical LULCC were not well explained before the Discussion section in the previous version of the manuscript, leading the readers to mistakenly assume that global land has experienced afforestation during the historical period. To avoid misunderstanding, we have added the description about the historical LULCC characteristics into the different parts of the Results section (see lines 110-111, 120-121, 183-184) and captions of figures such as Figs. 2 and 6 and Supplementary Figs. 2, 6, and 12. Following the reviewer's suggestion, we have moved some contents previously presented in the Discussion section to the front (see lines 187-193).

2. Lines 36-39: This does not seem to adequately represent the precipitation recycling literature and the role of forests in promoting additional ET and precipitation.

See e.g., <https://doi.org/10.1016/j.gloenvcha.2017.01.002>; <https://doi.org/10.1111/gcb.17195>

Response and action taken: According to the reviewer's suggestion, we have added these two references.

3. Lines 40-41: I would suggest the determinant feature here is something else, and not plant physiological and structural characteristics, per se...? Suggest revising... Consider, e.g., <https://doi.org/10.1111/gcb.17195>

Response and action taken: LULCC could mainly influence the scale and timing of terrestrial ET, as plant functional types (PFTs) differ in growing-season length, root development, leaf traits, and water-use efficiency (Sterling et al., 2013; Yang et al., 2023; Tang et al., 2025). We have revised this sentence to improve its

accessibility (see lines 40-43).

4. Lines 41-43: There is a much broader literature here. Could be better seated in that literature, e.g., <https://doi.org/10.1016/j.gloenvcha.2017.01.002>

Response and action taken: According to the reviewer's suggestion, we have added the reported reference, i.e., Ellison et al. (2017).

5. Lines 43-45: This overview seems very catchment-centric and does not seem to address larger regional impacts on water availability. Recommend referencing also some of the precipitation recycling literature... For articles the link this specifically to forest cover, see: <https://doi.org/10.1016/j.gloenvcha.2017.01.002>

Response and action taken: We have replaced Feng et al. (2016) by Ellison et al. (2017), to provide more solid evidence on the impacts of LULCC on water availability at the larger regional scale.

6. Line 53: Recommend: <https://www.nature.com/articles/s41561-022-00935-0>. Note that Feng et al., is much too catchment centric to adequately capture the relevant hydrologic cycle relationships and net outcomes. See also: <https://onlinelibrary.wiley.com/doi/full/10.1111/j.1365-2486.2011.02589.x>

Response and action taken: We have replaced Feng et al. (2016) with the suggested references, i.e., Hoek van Dijke et al. (2022) and Ellison et al. (2012), to better illustrate the previous findings on the relationship of variations in hydrological cycle and net outcomes with forest cover change at the broad spatial scale.

7. Lines 90-91: It is not immediately clear why a mean derived from multiple and contradictory models would necessarily be better than say any particular one of the models... Some may get it right. Some may get it wrong...? For example, why not average the models that seem to get it right...? Can you say more here...? Why is any one of these strategies "more accurate"...? I have more tolerance for the "constraint framework" employed here, but would like to see it better explained... Why does this strategy work...? But still, why not just restrict the analysis to the sole use of the B_{ts} framework...? And ignore what the other models do, or compare results from the B_{ts} framework alone with other models...??? You may also find the following article to be of interest in this context: <http://journals.ametsoc.org/doi/10.1175/JCLI-D-16-0213.1>

Response: First, we would like to clarify the core concept of the emergent constraint framework. Despite substantial differences across ESMs, relationships between two elements (X and Y , respectively) are implicit within ESM solutions of the complex equations and associated parameterizations (Hall et al., 2019). The spread in variable X and variable Y may be large across ESMs, but there is strong and robust relationship

between these two variables, i.e., $Y = f(X) + \varepsilon$, where ε is a relatively small departure from f . If X is a quantity that can be observed, then the relationship f may place a useful constraint on Y , provided the observation uncertainty in X is small compared to the range of simulated values. This approach has the label “emergent” because the function f cannot be diagnosed from a single ESM but rather becomes apparent through analysis of a suitably large ESM ensemble (Keenan et al., 2023).

In this respect, when applying the emergent constraint framework, bias correction and uncertainty reduction are not performed for one specific ESM, as produced in previous works focusing on model development issues (e.g., Clark et al., 2015; Bonan and Doney, 2018; Mcdowell et al., 2020). We do not need to evaluate which model simulation is right or wrong, nor do we consider the ensemble mean of model simulations to be totally right (i.e., more accurate). By contrast, simulation from each ESM is treated as one sample, and all samples are clustered together to identify an overall functional relationship across ESMs (i.e., f). Furthermore, the core concept of emergent constraint also determines that it is not possible to develop such constraint framework based on X alone. Therefore, we do not restrict the analysis to the sole use of variable X , but rather we perform constraint analysis based on the strong inter-model relationship between X and Y (i.e., f).

In the emergent constraint framework presented in this study, the variable Y corresponds to the trend in ET induced by historical LULCC or future afforestation (i.e., $\delta\text{ET}^{\text{LULCC}}$ and $\delta\text{ET}^{\text{AFF}}$, respectively), for which no direct observations exist; and X corresponds to the natural logarithm value of transpiration-specific Bowen ratio ($\ln(\text{B}_{\text{ts}})$), for which instead observational estimates are available. In this study, B_{ts} is used as X in emergent constraint framework instead of Tr that is mentioned in the reviewer’s conclusive comments, because there is a stronger inter-model correlation between B_{ts} and the target variable Y (i.e., $\delta\text{ET}^{\text{LULCC}}$ and $\delta\text{ET}^{\text{AFF}}$). It suggests that model spread in estimates of $\delta\text{ET}^{\text{LULCC}}$ and $\delta\text{ET}^{\text{AFF}}$ may be more closely related to the difference in model representation of B_{ts} , instead of that of single Tr . Here B_{ts} is transformed to its natural logarithm (1) to better reflect the inherently non-linear inter-model relationship (detailed explanations are provided in our response to Comment #13), and (2) to meet the linear assumption of emergent constraints (Hall et al., 2019). We find a strong and significant inter-model relationship between $\ln(\text{B}_{\text{ts}})$ and $\delta\text{ET}^{\text{LULCC}}$ ($r=0.931$, $p<0.01$), and between $\ln(\text{B}_{\text{ts}})$ and $\delta\text{ET}^{\text{AFF}}$ ($r=-0.878$, $p<0.01$) at the global scale, which is fundamental for the applicability of the emergent constraint framework. Results derived from one-thousand random sampling further enhance the robustness of the methods and increase the confidence of such significant inter-model relationships (Supplementary Fig. 2a-d). Moreover, additional experiments have been performed to test the validity of our results by focusing on the subsets from ESM ensemble. Each subset leaves one specific ESM out of the analysis of the emergent relationship. Results focusing on $\delta\text{ET}^{\text{LULCC}}$ and $\delta\text{ET}^{\text{AFF}}$ both show that their constrained values derived in all experiments distribute in a narrow range, and they are generally close to that obtained from the full ESM ensemble (Supplementary Tables 4 and 8). Such high consistency between results from these additional experiments performed on reduced ensemble members and those derived based on the

full ESM ensemble further demonstrate that our results are robust and not conditioned on the models selected for analysis. Given such insensitivity on specific ESM and to maximum the sample size used for emergent relationships, we therefore prefer to perform analysis on all ESMs rather than only on ESMs that seem to be right.

Furthermore, a comprehensive set of experiments is additionally performed (details in Discussion), to demonstrate that the emergent relationship revealed here is not only with the strong statistical underpinnings, but also is accompanied by a strong mechanistic interpretation. Results show that the large model spread in estimates of effect of historical LULCC (generally characterized as forest-cropland conversion) on ET originates from the ESM misrepresentation of the difference in surface energy partitioning (particularly latent heat partitioned into transpiration) between forests and croplands, which further helps explain the underlying mechanism of correcting modelled LULCC effect by introducing observed energy partitioning (Fig. 6).

Action taken: To improve clarity about the methodology, we have added more details about the emergent constraint framework, particularly its rationale in the Methods section (see lines 774-789). Furthermore, we have added the new Supplementary Table 8 in the revised version.

Supplementary Table 8. Constrained projection of future afforestation impact on terrestrial water cycle after excluding the specific model. For each sensitivity experiment by leaving specific model out of the emergent relationship, the associated inter-model correlation (r) between the effect of future (2015-2099) on annual evapotranspiration (δET^{AFF}) and the historical (1982-2014) natural logarithm value of global averaged transpiration-specific Bowen ratio ($\ln(B_{ts})$), and that between δET^{AFF} and the effect of future (2015-2099) on annual precipitation (δP^{AFF}), and the resulting constrained estimate of δET^{AFF} , δP^{AFF} , and the ultimate effect on terrestrial water availability ($\delta WA^{AFF} = \delta P^{AFF} - \delta ET^{AFF}$, see Eq. (11)) are provided below. Constrained values of δET^{AFF} and δP^{AFF} are expressed as mean \pm standard deviation.

Model left out	$r(\ln(B_{ts}), \delta ET^{AFF})$	$r(\delta ET^{AFF}, \delta P^{AFF})$	Constrained value (mm yr ⁻²)		
			δET^{AFF}	δP^{AFF}	δWA^{AFF}
BCC-CSM2-MR	-0.854 ($p < 0.01$)	0.712 ($p < 0.05$)	0.052 \pm 0.033	0.022 \pm 0.046	-0.030
CanESM5	-0.845 ($p < 0.01$)	0.622 ($p < 0.10$)	0.052 \pm 0.025	0.024 \pm 0.035	-0.028
CESM2	-0.889 ($p < 0.01$)	0.709 ($p < 0.05$)	0.050 \pm 0.033	0.023 \pm 0.041	-0.027
CMCC-ESM2	-0.890 ($p < 0.01$)	0.708 ($p < 0.05$)	0.055 \pm 0.033	0.023 \pm 0.040	-0.032
GFDL-ESM4	-0.879 ($p < 0.01$)	0.747 ($p < 0.05$)	0.053 \pm 0.033	0.035 \pm 0.044	-0.018
IPSL-CM6A-LR	-0.887 ($p < 0.01$)	0.788 ($p < 0.05$)	0.054 \pm 0.033	0.039 \pm 0.039	-0.016
MIROC-ES2L	-0.861 ($p < 0.01$)	0.707 ($p < 0.05$)	0.047 \pm 0.032	0.024 \pm 0.043	-0.024
NorESM2-LM	-0.929 ($p < 0.01$)	0.683 ($p < 0.10$)	0.048 \pm 0.029	0.013 \pm 0.034	-0.034
UKESM1-0-LL	-0.866 ($p < 0.01$)	0.723 ($p < 0.05$)	0.051 \pm 0.033	0.005 \pm 0.032	-0.045

8. Lines 114-116: There are of course so many reasons why these land use datasets are imperfect... They do not handle density and mass of biomass well (usually only area coverage, and perhaps now height), and most of these models are based on an extraordinarily small sets of vegetation types. Think about how many different

tree species there are out there in the world, all with different rates of ET production... How well do these datasets represent this degree of variation...? It is not just differences between trees, crops and grasses that matter, but rather all vegetation types and the relative density of vegetation cover (by this I do NOT mean % forest cover in an individual pixel, but actual density of vegetation coverage) ...

Response and action taken: We agree with the reviewer that current land use datasets only contain the vegetation area coverage and height information, and do not explicitly describe the density and biomass of different vegetation types. Simplification and aggregation of numerous vegetation types within these models may result in some inconsistency across models and between individual model and observed patterns in terms of cover fraction of PFTs. We have added associated contents in the Methods section (see lines 635-637).

9. Lines 122-123: Shouldn't the LULCC also be characterized here...? Are you arguing there is less or more tree, forest and vegetation cover...??? Some of this comes up later in the text. But it would be helpful here to better understand the directionality of these relationships...?

Response and action taken: To improve clarity, we have added the description about the characteristics of LULCC during the period 1982-2014 in the main text. Details are as follows: “According to an ensemble of historical simulations from twelve CMIP6 models, the LULCC, that is generally characterized by the conversion from forests to croplands occurring during the period 1982-2014, has increased terrestrial ET globally (δET^{LULCC}) by an average rate of 0.057 ± 0.155 mm yr⁻² (Fig. 1b).” (see lines 119-122).

10. Line 133: Add “The”

Response and action taken: We have added “the” in this sentence (see line 134).

11. Lines 139-141: Can this be taken further... Why doesn't more of the analysis address this latitudinal variation...? And how is it explained...?

Response: We agree with the reviewer that a more latitudinal analysis, or perhaps a regional-based analysis could be informative to evaluate results of the emergent constraint framework at finer scale. To this end, in the revised version of our manuscript, we have performed a series of new experiments over forty-four reference regions used in the Intergovernmental Panel on Climate Change Sixth Assessment Report (IPCC AR6), to provide specific constraints at regional level. Original IPCC AR6 reference regions include forty-six land regions and fifteen ocean regions, each with consistent regional climatic features (Iturbide et al., 2020). Here, all ocean regions and two land regions located in Antarctica are excluded from our analyses, since we focus exclusively on the vegetated land (Supplementary Fig. 7). To calculate the emergent constraints at the

regional scale, we first extract the gridded datasets from observation-based products and model outputs for each reference region based on the map provided by Iturbide et al. (2020). We then estimate the regional averaged δET^{LULCC} and δET^{AFF} for each model (Eqs. (1)-(3)), the regional mean levels of observed and modelled B_{ts} (Eq. (4)) and further $\ln(B_{ts})$. Consistent with our global-scale aggregated metrics, these regional-scale estimates are also obtained by spatial average weighting each grid-cell value within the region based on its land area.

Unconstrained results based on the original simulations from CMIP6 model ensemble generated for the historical period suggest that the LULCC-related signal is much stronger at the regional scale (magnitude ranging from -1.376 ± 1.480 mm yr⁻² to 0.886 ± 0.967 mm yr⁻² across reference regions, Fig. 4 and Supplementary Table 9) compared to the global mean level (0.057 ± 0.155 mm yr⁻², Fig. 1b). This finding is in line well with the reviewer's intuition. Over 26.0% of global vegetated land, LULCC is believed to dominate historical change in local ET (relative contribution higher than 50%), according to the original CMIP6 model simulations (Supplementary Table 9). Similarly, more pronounced signals emerge at regional scale under the future afforestation scenario. For example, compared to the global mean level (0.024 ± 0.048 mm yr⁻², Fig. 3d), unconstrained projections suggest a much higher δET^{AFF} in Central Africa (CAF) whose magnitude reaches 0.420 ± 0.515 mm yr⁻², indicating that afforestation could explain 74.1% of future change in local ET (CAF in Fig. 5a and Supplementary Table 10). These results highlight the key relevance of land use and land cover change in long-term variations of the regional water cycle.

We find that applying the emergent constraint always reduces the uncertainty in estimates of ET response (i.e., δET^{LULCC} and δET^{AFF}), although the degree of reduction varies across regions (Figs. 4 and 5a and Supplementary Tables 9 and 11). The inter-model correlation between δET^{LULCC} and $\ln(B_{ts})$ is statistically ($p < 0.05$) significant for twenty-one reference regions corresponding to 50.2% of the global vegetated land (hexagons with the diagonal lines in Fig. 4). Slightly more widespread patterns are found for the interplay between δET^{AFF} and $\ln(B_{ts})$ with twenty-five reference regions – covering 55.5% of the global vegetated land – showing statistically significant ($p < 0.05$) correlations (hexagons with the diagonal lines in Fig. 5a). These regions exhibit an average reduction in standard deviation after constraint by 21.7% and 24.9% during the historical and future scenario periods, respectively, more than 10-fold greater than the analogous estimates obtained for the remaining regions without significant correlations (1.8% and 1.3%) (Supplementary Tables 9 and 11).

Correction direction for estimates of δET^{LULCC} varies spatially, primarily because of the distinct types of land cover conversion occurring across regions during the period 1982-2014 and of their interplay with the specific background climate of the region. Over 50.2% of global vegetated land where the emergent relationship holds, areas characterized by an increase in forest area (17.0% of global vegetated land), generally shows a higher δET^{LULCC} after applying the emergent constraint compared to the unconstrained estimate, which is opposite to the global mean results (Fig. 4). By contrast, more regions have experienced a decrease

in forest area, and generally show a constrained estimate of δET^{LULCC} lower than that one obtained from the unconstrained outputs, consistently with the global mean results (Figs. 2 and 4). In this respect, δET^{LULCC} over 13.2% of global land turns from positive to negative after applying the emergent constraint (Fig. 4 and Supplementary Table 9).

Under the simulated future afforestation scenario, conversions from croplands to forests are expected to occur in most regions (Supplementary Table 10, exception cases are explained in Methods), generally leading to a higher constrained estimate of δET^{AFF} than the analogous metric derived from the original model ensemble (Fig. 5a). The constrained value of δET^{AFF} differs greatly across afforested regions, ranging from -0.158 ± 0.357 mm yr⁻² to 0.447 ± 0.171 mm yr⁻² (Fig. 5a). Occurrence of such spatial difference in δET^{AFF} may arise from (1) the varying magnitude of increase in forest cover (Supplementary Table 10), and (2) the divergent hydrological sensitivity to forest gain across regions which is largely controlled by the local climate and landscape conditions (Zhang et al., 2017). Previous evidence revealed that ET is generally more sensitive to forest cover change in drier regions, highlighting the importance of the ability to access deeper soil moisture in water-limited environments (Ning et al., 2020). Consistent with the global mean level (0.051 ± 0.031 mm yr⁻², Fig. 3d), constrained δET^{AFF} is positive over most (58.8%) part of the afforested land, suggesting the widespread positive effect of afforestation on ET as reported in previous studies (Teuling et al., 2019; Yang et al., 2023). On the other hand, the negative value of constrained δET^{AFF} still exists in some regions after constraint, possibly because the shading by increased foliage cover could decrease the amount of radiation reaching the ground and further lead to the reduction in soil evaporation, which counteracts the increase in Tr due to afforestation (Ukkola et al., 2016).

For twenty-two regions (covering 51.1% of global vegetated land) where emergent relationship and correlation between δET^{AFF} and δP^{AFF} across ESMs are both statistically significant, local δP^{AFF} is of comparable magnitude to δET^{AFF} after constraint, particularly in tropics and subtropics (Fig. 5a,b). For example, δP^{AFF} in North Eastern Africa (NEAF) increases from 0.008 ± 0.659 mm yr⁻² to 0.352 ± 0.470 mm yr⁻², along with the increase in δET^{AFF} from 0.091 ± 0.354 mm yr⁻² to 0.312 ± 0.253 mm yr⁻² after applying the emergent constraint (Supplementary Table 10). We estimate that sixteen out of twenty-two reference regions which cover 32.9% of global vegetated land, exhibit an increase in δWA^{AFF} after constraint, corresponding to a weakened negative effect of afforestation on WA and even a transition to positive effect in the constrained projection (Fig. 5c and Supplementary Table 10). Such weakening of negative afforestation effect on WA after constraint is particularly evident in Central and South America, Sub-Saharan Africa and South Asia, where constrained δP^{AFF} is substantially higher than unconstrained one (Fig. 5b,c). The existence of regions with higher magnitude of positive δP^{AFF} than that of δET^{AFF} (i.e., constrained $\delta WA^{AFF} > 0$) suggests that the increase in ET associated with afforestation could be outweighed by more pronounced moisture recycling in these regions. Such patterns emerge in regions such as NEAF, Western Africa (WAF), South Asia (SAS), and Eastern Australia (EAU) (Fig. 5), and appear broadly consistent with observational evidence documenting the

buffering effect of forest-climate feedback on local water depletion (O'Connor et al., 2021; Hoek van Dijke et al., 2022).

Overall, these results demonstrate that original ESM outputs tend to misrepresent the direction and magnitude of historical LULCC effect on regional ET (Fig. 4), and more importantly, overestimate the water scarcity induced by afforestation scenario, particularly in tropics and subtropics (Fig. 5c). These results ultimately support the value of bias-corrected simulations for a more robust assessment of land-based climate mitigation plans at the regional level.

Action taken: We have added the aforementioned results focusing on IPCC AR6 reference regions in the Results section (“Emergent constraints at the regional scale”) (see lines 261-357), and correspondingly, added the new Figs. 4 and 5 and Supplementary Fig. 7 and Tables 9-11 in the revised version. Meanwhile, we have added related discussion on findings obtained at the regional scale (see lines 389-393, 398-415, 472-484), and described the methodological details related to the experiments performed at the regional scale in the Methods section (“Spatial domain”) (see lines 700-718). To keep concise, Supplementary Tables 9-11 are not presented below.

Fig. 4 | Constrained effect of historical land use and land cover change on terrestrial evapotranspiration over IPCC AR6 reference regions. Within each hexagon, the filled color in the upper and lower parts represent the unconstrained estimate from the original CMIP6 model ensemble and the observationally constrained estimate, respectively. The size of the green dot indicates the relative reduction in standard deviation (RR_{σ}) after applying the hierarchical emergent constraint approach. The hexagons with the diagonal lines indicate that the emergent relationship is statistically significant at the 90% confidence level. In fact, for these hexagons, their emergent relationship's significance even passes the 95% confidence level. Specific values for making this plot are listed in Supplementary Table 9. Moreover, detailed information of these reference regions is provided in Supplementary Fig. 7.

a. δET^{AFF}

b. δP^{AFF}

c. δWA^{AFF}

Fig. 5 | Constrained projection of future afforestation impact on terrestrial water cycle over IPCC AR6 reference regions. (a) Constrained trend in annual evapotranspiration during the period 2015-2099 (δET^{AFF}), as derived under the future afforestation scenario (Methods). Within each hexagon, the filled color in the upper and lower parts represent the unconstrained estimate of δET^{AFF} from the original CMIP6 model ensemble and the observationally constrained estimate, respectively. The size of the green dot indicates the relative reduction in standard deviation (RR_{σ}) after applying the hierarchical emergent constraint approach. The hexagons with the diagonal lines indicate that the emergent relationship is statistically significant at the 90% confidence level. In fact, for these hexagons, their emergent relationship's significance even passes the 95% confidence level. **(b and c)** Same as **(a)**, but for the trends in annual precipitation (δP^{AFF}) and terrestrial water availability (δWA^{AFF}). The hexagons with the diagonal lines indicate that both the emergent relationship and the inter-model relationship between δET^{AFF} and δP^{AFF} are statistically significant at the 90% confidence level. Specific values for making these plots are listed in Supplementary Tables 10 and 11. Moreover, detailed information of these reference regions is provided in Supplementary Fig. 7.

Acronym	Full name	Acronym	Full name	Acronym	Full name
GIC	Greenland/Iceland	SAH	Sahara	CAU	Central Australia
NWN	Northwestern North America	WAF	Western Africa	EAU	Eastern Australia
NEN	Northeastern North America	CAF	Central Africa	SAU	Southern Australia
WNA	Western North America	NEAF	North Eastern Africa	NZ	New Zealand
CNA	Central North America	SEAF	South Eastern Africa	EAN	Eastern Antarctica
ENA	Eastern North America	WSAF	West Southern Africa	WAN	Western Antarctica
NCA	Northern Central America	ESAF	East Southern Africa	ARO	Arctic Ocean
SCA	Southern Central America	MDG	Madagascar	NPO	North Pacific Ocean
CAR	Caribbean	RAR	Russian Arctic	EPO	Equatorial Pacific Ocean
NWS	Northwestern South America	WSB	West Siberia	SPO	South Pacific Ocean
NSA	Northern South America	ESB	East Siberia	NAO	North Atlantic Ocean
NES	Northeastern South America	RFE	Russian Far East	EAO	Equatorial Atlantic Ocean
SAM	South American Monsoon	WCA	West Central Asia	SAO	South Atlantic Ocean
SWS	Southwestern South America	ECA	East Central Asia	ARS	Arabian Sea
SES	Southeastern South America	TIB	Tibetan Plateau	BOB	Bay of Bengal
SSA	Southern South America	EAS	East Asia	EIO	Equatorial Indian Ocean
NEU	Northern Europe	ARP	Arabian Peninsula	SIO	South Indian Ocean
WCE	Western and Central Europe	SAS	South Asia	SOO	Southern Ocean
EEU	Eastern Europe	SEA	South East Asia		
MED	Mediterranean	NAU	Northern Australia		

Supplementary Fig. 7. IPCC AR6 WGI reference regions. Full names corresponding to acronyms shown in figure are illustrated in the following table. Non-vegetated areas, defined as multi-year (1982-2014) average leaf area index (LAI) $0.15 \text{ m}^2 \text{ m}^{-2}$, are excluded in our analysis and are shown in grey. It should be noted that due to the limited number (nearly zero) of valid grid-cells, two polar regions (i.e., EAN and WAN) and twelve ocean regions (i.e., ARO, NPO, EPO, SPO, NAO, EAO, SAO, ARS, BOB, EIO, SIO, and SOO) are not considered in our regional-scale analysis. Original figure and associated files are provided in Iturbide et al.¹⁰.

12. Line 145: replace “univocally” with “unequivocally”

Response and action taken: We have modified the text according to the reviewer’s suggestion (see line 146).

13. Lines 165-167: This general relationship could be better explored here so that the reader has an easier time interpreting the relationships between these two variables... The discussion here seems inadequate...

Response: The non-linearity of the inter-model relationship between δET^{LULCC} and B_{ts} is identified based on the fact that natural logarithm regression has a better performance in fitting the inter-model relationship between δET^{LULCC} and B_{ts} over 1982-2014 compared with the linear regression, as reflected by a higher determination coefficient (r^2) for the former fitting function (Supplementary Fig. 3a). This regression model shows the p -value lower than 0.01, and therefore its result is statistically significant.

To further test whether the relationship between B_{ts} and δET^{LULCC} is inherently non-linear across models, we perform a statistical analysis by applying a random sampling technique with one-thousand replications. We therefore generate one-thousand data sets by randomly extracting samples from 60% of vegetated grid-cells globally without replacement. Each data set includes twelve δET^{LULCC} estimates and twelve associated B_{ts} values, each pair of which corresponds to one CMIP6 ESM. Subsequently, we fit linear, natural logarithmic, and quadratic regressions to the inter-model relationship between δET^{LULCC} and B_{ts} for each data set, and use r^2 and root mean squared error ($RMSE$) to identify the regression model that provides the best fit in each data set. We find that mean r^2 of natural logarithmic regression fitting reaches 0.878 and is higher than that of linear and quadratic ones (0.851 and 0.876) (Supplementary Fig. 4a). In addition, mean $RMSE$ of natural logarithmic regression fitting is 0.061 mm yr⁻², which is lower when compared to the fitting by linear and quadratic regressions (0.067 mm yr⁻² and 0.063 mm yr⁻²) (Supplementary Fig. 4b). As natural logarithmic regression has the best performance in fitting the inter-model relationship between B_{ts} and δET^{LULCC} , $\ln(B_{ts})$ is used in our emergent constraint framework as reference.

To further increase the confidence in our approach, we also compute and compare the inter-model correlation between δET^{LULCC} and $\ln(B_{ts})$, against that between δET^{LULCC} and B_{ts} , during one-thousand replications of random sampling similar to that described above. The Pearson's correlation coefficient between δET^{LULCC} and $\ln(B_{ts})$ derived based on these one-thousand data sets reaches 0.936 ± 0.005 , higher than that one between δET^{LULCC} and B_{ts} (0.922 ± 0.006) (Supplementary Fig. 2a). Similarly, δET^{AFF} also exhibits a stronger correlation with $\ln(B_{ts})$ across models compared to that one obtained with B_{ts} (-0.878 ± 0.006 vs. -0.866 ± 0.006) (Supplementary Fig. 2c). Given the proportional relation between the correlation strength and the performance of emergent constraint, these additional analyses further confirm the utility of applying a log-transformation to B_{ts} in emergent relationship.

Altogether, these additional experiments demonstrate the robustness of our methodology and clarify the utility of applying log-transformation to B_{ts} to better capture the inherent relationship between LULCC effect and surface energy partitioning across models.

Action taken: In the revised version, we have added these contents into the new Supplementary Text 2 (“Emergent constraint results derived based on B_{ts} ” section) to keep the main text reasonably concise, and briefly recalled them in the main text (see lines 815-817). Correspondingly, we have added the new Supplementary Figs. 2 and 4.

Supplementary Fig. 2. Inter-model correlation and associated significance level for one-thousand random sampling. (a) Frequency distribution of one-thousand sets of inter-model correlation (r) between the effect of historical land use and land cover change (generally characterized as forest-cropland conversion) on annual evapotranspiration (δET^{LULCC}) and the global averaged transpiration-specific Bowen ratio (B_{ts}), as well as r between δET^{LULCC} and natural logarithm value of B_{ts} (i.e., $\ln(B_{ts})$) during the period 1982-2014. Each data set is generated by randomly extracting samples from 60% of vegetated grid-cells globally without replacement. Each set includes twelve δET^{LULCC} estimates and twelve associated B_{ts} values, each pair of which corresponds to one CMIP6 ESM. Distribution averages are shown as dashed horizontal lines. (b) Same as (a), but for the p -value associated to the inter-model correlation coefficient between δET^{LULCC} and B_{ts} , and that between δET^{LULCC} and $\ln(B_{ts})$. (c and d), Same as (a and b), but for r with the effect of future (2015-2099) afforestation on annual evapotranspiration (δET^{AFF}), and the associated p -value. During one-thousand random sampling procedure, each set includes nine δET^{AFF} estimates and nine associated B_{ts} values, each pair of which corresponds to one CMIP6 ESM. (e and f), Same as (a and b), but for r between δET^{AFF} and the modelled effect of future (2015-2099) afforestation on annual precipitation (δP^{AFF}), and the associated p -value.

Supplementary Fig. 4. Performance comparison of different regression model types in fitting the inter-model relationship for one-thousand random sampling. (a) Boxplot of one-thousand sets of determination coefficient (r^2) of linear, natural logarithmic, and quadratic regression models in fitting the inter-model relationship between the effect of land use and land cover change on annual evapotranspiration (δET^{LULCC}) and the global averaged transpiration-specific Bowen ratio (B_{ts}) during the period 1982-2014. Each data set is generated by randomly extracting samples from 60% of vegetated grid-cells globally without replacement. Each set includes twelve δET^{LULCC} estimates and twelve associated B_{ts} values, each pair of which corresponds to one CMIP6 ESM. Boxplot elements: box = values of 25th and 75th percentiles; horizontal line = median; rectangle = mean; whiskers = values of 10th and 90th percentiles. (b) Same as (a), but for root mean square error (RMSE).

14. Lines 171-173: relative to what...? I guess, to “the original CMIP6 ensemble mean”. Please improve the quality of the description...

Response and action taken: To improve the clarity, we have revised this sentence. Details are as follow: “Emergent constraint results show that historical LULCC has led to a decrease in ET with a global average rate of $-0.031 \text{ mm yr}^{-2}$ during the period 1982-2014, which is opposite in sign to the original CMIP6 ensemble mean (0.057 mm yr^{-2}) (Fig. 2b).” (see lines 178-180).

15. Lines 175-176: More clearly characterizing change in your independent variable (LULCC) will greatly help improve clarity on the directionality of what you are trying to illustrate and argue, throughout...

Response and action taken: We have added the description of characteristics of historical LULCC in this sentence to improve clarity. Details as follows: “After applying the emergent constraint, the transition from

positive to negative δET^{LULCC} globally during the past three decades results from the combination of (1) the positive relationship between the effect of LULCC (characterized as forest-cropland conversion globally) and $\ln(B_{ts})$ and (2) the overall overestimation of B_{ts} in current-version ESMs (Fig. 2a and Supplementary Fig. 5).” (see lines 180-185).

Meanwhile, we have further clarified the description of results and interpretation related to the issue raised by the reviewer to help the readers easier to follow (please, see also our response to your previous comments).

16. Line 203: delete “the”

Response and action taken: We have deleted “the” in this sentence (see line 206).

17. Lines 215-217: This is illogical. If forests do not evapotranspire, they cannot grow. Perhaps this should be rephrased. Also, why the strong distinction between δET^{AFF} , and that for LULCC? Isn't change in AFF just a share of change in LULCC?

Response: The negative inter-model relationship between δET^{AFF} and $\ln(B_{ts})$ revealed here refers to the fact that ESMs with low $\ln(B_{ts})$ values generally have a large ET increase induced by future afforestation, while ESMs with high $\ln(B_{ts})$ values show a weak increase or even decrease in ET under the same afforestation scenario. Global land has experienced massive deforestation during the past decades, and LULCC over 1982-2014 is generally characterized as a conversion from forests to croplands at the global extent (Fig. 1a). The opposite in direction of land cover conversion during the historical period (forest-cropland conversion) and that under the future afforestation scenario (cropland-forest conversion) explains why (1) effect of historical LULCC on global ET (i.e., δET^{LULCC}) is negative, while effect of future afforestation on global ET (i.e., δET^{AFF}) is positive; and furthermore, (2) the inter-model relationship between δET^{LULCC} and $\ln(B_{ts})$ is positive, while inter-model relationship between δET^{AFF} and $\ln(B_{ts})$ is negative. It should be noted that “AFF” and “LULCC” mentioned here do not have a subordinate relationship. We use “LULCC” and “AFF” to represent the land use change during the historical period (1982-2014) and the future afforestation scenario period (2015-2099), respectively. Terms with superscripts “LULCC” and “AFF” correspond to results focusing on the historical period and the future scenario period, respectively.

Action taken: In the revised version, we have improved the explanation of the so-called “AFF” runs (see lines 764-766), to clarify that they specifically refer to the land use change under the future afforestation scenario, instead of a share of “LULCC”. We have also better framed the context to help the readers easier to understand the underlying meaning of the negative inter-model relationship between δET^{AFF} and $\ln(B_{ts})$. Details are as follows: “ESMs with low $\ln(B_{ts})$ values generally have a large ET increase induced by future

afforestation, while ESMs with high $\ln(B_{ts})$ values show a weak increase or even decrease in ET under the same afforestation scenario.” (see lines 226-228). As mentioned above, the opposite in direction of land cover conversion during the historical period and under the future scenario period leads to the distinction in sign of δET^{LULCC} and δET^{AFF} , and also the opposite in their relationship with $\ln(B_{ts})$. We therefore believe that the addition of the description about the characteristics of historical LULCC (i.e., conversion from forests to croplands in general) throughout the text, could further highlight such opposite in direction of land cover conversion between two periods, which also helps the readers to better understand and contextualize our results.

18. Lines 287-288: “Global LULCC occurred during the last three decades have been...” awkward/incomprehensive... revise...

Response and action taken: According to the reviewer’s suggestion, we have modified this sentence to improve its clarity. Details are as follows: “LULCC occurred over the past three decades are prominently characterized by a widespread conversion from forest to cropland patterns (Fig. 1a).” (see lines 365-367).

19. Lines 288-289: I tend to think that some of this should have been discussed above, in order to make the numbers more easily understandable and interpretable...

Response and action taken: According to the reviewer’s suggestion, we have added the description of characteristics of historical LULCC, i.e., the conversion from forests to croplands globally, in earlier parts of the Results section, to help the readers easier to understand the work.

20. Lines 311-315: The discussion that occurs here at the end of this paragraph is helpful for understanding the basic parameters of the article. But it comes very late in the discussion... Characterizing LULCC in this way helps to better understand the findings regarding directionality. Thus, I would recommend bring in these basic parameters much earlier in the text... It will make it significantly easier to understand and follow the basic findings...

Response and action taken: We agree with the reviewer that a description of historical LULCC characteristics in earlier parts of the manuscript would help the readers to better understand the findings regarding directionality (e.g., the sign of δET^{LULCC}). We have moved associated contents occurring in the Discussion section to the front (see lines 187-193), and meanwhile, we have recalled the characteristics of historical LULCC in different parts of the Results section (see lines 110-111, 120-121, 183-184), to improve the overall clarity of the work.

21. Lines 315-319: This also suggests that there is an important and very strong latitudinal component to this analysis that is actually not assessed...??? I searched the text for the word "latitude" and noticed that it only came up once... The seems to be a big gap in the analysis. Shouldn't these general relationships be very strongly latitudinally determined...???

Response and action taken: As illustrated previously, we have substantially reshaped the manuscript to focus more on the regional scale by performing a series of new experiments over forty-four IPCC AR6 reference regions. We prefer to follow the regional scale analysis than the analysis of latitudinal patterns because compensation along latitudinal belts could hide important regional signals. Results show the strong emergent relationship and the associated good correction performance particularly in tropics and subtropics. We believe that our response and actions provided to the previous comment (i.e., Specific Comment #11) have already addressed this specific issue.

References:

1. Bonan, G. & Doney, S. Climate, ecosystems, and planetary futures: The challenge to predict life in Earth system models. *Science* **359**, eaam8328 (2018).
2. Clark, M. et al. Improving the representation of hydrologic processes in Earth System Models. *Water Resour. Res.* **51**, 5929-5956 (2015).
3. Ellison, D., Futter, M. & Bishop, K. On the forest cover–water yield debate: from demand- to supply-side thinking. *Glob. Change Biol.* **18**, 797-1196 (2012).
4. Ellison, D. et al. Trees, forests and water: Cool insights for a hot world. *Glob. Environ. Change* **43**, 51-61 (2017).
5. Hall, A., Cox, P., Huntingford, C. & Klein, S. Progressing emergent constraints on future climate change. *Nat. Clim. Change* **9**, 269-278 (2019).
6. Hoek van Dijke, A. et al. Shifts in regional water availability due to global tree restoration. *Nat. Geosci.* **15**, 363-368 (2022).
7. Iturbide, M. et al. An update of IPCC climate reference regions for subcontinental analysis of climate model data: definition and aggregated datasets. *Earth Syst. Sci. Data* **12**, 2959-2970 (2020).
8. Keenan, T. et al. A constraint on historic growth in global photosynthesis due to rising CO₂. *Nat. Clim. Change* **13**, 1376-1381 (2023).
9. McDowell, N. et al. Pervasive shifts in forest dynamics in a changing world. *Science* **368**, 964 (2020).
10. Ning, T. et al. Effects of forest cover change on catchment evapotranspiration variation in China. *Hydrol. Process.* **34**, 2219-2228 (2020).
11. O’Conner, J. et al. Forests buffer against variations in precipitation. *Glob. Change Biol.* **27**, 4686-4696 (2021).
12. Sterling, S., Ducharne, A. & Polcher, J. The impact of global land-cover change on the terrestrial water cycle. *Nat. Clim. Change* **3**, 385-390 (2013).
13. Tang, T., Ge, J., Cao, J. & Shi, H. Land water availability altered by historical land use and land cover change. *npj Clim. Atmos. Sci.* **8**, 230 (2025).
14. Teuling, A. et al. Climate change, reforestation/afforestation, and urbanization impacts on evapotranspiration and streamflow in Europe. *Hydrol. Earth Syst. Sci.* **23**, 3631-3652 (2019).
15. Ukkola, A. et al. Reduced streamflow in water-stressed climates consistent with CO₂ effects on vegetation. *Nat. Clim. Change* **6**, 75-78 (2016).
16. Yang, Y. et al. Evapotranspiration on a greening Earth. *Nat. Rev. Earth. Environ.* **4**, 626-641 (2023).
17. Zhang, M. et al. A global review on hydrological responses to forest change across multiple spatial scales: Importance of scale, climate, forest type and hydrological regime. *J. Hydrol.* **546**, 44-59 (2017).

First, we would like to thank the two reviewers for their insightful and constructive comments. We tried to address in this revision their remaining comments and suggestions to improve the robustness of the analysis and the quality of the text.

In the following, we respond to each reviewer's comment by referring to line numbers of the revised tracked version, when not differently indicated. References cited along with this response letter are reported at the bottom of the document.

Reviewer #1: Questions and our responses

I thank the authors for the in-depth and comprehensive revisions, and for addressing all my major and minor concerns. Especially, the inclusion of the regional assessment in the form of to-the-point visuals makes this study considerably stronger and clearer.

Overall, I still think that the global constraint the authors place at the centre of the manuscript is a) IMO not correctly interpreted and b) does not add much to our understanding of the global effects on the hydrological cycle induced by LULCC. In their rebuttal, the authors claim that I did not understand the analysis; however, they repeat the same rationale as in the manuscript: To demonstrate that the effect is non-negligible, they present the numbers as a relative change rather than an absolute change. As I wrote in my first review report, one can put small numbers in relation to each other so they appear higher; this is misleading. Your global constraint is $0.031 \pm 0.081 \text{ mm yr}^{-2}$; again, a) this is a very small number in the global hydrological cycle and it does not become more impactful when expressed as a relative contribution to the overall ET trend, and b) the interpretation that your constraint is qualitatively different from the CMIP6 mean is a huge over-interpretation. Yes, it switches the sign in the mean, but your uncertainty estimate is considerably larger than the effect size. Is the sign switch still present if you use a bootstrap approach when deriving the emergent relationship in the model ensemble (randomly sample models, conduct your EC analysis, and test whether the sign switch is robust, e.g., EC in Winkler et al., 2019)? Returning to the small effect size, looking at your PDFs in Figure 2, my interpretation is that neither the unconstrained nor the constrained estimate is robustly non-zero (if you leave out GISS-E2-1-G in your multi-model mean, the unconstrained mean estimate is likely spot-on zero).

Thus, to back your central claim — “The constraint reverses the sign of the original ESM estimates obtained over the observational period ($0.057 \pm 0.155 \text{ mm yr}^{-2}$, mean \pm s.d.) and narrows the inter-model spread, ultimately leading to a more robust global estimate of the LULCC-driven ET change of $-0.031 \pm 0.081 \text{ mm yr}^{-2}$ ” — you must a) test that the sign switch is robust, and b) show that your effect is statistically different from zero.

Bottom line, I stick to my recommendation to rearrange the manuscript to focus on your new analysis on hot-spots of land-use and land-cover change (LULCC) and ET responses, and give the different regional ET

responses the spotlight, as this is where these water-cycle responses matter. Therefore, the manuscript should be revised accordingly before being considered for publication in Nature Communications.

We thank the reviewer for her/his comments. We would like to stress that the emergent constraint approach applied in Winkler et al. (2019), which is typically referred as the classic emergent constraint originally proposed by Cox et al. (2013), presents some key differences from the hierarchical emergent constraint approach applied in our study. This classic emergent constraint approach involves convolving the prediction error implied by the regression fit of the scatter plot to the inter-model relationship between x and y , with the observational uncertainty in x , to produce a constrained distribution for the y -axis variable (Cox et al., 2018). The prediction error in classic emergent constraint framework is reflected by the probability contours around the best-fit straight line, which are derived from bootstrap resampling of the model pairs (i.e., (x_i, y_i) , where i indexes each model). In other words, the classic emergent constraint tempts to simply use the regression line itself to project the observational estimate of y , and the prediction error for fit can only be approximated empirically via bootstrap. In contrast, the hierarchical emergent constraint relates model-simulated x and y , and observations of x through conditional-probability distributions derived from Bayes' theorem, with all sources of uncertainty explicitly represented as probability distributions (Bowman et al., 2018). The prediction error therefore can be analytically derived from the posterior distribution within this framework, avoiding the need for approximation by bootstrap resampling. In this respect, there is no necessity to use a bootstrap approach when deriving the emergent relationship within the hierarchical emergent constraint framework.

To further assess the validity of our approach and better compare its robustness with methods suggested by Rev. 1, we have performed a set of additional experiments based on the classic emergent constraint approach (Cox et al., 2013; Winkler et al., 2019). The obtained estimates are then compared against analogous values based on the hierarchical emergent constraint approach currently shown in the text. Such assessment allows to build confidence on the robustness of our results about the sign switch of historical land use and land cover changes effect on global evapotranspiration (δET^{LULCC}) after constraint. Following the approach reported in Winkler et al. (2019), we have applied bootstrapping to estimate the 68% confidence of the emergent linear relationship between δET^{LULCC} and natural logarithm of transpiration-specific Bowen ratio ($\ln(B_{ts})$) across CMIP6 ESMs. To this aim, we have randomly resampled the data with replacement keeping the resample size equal to the size of the original sample (i.e., twelve). We then have computed the least-squares linear best fit for the resampled data, and repeated this procedure one-thousand times, and further derived the 68% confidence contours of equal probability based on the set of one-thousand random regression lines. By combining this 68% confidence contours estimated by bootstrapping with $\ln(B_{ts})$ observations and associated uncertainty, we have finally derived the probability density function of the classic emergent constraint on δET^{LULCC} .

Results based on the classic emergent constraint show that historical LULCC has led to a decrease in ET

with a global average rate of $-0.048 \text{ mm yr}^{-2}$ during the period 1982-2014, which is generally consistent with the current estimate derived based on the hierarchical emergent constraint ($-0.031 \text{ mm yr}^{-2}$) and shown in our manuscript. Moreover, both sets of results consistently show a reversal in the sign of original ESM estimates (0.057 mm yr^{-2}). The high consistency emerging among the two sets of results demonstrates the substantial independence of our results on the selection of emergent constraint type. In addition to the new experiment described above, the previous sets of sensitivity experiments performed on the selection of the model ensemble (by iteratively leaving one specific model out of the emergent relationship) and on the source of observational data also show a consistently sign switch of $\delta\text{ET}^{\text{LULCC}}$ after applying the emergent constraint. We believe that all these additional experiments together demonstrate the robustness of our constrained results about the negative value of global $\delta\text{ET}^{\text{LULCC}}$.

Furthermore, the inter-model spread after applying the classic emergent constraint ranges between -0.136 and 0.042 mm yr^{-2} expressed by the 68% interval assuming a Gaussian distribution and corresponds to a standard deviation of 0.089 mm yr^{-2} . The wider spread obtained when replacing the hierarchical emergent constraint by the classic one (0.081 mm yr^{-2} vs. 0.089 mm yr^{-2}) is consistent with findings reported in previous literature (Bowman et al., 2018). Such better performance of hierarchical emergent constraint arises from its analytical Bayesian conditioning and the absence of bootstrap-induced resampling noise. To minimize the uncertainty and further derive a more robust ESM-based assessment on the hydrological effect of LULCC, we therefore adopt the hierarchical emergent constraint instead of the classic one. To further illustrate the efficacy of the hierarchical emergent constraint in reducing inter-model spread, we have additionally performed the F -test in the revised version to evaluate whether the post-constraint variance in $\delta\text{ET}^{\text{LULCC}}$ estimate is significantly smaller than that of the original CMIP6 model ensemble. Test results show a p -value of 0.020, demonstrating that applying hierarchical emergent constraint could significantly reduce the uncertainty in $\delta\text{ET}^{\text{LULCC}}$ estimate originated from CMIP6 ESMs.

We point out that the LULCC effect at the global mean level is not statistically different from zero (one-sample t -test, $p > 0.05$), even after applying the emergent constraint. This situation occurs largely due to the fact that the contribution of LULCC on ET is relatively small at the global mean level, which is subject to spatial compensatory effects that may originate from areas with diverse LULCC and opposite ET response. In light of these considerations, we agree with the reviewer that the work should prominently focus on a regional-scale assessment. To this end, in the new round of revision, we have restructured the manuscript to focus more on the hotspots of LULCC and ET response. A detailed explanation has been given below.

Action taken: We have added the aforementioned results based on the classic emergent constraint in Supplementary Text 3 (“Results obtained within the classic emergent constraint framework” section), and briefly recalled them in the main text (see lines 205-212 and 867-872). To further improve the robustness of our emergent constraint results derived for the future scenario period, we have also used the classic emergent

constraint on the projections of afforestation impact on ET (δET^{AFF}) over 2015-2099. Correspondingly, we have added the new Supplementary Fig. 7 in the revised version. We have also added the F -test results in the main text (see lines 197-199). Furthermore, to make the manuscript more concise and readable and better focused on the core topic, we have streamlined the global-scale analyses in the main text and particularly moved most of the analyses related to the global impact of future afforestation into Supplementary Text 4 (“Details in constrained projections of global impact of future afforestation” section). Meanwhile, we have added more details about the results derived at the regional scale into the main text. The new contents have been structured in two Results sections to present the hotspot regions of LULCC and hydrological responses and to clarify the differences across regions in hydrological responses (“Emergent constraints at the regional scale” section and “Improved hydrological impact assessment under future afforestation scenarios” section).

Supplementary Fig. 7. Constrained results derived based on the classic emergent constraint approach. (a) Emergent relationship between the modelled effect of land use and land cover change on annual evapotranspiration (δET^{LULCC}) and the modelled natural logarithm value of global averaged transpiration-specific Bowen ratio ($\ln(B_{ts})$) during the period 1982-2014. LULCC over this period is generally characterized as the conversion from forests to croplands (Fig. 1a). Each dot denotes a CMIP6 ESM result, which corresponds to δET^{LULCC} estimate shown in Fig. 1b. The green solid line indicates the best-fit regression line across ESMs, with correlation coefficient (r) provided in the panel. Shaded areas around the best linear fit show the 68% confidence interval estimated by bootstrapping (Supplementary Text 3). The vertical blue dashed line and shaded areas represent the observation-based estimate of $\ln(B_{ts})$ and its uncertainty (one standard deviation). Such observation-based estimate is derived from an eight-member ensemble of observation-based combined datasets (Methods). The horizontal red dashed line shows the resulting constrained estimate of δET^{LULCC} based on the classic emergent constraint approach^{11,12}. (b) The probability density functions of global averaged δET^{LULCC} for the original results of CMIP6 ESMs (blue)

and the observationally constrained results (red). (c) and (d) Same as (a) and (b), but for the effect of future (2015-2099) afforestation on annual evapotranspiration (δET^{AFF}).

References:

1. Bowman, K., Cressie, N., Qu, X. & Hall, A. hierarchical statistical framework for emergent constraints: Application to snow-albedo feedback. *Geophys. Res. Lett.* **45**, 13050-13059 (2018).
2. Cox, P. et al. Sensitivity of tropical carbon to climate change constrained by carbon dioxide variability. *Nature* **494**, 341-344 (2013).
3. Cox, P., Huntingford, C. & Williamson, M. Emergent constraint on equilibrium climate sensitivity from global temperature variability. *Nature* **553**, 319-322 (2018).
4. Winkler, A., Myneni, R., Alexandrov, G. & Brovkin, V. Earth system models underestimate carbon fixation by plants in the high latitudes. *Nat. Commun.* **10**, 885 (2019).

Reviewer #2: Questions and our responses

I appreciate the effort the authors have put into revising the manuscript. I find it is much improved. I likewise appreciate the effort that the Authors have put into looking at and assessing such broad range of datasets and resources. The attempts at producing more realistic estimates of ET, in particular, are both compelling and fascinating. I do still have many questions. But I think it is appropriate to move forward to eventual publication of this submission.

I have still included some comments here and there in the pdf submission. Mostly minor things. But there are still some questions that continue to nag at me... I do think, for example, that some of the discussion about the relative variation in estimations of the Bowen Ratio (B_{ts}), for example, that are available in the Supplementary information, could be brought forward a bit. And I am still a bit concerned about our ability to estimate ET based on tree type, since there is SO MUCH variation in the real world empirical output that is not anywhere near adequately captured in the model simplifications...

That said, I do think the Authors do an exceptional job of putting this information together and assessing it in meaningful and important ways... Thus, though I would ask the Authors to consider my remaining comments, I do think this submission is more or less ready for publication.

We thank the reviewer for her/his overall positive comments.

In accordance to the reviewer's suggestions, we have moved the original Supplementary Fig. 5b to the main text (i.e., Fig. 2c) to help the readers better understand the difference of transpiration-specific Bowen ratio (B_{ts}) between ESM simulations and observation-based estimates.

We agree with the reviewer that model simplification of vegetation types could result in the bias of ET pattern and its response, and also the magnitude of B_{ts} mentioned above. Such model limitations exactly motivate us to use the hierarchical emergent constraint approach to correct the ESM-based signal of ET response, thereby providing a more robust assessment of global hydrological effect of massive land use change. In other words, the difference in magnitude and even direction of ET response before and after applying the emergent constraint framework is precisely the result of a reduction in the model bias arising from the simplification of vegetation types, uncertain parameterization and partial representation of some key biophysical processes (Duveiller et al., 2018; Forzieri et al., 2018a,b).

Specific comments and associated responses have been summarized below, according to the annotations shown in the pdf document uploaded.

Specific comments:

1. Line 60: Replace "contradicting" with "contradictory"

Response and action taken: We have changed “contradicting” to “contradictory”, as proposed by the reviewer (see line 60).

2. Line 60: Replace “that” with “i.e., that”

Response and action taken: We have modified the text according to the reviewer’s suggestion (see line 60).

3. Line 90: Add “an”

Response and action taken: We have added “an” in this sentence (see line 90).

4. Line 93: Add “the” between “over” and “reference regions”

Response and action taken: We have added “the” in this sentence (see line 93).

5. Line 103: Add “the” between “from” and “LUH2 dataset”

Response and action taken: We have added “the” in this sentence (see line 103).

6. Line 123: Replace “with” with “with regard to”

Response and action taken: We have modified the text according to the reviewer’s suggestion (see line 123).

7. Line 124: Replace “rising” with “raising”

Response and action taken: We have changed “rising” to “raising”, as proposed by the reviewer (see line 124).

8. Line 126: Add “further” before “underlies”

Response and action taken: We have added “further” in this sentence (see line 126).

9. Line 128: Replace “yet” with “and”

Response and action taken: We have modified the text according to the reviewer’s suggestion (see line 128).

10. Line 129 “global average signal ranging from 0.005 mm yr⁻² to 0.380 mm yr⁻², Fig. 1b”: were any of the signs negative?

Response: Given the subject of this sentence and its overall context, the values reported here correspond to the results derived based on the seven ESMs agreeing on the overall positive sign of δET^{LULCC} .

11. Line 130: Add “the” before “relative contribution”

Response and action taken: We have added “the” in this sentence (see line 130).

12. Line 136: Replace “negative” with “it is negative”

Response and action taken: We have modified the text according to the reviewer’s suggestion (see line 136).

13. Line 157: Delete “instead”

Response and action taken: We have deleted “instead” in this sentence (see line 157).

14. Line 160: Add “the” before “observation-based”

Response and action taken: We have added “the” in this sentence (see line 159).

15. Lines 169-170: These two points here and just above suggest that more analysis of variation in B_{ts} estimates may also be useful, as with the analysis of ET estimates...?

Response: Our study primarily focuses on the ET response to LULCC derived from ESM simulations, and applies the emergent constraint approach to correct such modelled ET response signal and reduce its uncertainty, thereby gaining a more robust assessment of the hydrological effect of massive land use change both at the global and regional scales. By contrast, despite the certain divergence in B_{ts} estimates across ESMs, correcting modelled B_{ts} signal makes not much sense, since observational estimates of B_{ts} are available which can provide us with the actual condition regarding B_{ts} magnitude at different spatial scales. The significance of the inter-model difference in B_{ts} for our study lies in its significant correlation with the modelled ET response signal (Fig. 2a), suggesting that discrepancy in the assessment of ET response to LULCC in CMIP6 ESM simulations stems from the model difference in representing the B_{ts} magnitude. Our experiments focusing on the difference in B_{ts} between cropland and forests reveal that the occurrence of an unrealistic increase in ET during the conversion from forests to croplands in model simulations may originate from the substantial underestimation of B_{ts} for croplands and/or the overestimation of B_{ts} for forests in ESMs (Fig. 6).

These results further demonstrate that the significant relationship between B_{ts} and ET response not only has strong statistical underpinnings, but also arises from a physical mechanism at work in the ESM ensemble (Fig. 6). These findings allow us to correct the modelled ET response to LULCC by introducing observed value of B_{ts} within the emergent constraint framework. In view of the above-mentioned considerations, our study does not focus excessively on the variation in B_{ts} estimates across ESMs, but rather on its inter-model relationship with the ET response.

16. Line 175: Replace “constrain” with “constraint”

Response and action taken: We have changed “constrain” to “constraint”, as proposed by the reviewer (see line 178).

17. Lines 175-176: How broadly do the B_{ts} measures range...? I see now that much of this is addressed in the Supplement... It would perhaps be meaningful to bring a little more of the information about variation in B_{ts} estimates forward to the text.

Response and action taken: The global mean B_{ts} based on the eight-member ensemble of observation-based datasets ranges from 1.178 to 1.726. Meanwhile, the global mean B_{ts} based on the CMIP6 model ensemble ranges from 1.422 to 3.897. Model simulations show a much higher overall magnitude of B_{ts} and its natural logarithm compared with observational estimates (1.989 ± 0.732 vs. 1.511 ± 0.197 for B_{ts} ; 0.633 ± 0.314 vs. 0.404 ± 0.137 for $\ln(B_{ts})$), suggesting a clear overestimation of B_{ts} in current-version ESMs. We agree with the reviewer that more information about the variation in B_{ts} estimates would help the readers to better understand and follow the work. To this end, we have moved the original Supplementary Fig. 5b to the main text (i.e., Fig. 2c), and added the description of associated results (see lines 168-170, 187 and 188).

18. Line 184: It looks like you have looked at some of this variation... But perhaps it should also be mentioned further above. Why assess ET variation, but not B_{ts} variation in the same ways...?

Response: We believe that our response and actions provided to the previous comments (i.e., Specific Comments #15 and #17) have already addresses this specific issue.

19. Line 209 “WA”: not sure I would use an acronym here.

Response: The acronym “WA”, which represents the terrestrial water availability, has already been defined in the Introduction section (see line 46). To be fully consistent with the legend shown in Fig. 5 and Supplementary Tables 9 and 11 and with formula expression shown in Methods (Eq. (11)), we have kept this

acronym in the revised version.

20. Line 211: Replace “origin” with “originate” or “derive”

Response and action taken: We have changed “origin” to “originate”, as proposed by the reviewer (see line 274).

21. Lines 213-319: These sentences could be better condensed/shortened.

Response and action taken: According to the reviewer’s suggestion, we have modified these sentences to improve the conciseness. Details are as follows: “Global forest area is projected to increase significantly ($p < 0.01$, Mann-Kendall test) during 2015-2099 under the SSP1-2.6 scenario, and to decline under the SSP3-7.0 scenario (Fig. 4a). Accordingly, forest area in SSP1-2.6 is higher than that in SSP3-7.0 over most of global land (Fig. 4b), reflecting the former’s afforestation-oriented sustainability policies versus the latter’s limited climate governance^{41,59} (details in Methods).” (see lines 275-280).

22. Line 216: Delete “part”

Response and action taken: We have deleted “part” in this sentence (see line 278).

23. Lines 226-228: This variation again suggests that the model variation in B_{ts} is important and needs more investigation.

Response: We agree with the reviewer that ESMs exhibit a clear divergence in B_{ts} estimates and that the parameter is also key to the model's representation of land-atmosphere interactions. In ESMs, B_{ts} is strongly related to the representation of water flows from soil to atmosphere, with stomata control of plant transpiration being a crucial step. This is recognized as a major source of uncertainty in vegetation models and certainly calls for increased efforts from the modelling community to improve the model representation of the processes underlying B_{ts} . Concerning the specific goal of this study, we consider that performing additional analyses to investigate the inter-model spread in B_{ts} estimates is outside the scope of our work, and is of limited significance, since observational estimates of B_{ts} are readily available. In contrast, our study focuses on the strong correlation between such inter-model differences in B_{ts} and those in ET response (i.e., δET^{LULCC} and δET^{AFF}) estimates (Fig. 2 and Supplementary Fig. 9), which suggests that model bias in representing B_{ts} results in an inaccurate assessment of ET response in CMIP6 simulations. More importantly, this strong correlation allows us to correct the modelled ET response by introducing observed value of B_{ts} within the emergent constraint framework, thereby gaining a more robust ESM-based assessment on the hydrological effect of

massive land use change.

24. Lines 288-290: And what might the effects on the estimation of B_{ts} be...?

Response: Different from ET response (i.e., δET^{LULCC} and δET^{AFF}) for which no direct observations exist, observational estimates of B_{ts} are available. A more robust estimate of B_{ts} can be directly obtained based on the observation-based datasets, without relying on the emergent constraint approach. Therefore, we do not consider meaningful to apply the emergent constraint to the estimation of B_{ts} .

25. Lines 327-331: Perhaps highlight in which regions.

Response: According to the reviewer's suggestion, we have specified the regions where the negative value of δET^{AFF} still exists after constraint (see lines 321-325). These specific regions include Northeastern South America (NES) and Southeast Asia (SEA).

26. Line 364: Add "the" before "LULCC effect"

Response and action taken: We have added "the" in this sentence (see line 367).

27. Line 365: Add "that" after "LULCC"

Response and action taken: We have added "that" in this sentence (see line 368).

28. Line 366: Replace "are" with "is"

Response and action taken: We have modified the text according to the reviewer's suggestion (see line 369).

29. Line 483 "is difficult to leave": Awkward, revise.

Response and action taken: According to the reviewer's suggestion, we have modified this sentence to improve its clarity. Details are as follow: "In addition, for regions like Tibetan Plateau, moisture tends to be sustained because the prevailing winds blow towards the mountains, and the orographic lifting of moisture leads to repeated rainfall⁷⁶." (see lines 487-489).

30. Line 484: Delete "the"

Response and action taken: We have deleted “the” in this sentence (see line 489).

31. Line 491: Replace “based on” with “to”

Response and action taken: We have modified the text according to the reviewer’s suggestion (see line 495).

32. Line 657 “Observation-based products”: I do not know how carefully these different ET sources are compared in the analysis... I imagine that there is quite significant variation across these different estimation and measurement types and would really like to see a careful analysis of this... I suppose one could write a paper just on this... The variation, trend divergence and implications are significant for the estimation of climate and P impacts... I understand that this presumably extends well beyond the parameters of this analysis. But my guess is that there is very significant variation here... And presumably this is the core of the problem with identifying and using appropriate ET estimates in the ESM's... My guess is that such measures also affect the estimation of the B_{ts} as well... In looking over the data in the Supplement, I see that these suspicions are all true... And the variation is quite significant across the various sources... I do appreciate the attention to the details of analysis here...

Response: The magnitude of mean annual ET during the period 1982-2014 at the global extent shows certain difference across the four observation-based datasets used in this study (i.e., LAI-based upscaling, GLEAM v3.8a, Gerrits’ model, and PML-V2), ranging from 516.7 mm yr⁻¹ to 572.2 mm yr⁻¹. Given the fact that our study primarily focuses on the observationally constrained assessment of the impact of massive land use change on ET and terrestrial water availability, a detailed inter-comparison of the spatiotemporal patterns among these ET datasets is therefore beyond the scope of this study. We thank the reviewer for this valuable suggestion, which will motivate and guide our future research. The spread among ET datasets also results in the divergence in the corresponding mean annual B_{ts} estimates, ranging from 1.178 to 1.726, with an ensemble level of 1.511±0.197 (mean±s.d.). These findings shown above are aligned with the reviewer’s intuition.

We acknowledge the uncertainty inherent in “observed” ET and B_{ts}, and therefore we believe that it is essential to consider different observation-based datasets in our study. Using an ensemble mean of different observation-based datasets provides a more robust benchmark than any single dataset, as ensemble averaging has been proved to reduce random errors and noise and smooth dataset-specific biases (Pan et al., 2020; Cai et al., 2024). Therefore, the ensemble mean serves as an appropriate reference when applying the emergent constraint (i.e., the vertical blue line in Fig. 2). Moreover, comparing B_{ts} estimates across observation-based datasets allow us to quantify the observational uncertainty. The hierarchical emergent constraint framework explicitly considers this uncertainty and propagates into the final result, thereby deriving a more robust observationally constrained assessment (i.e., $\sigma^2(x_0)$ in Eqs. (6)-(8)).

References:

1. Cai, Y. et al. Reconciling global terrestrial evapotranspiration estimates from multi-product intercomparison and evaluation. *Water Resour. Res.* **60**, e2024WR037608 (2024).
2. Duveiller, G. et al. Biophysics and vegetation cover change: a process-based evaluation framework for confronting land surface models with satellite observations. *Earth Syst. Sci. Data* **10**, 1265-1279 (2018).
3. Forzieri, G. et al. Evaluating the interplay between biophysical processes and leaf area changes in land surface models. *J. Adv. Model. Earth Syst.* **10**, 1102-1126 (2018a).
4. Forzieri, G., Alkama, R., Miralles, D. & Cescatti, A. Response to Comment on “Satellites reveal contrasting responses of regional climate to the widespread greening of Earth”. *Science* **360**, eaap9664 (2018b).
5. Pan, S. et al. Evaluation of global terrestrial evapotranspiration using state-of-the-art approaches in remote sensing, machine learning and land surface modeling. *Hydrol. Earth Syst. Sci.* **24**, 1485-1509 (2020).

Reviewer #1: Comments

I thank the authors for the detailed additional analyses and the restructuring/revisions of their manuscript. I appreciate that the authors took my concerns about presenting a global constraint that is not sig. different from zero as the main finding into account and agreed that the work should prominently focus on a regional-scale assessment. Thank you for the constructive discussion here. After three rounds of review and given the rebuttals to Reviewer 2 also, this work is ready for publication in Nature Communications.

We thank the reviewer for her/his overall positive comments.

Reviewer #2: Comments and our responses

I am impressed at the lengths the Authors have gone to, to respond to all the very detailed comments of both Reviewers (myself included). I think this is an impressive piece of work and look forward to seeing it in print. I do find that I am repeatedly inspired, in reading this text, to think about further issues that could be addressed and discussed. But this is perhaps the best reflection of the fact that this is an important piece of work and should further inspire others to take these ideas and points still further in future endeavours. Thus, rather than go into additional detailed comments, I would like to encourage publication at this time and hope that others will find this publication as thought provoking as I have.

I did notice here and there that some very minor editing of the manuscript is still possible... The attached provides some indication of possible improvements. Please note that I have made no attempt to be exhaustive. Thus more editing is likely possible... However, these editing issues are minor and not worthy of further discussion.

We thank the reviewer for her/his overall positive comments. Specific comments and associated responses have been summarized below, according to the annotations shown in the pdf document uploaded.

Specific comments:

1. Line 18: Replace “effect” with “effects”

Response and action taken: To meet the journal’s requirement for Abstract section, we have modified this sentence completely, and the word “effect” is no longer used.

2. Line 20: Replace “effect” with “effects”

Response and action taken: To comply with the journal’s word limit for Abstract section, we have deleted this sentence.

3. Line 46: If anthropogenic change includes re/afforestation, there can also be positive effects... This may suggest only negative effects are the likely outcome...

Response and action taken: We agree with the reviewer that LULCC-driven processes can generate not only negative but also positive impacts on socio-ecological systems and services. To avoid this one-sided interpretation, we have modified this sentence to present a more balanced perspective. Details are as follows: “Such LULCC-driven processes may lead to shifts in soil moisture and total terrestrial water availability (WA, defined as P minus ET)^{5,9} and trigger cascading detrimental or beneficial impacts on socio-ecological systems

and services crucial for the human well-being, such as biodiversity conservation¹¹, carbon uptake and storage^{12,13}, food security and timber production¹⁴.” (see lines 34-39).

4. Line 51: New sentence

Response and action taken: According to the reviewer’s suggestion, we have split the original sentence into two separate sentences (see lines 39-43).

5. Line 122: Replace “effect” with “effects” and replace “is” with “are”

Response and action taken: We have modified the text according to the reviewer’s suggestion (see line 112).

6. Line 126: Replace “underlies” with “underscores”

Response and action taken: We have modified the text according to the reviewer’s suggestion (see line 116).

7. Line 150: Replace “effect” with “effects”

Response and action taken: We have modified the text according to the reviewer’s suggestion (see line 140).

8. Line 295: Add “the” before “original CMIP6 model ensemble”

Response and action taken: We have added “the” in this sentence (see line 285).

9. Line 429: Replace “that” with “the”

Response and action taken: We have modified the text according to the reviewer’s suggestion (see line 419).